# A chaperone-proteasome-based fragmentation machinery is essential for aggrephagy

Mario Mauthe [1] ✉, Nicole van de Beek[1,2], Muriel Mari[1,2], Giel Korsten[3], Parisa Nobari[1], Kennith B. Castelino[2], Eduardo P. de Mattos[1], Ibtisam Ouhida[1], Jesse L. Dijkstra[1], Sabine Schipper-Krom[4], Laura R. de la Ballina [5,6], Monja R. Mueller[1,2], Anne Simonsen[5,6], Mark S. Hipp [1,7], Lukas C. Kapitein [3], Harm H. Kampinga [1] ✉ & Fulvio Reggiori [1,2,8] ✉

Perturbations in protein quality control lead to the accumulation of misfolded proteins and protein aggregates, which can compromise health and lifespan. One key mechanism eliminating protein aggregates is aggrephagy, a selective type of autophagy. Here we reveal that fragmentation is required before autophagic clearance of various types of amorphous aggregates. This fragmentation requires both the 19S proteasomal regulatory particle and the DNAJB6-HSP70-HSP110 chaperone module. These two players are also essential for aggregate compaction that leads to the clustering of the selective autophagy receptors, which initiates the autophagic removal of the aggregates. We also found that the same players delay the formation of disease-associated huntingtin inclusions. This study assigns a novel function to the 19S regulatory particle and the DNAJB6-HSP70-HSP110 module, and uncovers that aggrephagy entails a piecemeal process, with relevance for proteinopathies.

Protein aggregates are considered as non-functional assemblies that can adopt amyloid or amorphous conformations, depending on the stress conditions and structural features of the proteins that trigger the aggregation process[1,2]. Amorphous conformations distinguish from amyloid ones because they lack highly-ordered structures such as cross-β secondary structures and fibrillar morphology[1,2]. Aggregates often conglomerate into larger entities, here referred to as inclusions. Aggregation is known to disrupt multiple cellular pathways, leading to dysfunction and disease[3-7]. Thus, it is vital for organisms to dissolve or eliminate protein aggregates.

Molecular chaperones play a central role in the cell's protein quality control by assisting protein folding[8], preventing protein aggregation and facilitating misfolded protein disposal by the ubiquitin-proteasome system (UPS) or chaperone-mediated autophagy (CMA)[9]. Central to the UPS-mediated degradation are 26S proteasomes, which are composed of one 20S core particle (CP) with the proteolytic activity and one or two 19S regulatory particles (RPs)[10]. 19S RPs bind ubiquitinated substrates and deubiquitinate them to recycle ubiquitin (Ub) while their ATPases unfold and convey the substrates into the 20S CP for degradation[10].

[1]Department of Biomedical Sciences, University of Groningen, University Medical Center Groningen, Groningen, The Netherlands. [2]Department of Biomedicine, Aarhus University, Aarhus, Denmark. [3]Department of Biology, Section Cell Biology, Neurobiology and Biophysics, University of Utrecht, Utrecht, The Netherlands. [4]Department of Medical Biology, University of Amsterdam, Academic Medical Center, Amsterdam, The Netherlands. [5]Department of Molecular Cell Biology, Institute for Cancer Research, The Norwegian Radium Hospital, Montebello, Oslo, Norway. [6]Centre for Cancer Cell Reprogramming, Division for Cancer Medicine, Institute for Clinical Medicine, University of Oslo, Oslo, Norway. [7]School of Medicine and Health Sciences, Carl von Ossietzky University Oldenburg, Oldenburg, Germany. [8]Aarhus Institute of Advanced Studies (AIAS), Aarhus University, Aarhus, Denmark. ✉e-mail: m.mauthe@umcg.nl; h.h.kampinga@umcg.nl; f.reggiori@aias.au.dk

Macroautophagy (hereafter autophagy) is essential for the lysosomal clearance of dysfunctional intracellular material, including protein aggregates[11]. Through specific recognition by selective autophagy receptors (SARs), the autophagy-related (ATG) machinery is recruited for autophagosome formation around the targeted cargo[12]. Aggrephagy, the selective autophagic degradation of protein aggregates involves aggregate ubiquitination and subsequent recognition by SARs such as p62/SQSTM1, NDP52 and TAX1BP1 in most cases[9,12,13]; however, recognition can also be Ub-independent[14]. Findings in animal models and patients with loss of a SAR or an *ATG* gene results in protein aggregate accumulation in various organs[15–19], highlighting the physiological relevance of aggrephagy. However, the large amyloid-like aggregates seen in cells expressing aggregation-prone proteins, characteristic of proteinopathies such as Alzheimer's disease, Parkinson's disease and Huntington's disease, are persistent, suggesting that they are more resilient to aggrephagic degradation[20].

An alternative way of eliminating protein aggregates is through disaggregation, a process in which chaperones such as HSP70 extract individual polypeptides from aggregates[21,22]. While refolding after disaggregation is typically preferred[23,24], UPS-mediated degradation is also possible[25]. A disaggregation machinery was first described in yeast, where Hsp104 is essential to dissolve aggregated proteins after heat shock[26–29]. In metazoa that lack Hsp104, specific HSP70 co-chaperones such as HSP110 nucleotide exchange factor and DNAJ proteins equip HSP70 with disaggregation activity[21,22]. There are approximately 50 mammalian J-domain proteins (JDPs) but so far, only DNAJA1, DNAJA2, DNAJB1 and DNAJB4 have been shown to participate in disaggregation of amorphous or amyloid-like aggregates[22,30–37].

While aggrephagy and disaggregation have been studied independently, it remains to be understood whether a functional link between them exists. Investigating the turnover of a chemically inducible aggregate[38,39], we found that DNAJB6, together with HSP70 and HSP110, is crucial for aggregate fragmentation, but also requires proteasomal 19S RPs. Their combined actions are also paralleled by cargo compaction that, in turn, is critical for SAR clustering, a prerequisite for the subsequent formation of autophagosomes. As the DNAJB6-HSP70-HSP110 module and 19S RPs are also required for the autophagic removal of puromycin-induced aggregates and in lessening the accumulation of disease-associated huntingtin aggregates, our study reveals the existence of a fragmentation machinery, or fragmentase, with a widespread role in aggrephagy.

## Results

### A subset of chaperones and co-chaperones is required for the fragmentation and lysosomal turnover of amorphous aggregates

To uncover the molecular principles underlying aggrephagy, we employed the chemically inducible Particles Induced by Multimerization (PIM) system, similar to the AgDD system[40], which is based on multiple FKBP domains and their property of multimerizing upon rapalog2 addition[41]. Protein aggregates are rapidly formed upon rapalog2 treatment at concentrations that do not inhibit mTOR and therefore do not induce autophagy[38]. By fusion to a tandem mCherry–green fluorescent protein (GFP) tag, lysosomal delivery can be detected by the loss of the GFP signal due to its quenching in acidic compartments[38] (Fig. 1a). We integrated the mCherry–GFP-tagged PIM construct in Flp-In T-Rex U2OS cells, generating a dualPIM cell line, with a tetracycline (TTC)-regulatable expression of the non-toxic PIM reporter (Extended Data Fig. 1a). As in transiently transfected cells[38], the dualPIM cells rapidly formed mCherry–GFP-positive puncta upon rapalog2 addition for 30 min (Extended Data Fig. 1b). From 2 h onwards, the first mCherry-only puncta (magenta) appeared indicative of lysosomal uptake and these numbers increased over time, with lysosomal localization being complete approximately 24 h after rapalog2 treatment (Extended Data Fig. 1b–d). The automated measurement of the conversion of mCherry–GFP into mCherry-positive puncta by live-cell imaging can be used as a proxy to measure lysosomal dualPIM aggregate degradation over time (Extended Data Fig. 1c,d) and to calculate degradation rates (Extended Data Fig. 1e). Treatment with the autophagy inhibitor Bafilomycin A1 (BAF)[42] or SAR405 (ref. 43) blocked dualPIM aggregate turnover confirming they indeed are delivered into lysosome (Extended Data Fig. 1f–h). Immuno-electron microscopy (IEM) using antibodies against GFP and LAMP2, a lysosomal marker protein (Fig. 1b) revealed that dualPIM puncta have an amorphous structure, with diameters of around 150–450 nm at 30 min after rapalog2 treatment and coalesce into clusters of 1.0–1.6 µm after 2 h (Fig. 1b). In agreement with the live-cell imaging, amorphous dualPIM aggregates were detected inside LAMP2-positive lysosomes at 6 h after rapalog2 treatment (Fig. 1b).

A question that emerged from the IEM analyses was how cytoplasmic dualPIM aggregates that have diameters ranging from 1.0–1.6 µm can be delivered into the interior of lysosomes, which have a diameter of 0.1–0.6 µm in U2OS cells. We hypothesized that they may first need to undergo fragmentation, a notion indirectly supported by the observation that larger dualPIM aggregates persist (Extended Data Fig. 1i). Live-cell microscopy revealed that large puncta indeed are not delivered en bloc into lysosomes. Rather, small dualPIM fragments detach from larger clusters before lysosomal targeting (Fig. 1c, Extended Data Fig. 1j and Supplementary Video 1). This first in vivo visual demonstration of a cytoplasmic fragmentation activity in mammalian cells incited us to further investigate this process.

HSP70 is a central component of disaggregation machineries[44], which have been shown to be able to fragment specific aggregates in vitro[22,30,35,37]. Indeed, treatment with pharmacological inhibition of HSP70 using VER-155008 (VER)[45] led to the cytoplasmic accumulation of dualPIM aggregates, which coalesced into big inclusions that were not delivered into lysosomes (Fig. 1e,f). In contrast, hindering actin filaments and microtubules, which are crucial for the formation and disassembly of aggresomes, stress granules and JUNQ bodies[46–49], using

**Fig. 1 | A specific subset of chaperones and co-chaperones are required for the fragmentation and subsequent lysosomal turnover of amorphous aggregates. a**, Schematic description of the dualPIM system. **b**, IEM analyses of dualPIM aggregates induced for 30 min, 2 h or 6 h. Cryosections were labelled individually with anti-GFP (30 min and 2 h time points) or in combination with anti-LAMP2 antibodies (6 h time point). Insets are shown below the overviews. Scale bars, 500 nm (overviews) and 250 nm (insets). **c**, HeLa cells transiently expressing dualPIM were imaged until 16 h after triggering aggregation with rapa2 treatment. Time-lapse images showing multiple fragmentation (white arrows) and degradation (white arrowhead) events are presented in the panel. See also Supplementary Video 1 and Extended Data Fig. 1f. CM, control medium. **d**, DualPIM cells were treated ± 40 µM VER for 19 h. **e,f**, Quantification of the average dualPIM puncta size at 19 h (**e**) and degradation rates (**f**). *P* = 0.0055 (**e**); *P* < 0.0001 (**f**). **g**, DualPIM cells were transfected with the indicated siRNAs (also Supplementary Table 1) and imaged for 24 h. Representative images at 24 h are shown. **h,i**, Quantification of dualPIM puncta average size at 24 h (*P* values versus siCtr: *P* = 0.001 (siCtr + BAF); *P* = 0.024 (siHSP70/siHSPA1A); *P* < 0.001 (siHSP110); *P* < 0.001 (siDNAJA2); *P* < 0.001 (siDNAJB6)) (**h**) and degradation rates (*P* values versus siCtr: *P* < 0.001 (siCtr + BAF); *P* = 0.002 (siHSP70/siHSPA1A); *P* < 0.001 (siHSP110); *P* = 0.004 (siBAG3); *P* < 0.001 (siDNAJB6)) (**i**) of the experiment are shown. Images in **c** were acquired using the Nikon Eclipse microscope and images from **d** and **g** using the IncuCyte S3 system. Data are presented as mean ± s.d. Error bars represent the s.d. of three (**e,f,h,i**, siHSPB1, siHSPB7, siHSP110, siDNAJA1, siDNAJA2, siDNAJB1 and siDNAJB4 condition), four (**h,i**, siHSPBP1, siHSP70/siHSPA1A, siBAG3, siDNAJB2 and siDNAJB6 condition), eight (**h,i**, siCtr condition) or ten (**h,i**, siCtr + BAF condition) independent experiments. *\*P* < 0.05, *\*\*P* < 0.01 and *\*\*\*P* < 0.001. *P* values were calculated by a two-tailed *t*-test with Bonferroni correction (**e,f**) or one-way analysis of variance (ANOVA) following Dunnett's post hoc test (**h,i**). Scale bars, 20 µm, except in **a** (5 µm).

cytochalasin D and nocodazole, respectively, did not affect dualPIM aggregate fragmentation and only moderately their lysosomal delivery (Extended Data Fig. 1f–h).

HSP70 activity is guided and enhanced by co-chaperones and other cofactors[44,50]. To determine which ones are required for dual-PIM aggregate fragmentation, we depleted several of them by siRNA

(Extended Data Fig. 2a) before assessing dualPIM aggrephagy. While BAF treatment did not increase but rather decreased dualPIM puncta size (but blocked dualPIM turnover) (Fig. 1h,i), HSPA1A (HSP70) knockdown inhibited fragmentation and degradation, whereas deple-tion of its homologue HSPA8 (HSC70) had no effect (Extended Data Fig. 2b). Silencing of most of the other tested chaperones had no effect,

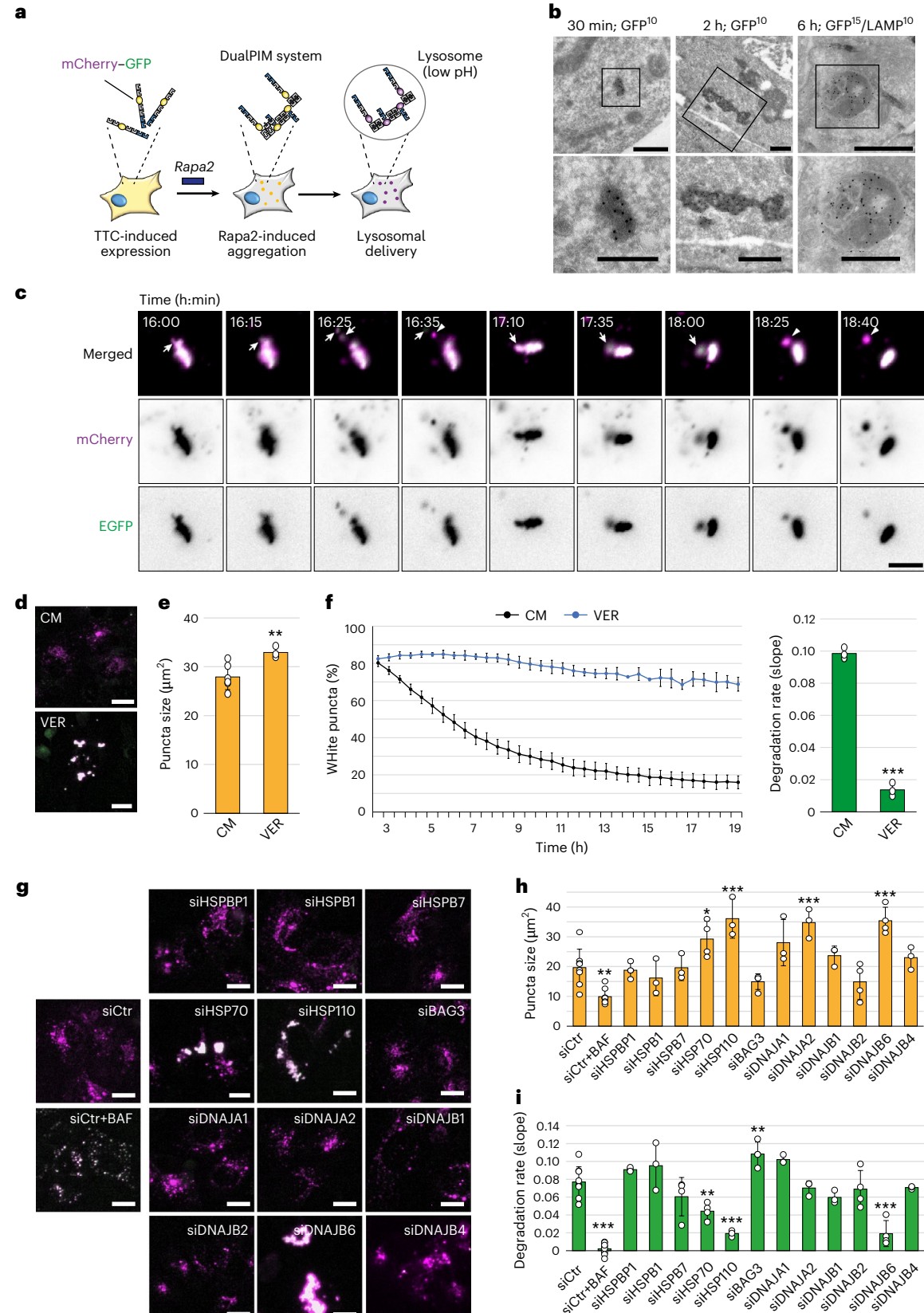

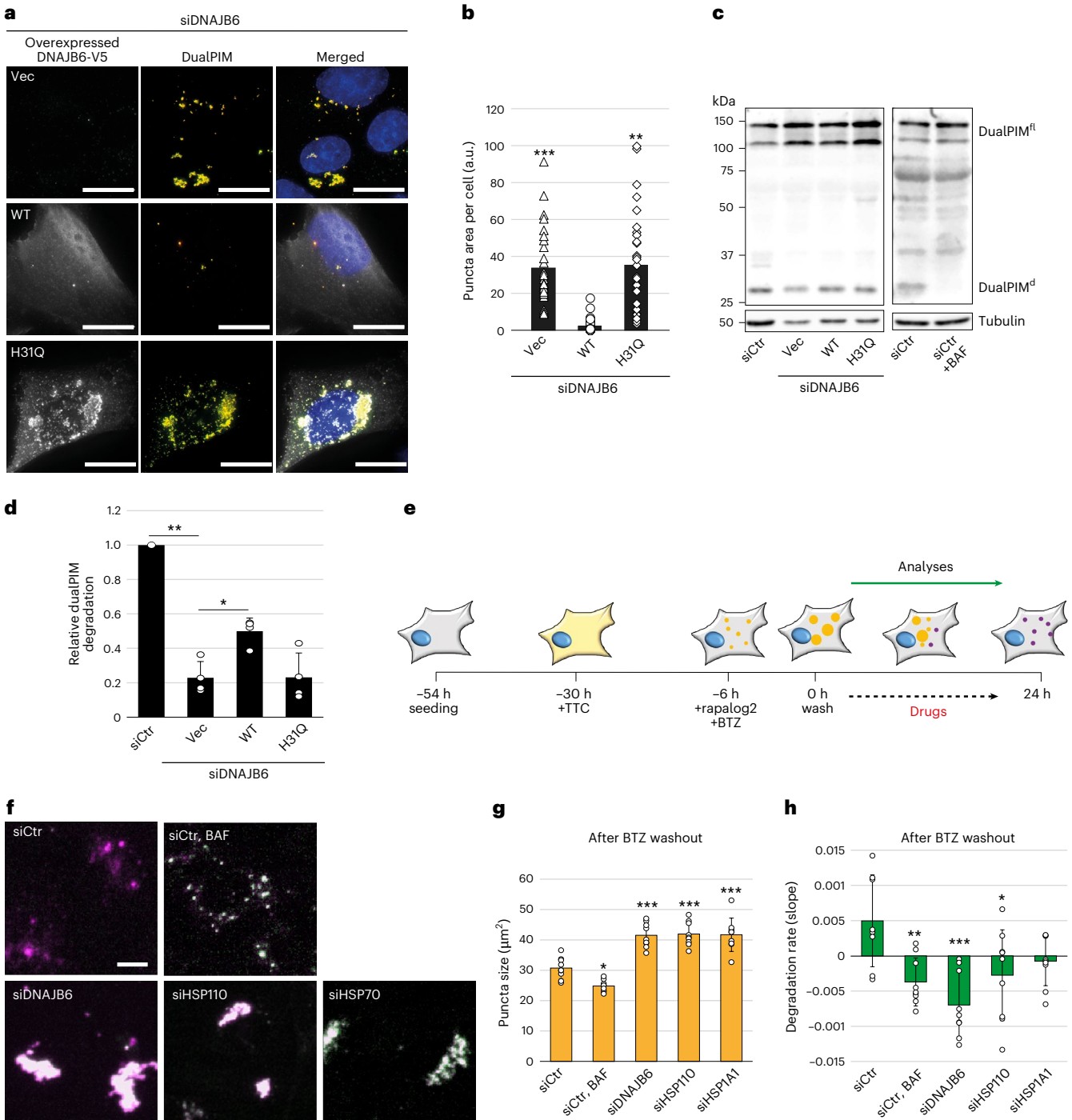

**Fig. 2 | DNAJB6 engages HSP70 and HSP110 to convey dualPIM aggregates to lysosomes. a**, DualPIM cells were transfected with the siDNAJB6 and transfected with an empty vector (Vec) or a plasmid overexpressing either V5-DNAJB6[WT] (WT) or V5-DNAJB6[H31Q] (H31Q) for 24 h, and aggregation was induced for 6 h. Representative images of anti-V5-stained cells are shown. **b**, The area of dualPIM puncta signal per cell was quantified and compared with WT. $P < 0.001$ (WT versus Vec); $P = 0.0024$ (WT versus H31Q). **c**, DualPIM cells were transfected with siCtr or siDNAJB6 (for validation of the knockdown and transfection see Extended Data Fig. 3a) and transfected with Vec or a plasmid expressing either V5-DNAJB6[WT] (WT) or V5-DNAJB6[H31Q] (H31Q) for 24 h before aggregation was induced for 20 h. As a control, cells were treated with 100 nM BAF. **d**, Relative dualPIM aggregate degradation was quantified as in Extended Data Fig. 3c and values are relative to the siCtr control. DualPIM[fl], full-length dualPIM; dualPIM[d], lysosomal dualPIM degradation product. $P = 0.0015$ (siCtr versus Vec); $P = 0.015$ (WT versus Vec). **e**, Schematic description of the experiment design, showing when cells were

seeded, treated with BTZ, washed, treated with drugs and analysed. **f**, DualPIM cells were transfected with siCtr, siDNAJB6, siHSP110 (combination of siHSPH1, siHSPH2, siHSPH3) or siHSPA1A, and aggregation was induced in the presence of 100 nM BTZ for 6 h before washing BTZ away and imaging over time; 100 nM BAF was added to siCtr-treated dualPIM cells as a control. Representative images at 17 h post-BTZ washout. **g,h**, Quantifications of dualPIM puncta average size at 17 h ($P$ values versus siCtr: $P = 0.01$ (siCtr + BAF); $P < 0.001$ (siDNAJB6); $P < 0.001$ (siHSP110); $P < 0.001$ (siHSP70)) (**g**) and the degradation rates ($P$ values compared with siCtr: $P = 0.04$ (siCtr + BAF); $P < 0.001$ (siDNAJB6); $P < 0.011$ (siHSP110); $P < 0.078$ (siHSP70)) in the experiment are shown (**h**). Images in **a** were acquired using the DeltaVision microscope and images in **f** using the IncuCyte S3 system. Data are mean ± s.d. Error bars show s.d. of three (**b,d**) and eight (**g,h**) independent experiments. *$P < 0.05$, **$P < 0.01$ and ***$P < 0.001$. $P$ values were calculated by a two-tailed $t$-test with Bonferroni correction (**b,d**) or one-way ANOVA following Dunnett's post hoc test (**g,h**). Scale bars, 20 μm.

including HSPB7, which has been associated with autophagic clearance of aggregates[51], and the HSP70 co-chaperone BAG3, a nucleotide exchange factor (NEF) of HSP70 and component of CASA/BIPASS, which targets ubiquitinylated, HSP70-bound protein substrates for autophagic degradation upon proteasome inhibition[52–54]. Consistently, specific inhibition of HSP70 interaction with all members of the BAG protein family using JG-98 (ref. 55) had no effect on dualPIM aggregate degradation (Extended Data Fig. 2c–e). Instead, siRNAs targeting HSP110 (HSPH1, HSPH2 and HSPH3 in combination), which has been suggested to be the critical NEF in HSP70-dependent disaggregation and fragmentation[22,30,37,56] led to both dualPIM aggregate accumulation into inclusions and prevented lysosomal delivery (Fig. 1g,i). Notably, neither depletion of DNAJB1, a member of the JDPs, key in both fragmentation of preformed α-synuclein aggregates in vitro[21,22,31,32] and many other HSP70-dependent disaggregation activities[31,32,37,57], nor most other JDPs, impeded dualPIM fragmentation and degradation.

To exclude known functional redundancies, we also simultaneously depleted DNAJB1 together with DNAJA1 and DNAJA2 (Extended Data Fig. 2f–h) but no degradation impairment was observed. Although the depletion of the tested proteins was effective (Extended Data Fig. 2b), we cannot exclude that some of them may have a residual activity sufficient to partially sustain their function. However, in DNAJB6-depleted cells, dualPIM inclusions remained large and did not end up in lysosomes (Fig. 1g,i). In line, colocalization of dualPIM puncta with LAMP1-positive lysosomes was significantly reduced (Extended Data Fig. 2i), suggesting that lysosomal degradation of dualPIM aggregates requires fragmentation by HSP70-HSP110 module and DNAJB6.

## DNAJB6 requires HSP70 and HSP110 for fragmentation

To determine whether DNAJB6 cooperated with the HSP110-HSP70 module, we turned to a DNAJB6 point mutant, DNAJB6[H31Q], which cannot interact with and transfer substrates to HSP70 (ref. 58). DNAJB6-depleted cells were transfected with an empty vector or a plasmid overexpressing DNAJB6[WT] or DNAJB6[H31Q] before examining dualPIM puncta (Fig. 2a). While smaller in cells complemented with DNAJB6[WT], dualPIM puncta size remained large in cells expressing DNAJB6[H31Q], like in cells back-transfected with the empty vector (Fig. 2b). A similar phenotype was observed upon DNAJB6[WT] or DNAJB6[H31Q] overexpression in U2OS cells (Extended Data Fig. 3b). While we could not detect DNAJB6[WT] on dualPIM puncta, the DNAJB6[H31Q] mutant was strongly associated with them (Fig. 2a and Extended Data Fig. 3b), suggesting that DNAJB6 dynamically interacts with dualPIM aggregates independently from HSP70 but gets trapped there when it cannot interact with HSP70. We also assessed the lysosomal dualPIM aggregate turnover by western blot analysis. In dualPIM cells, a 25-kDa mCherry fragment formed 6 h post-rapalog2 treatment that was absent if BAF was added, showing

that this fragment is a lysosomal degradation product (Extended Data Fig. 3c). DNAJB6 depletion significantly reduced this degradation, which was partially reversed by back-transfections of DNAJB6[WT] but not DNAJB6[H31Q] (Fig. 2c,d). We also examined whether DNAJB6 is required to recruit HSP70 to dualPIM puncta and found that this is not the case (Extended Data Fig. 3d). We detected even more dualPIM puncta positive for HSP70 in DNAJB6 knockdown cells, suggesting that blocked degradation could perturb HSP70 dynamics on aggregates. HSP70 still colocalized with dualPIM puncta in cells treated with BAF, indicating that HSP70 does not disassociate from these aggregates (Extended Data Fig. 3e). Thus, although DNAJB6 is not required for HSP70 recruitment to aggregates per se, the local DNAJB6-HSP70 interaction is crucial for lysosomal dualPIM aggregate delivery, likely by facilitating HSP70-dependent aggregate fragmentation.

To determine whether DNAJB6, HSP70 and HSP110 are indeed required to fragment the dualPIM aggregates and prime them for lysosomal clearance, we took advantage of the observation that the reversible proteasome inhibitor BTZ at high concentrations (50–100 nM) induces the formation of large dualPIM inclusions (Fig. 2e and Extended Data Fig. 4a–c). We first performed a BTZ serial dilution experiment in both dualPIM cells and HEK cells stably expressing the proteasomal activity reporter Ub-R-GFP that is degraded the Ub fusion degradation pathway[59]. Proteasome activity was significantly blocked at 12.5 nM and 25 nM BTZ, whereas fragmentation was unaffected (Extended Data Fig. 4a,b). Notably, a significant effect on dualPIM puncta size was seen at higher BTZ concentrations (50 and 100 nM), suggesting a possible proteasomal involvement in fragmentation. Upon washing away BTZ after 6 h, these large dualPIM inclusions were cleared through the generation into small fragments before delivery into lysosomes (Extended Data Fig. 4f and Supplementary Video 2). In contrast, dualPIM inclusion degradation was completely blocked when the BTZ washout was performed in the presence of BAF or VER (Extended Data Fig. 4e). However, while dualPIM inclusions were normally fragmented in BAF-treated cells, VER presence inhibited fragmentation (Extended Data Fig. 4c,d). We repeated the BTZ washout experiment in dualPIM cells depleted of DNAJB6, HSP70 or HSP110 (Fig. 2f,g) and found that lysosomal dualPIM aggregate turnover was inhibited (Fig. 2h). These data indicated that HSP70, HSP110 and DNAJB6 form a fragmentation module that is required for lysosomal degradation of large inclusions.

## Aggregate fragmentation requires 19S RPs

Next, we asked whether the DNAJB6-HSP70-HSP110 machinery needs the assistance of an ATPase for fragmentation as in yeast, where Hsp70 collaborates with Hsp104 (refs. 24,28). Inhibition of the valosin-containing protein (VCP/p97), a member of AAA-ATPase family that has been shown to be required for the disaggregation of

**Fig. 3 | The 19S RP, but not the 20S CP, is required for dualPIM aggregate fragmentation. a**, DualPIM puncta size was measured upon knockdown of the indicated siRNAs (Supplementary Table 1) after 19 h of aggregate induction in dualPIM cells. *P* values versus siCtr: *P* < 0.001 (siPSMC1); *P* < 0.001 (siPSMC2); *P* < 0.001 (siPSMC3); *P* < 0.001 (siPSMC4); *P* < 0.001 (siPSMC5); *P* < 0.001 (siPSMC6); *P* < 0.001 (siPSMD1); *P* < 0.001 (siPSMD2); *P* = 0.002 (siPSMD3); *P* < 0.001 (siPSMD6); *P* < 0.001 (siPSMD7); *P* < 0.001 (siPSMD8); *P* < 0.001 (siPSMD11); *P* < 0.001 (siPSMD14). **b**, GFP-positive cell areas were measured after knockdown of the indicated siRNAs in the Ub-R-GFP cells. *P* values versus siCtr: *P* < 0.001 (siPSMA1); *P* < 0.001 (siPSMA2); *P* < 0.001 (siPSMA3); *P* < 0.001 (siPSMA6); *P* < 0.001 (siPSMA7); *P* < 0.001 (siPSMB1); *P* < 0.001 (siPSMB2); *P* < 0.001 (siPSMB3); *P* < 0.001 (siPSMB4); *P* < 0.001 (siPSMB5); *P* < 0.001 (siPSMB6); *P* < 0.001 (siPSMB7); *P* < 0.001 (siPSMC1); *P* < 0.001 (siPSMC2); *P* < 0.001 (siPSMC3); *P* < 0.001 (siPSMC4); *P* < 0.001 (siPSMC5); *P* < 0.001 (siPSMC6); *P* < 0.001 (siPSMD1); *P* < 0.001 (siPSMD2); *P* < 0.001 (siPSMD3); *P* < 0.001 (siPSMD4); *P* < 0.001 (siPSMD6); *P* < 0.001 (siPSMD7); *P* < 0.001 (siPSMD8); *P* < 0.001 (siPSMD11); *P* < 0.001 (siPSMD12); *P* < 0.001 (siPSMD13); *P* < 0.001 (siPSMD14). **c**, Schematic model of a 26S proteasome indicating the 19S

RP and the 20S CP, also listing the subunits examined in this study. **d**, Representative images at 24 h are presented for the experiment shown in **a**. **e**, DualPIM cells were transfected with siCtr, siPSMB2, siPSMB5, siPSMC1 or siPSMC5, and aggregation was induced for 20 h. Relative dualPIM aggregate degradation was quantified as in Fig. 2d and values are relative to the siCtr control. *P* values versus siCtr: *P* = 0.032 (siPSMC1); *P* = 0.036 (siPSMC5). **f**, DualPIM cells were transfected with siCtr, siPSMB2 or siDNAJB6, and aggregation was induced for 5 h. Fixed cells were stained with antibodies against PSMC5. Signal intensities of PSMC5 were quantified in the dualPIM aggregate ROIs and expressed as fold change relative to siCtr. **g**, DualPIM cells were transfected with siCtr, siPSMC5 or siDNAJB6, and aggregation was induced for 5 h. Fixed cells were stained with antibodies against PSMB2. Signal intensities of PSMB2 were quantified as in **g** (*P* = 0.014, siPSMC5 versus siCtr). Images were acquired using the SP8X DLS confocal/light-sheet microscope (**f,g**) and using the IncuCyte S3 system (**d**). Data are mean ± s.d. Error bars represent the s.d. of three independent experiments. *\*P* < 0.05, *\*\*P* < 0.01 and *\*\*\*P* < 0.001. *P* values were calculated by a two-tailed *t*-test with Bonferroni correction (**e,g**) or one-way ANOVA following Dunnett's post hoc test (**a,b**). Scale bars, 20 μm.

Tau aggregates[60,61] by NMS-873 (ref. 60), did not interfere with dual-PIM fragmentation and only minimally reduced lysosomal clearance (Extended Data Fig. 5a,b). As ATPase-containing 19S RPs fragment Tau fibrils in vitro[62], we next individually depleted all the proteasomal subunits in dualPIM cells before measuring aggregate size (Fig. 3a). While

20S CP subunit depletion showed no effect, knockdown of most of 19S RP subunits, including the six ATPases, prevented dualPIM aggregate fragmentation (Fig. 3a,d). To verify whether knockdown of individual subunits indeed inhibit proteasomal degradation, we repeated the screen in Ub-R-GFP cells. Depletion of most 19S RP and 20S CP subunits

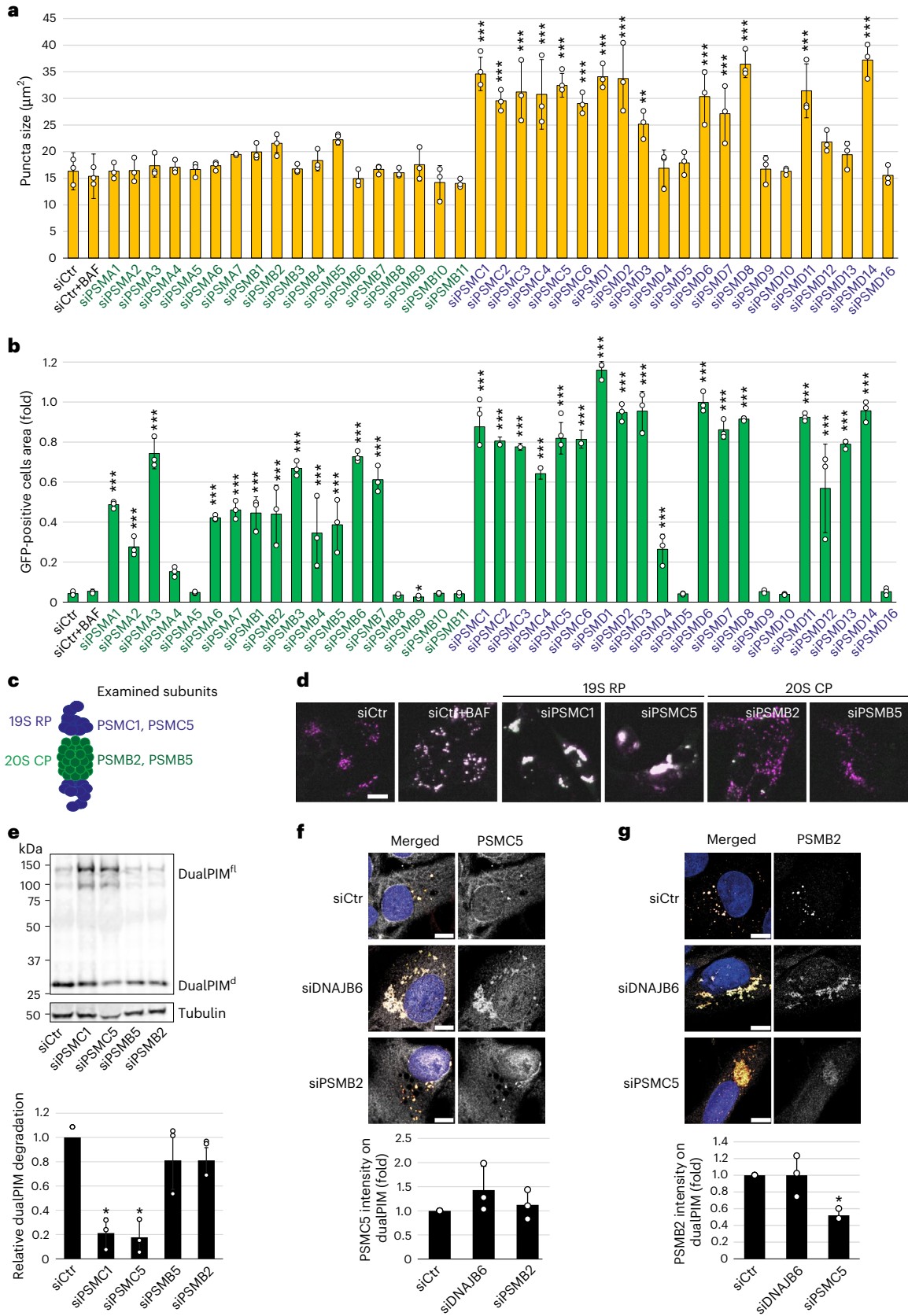

inhibited degradation of UB-R-GFP (Fig. 3b), with 19S RP subunit knockdown somewhat having a stronger impact. These data also revealed that the fragmentation defect of 19S-depleted cells is not caused by proteasomal activity inhibition.

We next selected two subunits of both the 20S CP (PSMB2/β4 and PSMB5/β5) and 19S RP (PSMC2/Rpt1 and PSMC5/Rpt6) for further studies (Fig. 3c,d). Depletion of the selected 19S RP subunits, but not the 20S CP subunits (Extended Data Fig. 5c), blocked dualPIM aggregate degradation (Fig. 3e). The knockdown of these 19S RP and 20S CP subunits increased the amount of ubiquitinated proteins in cell extracts (Extended Data Fig. 5d). Native gel electrophoresis furthermore revealed that while RP abundance, both as free 19S and in the 26S, was strongly reduced upon depletion of its subunits PSMC1 or PSMC5, the amount of 20S increased (Extended Data Fig. 5e). Conversely, knockdown of the 20S subunits PSMB2 or PSMB5 decreased the amount of proteasomal particles (Extended Data Fig. 5e) without abolishing the complete proteasome (26S fractions) or the 19S particles. Of note, when measuring in gel proteasomal proteolytic activity[63], the 20S activity was reduced by PSMB2 or PSMB5 depletion but not upon PSMC1 or PSMC5 knockdown (Extended Data Fig. 5e), whereas the activity of the single-capped and double-capped 26S complexes was reduced by both. These results confirmed that the depletion of the 19S and 20S subunits was specific and effective. Moreover, analysis of lysosome number and activity showed that depletion of the tested proteasomal subunits, or that of DNAJB6, had no effect on this degradative organelle (Extended Data Fig. 5f). Both 19S RP and 20S CP were found on dualPIM puncta in presence and absence of DNAJB6 (Fig. 3f,g), indicating that the proteasome was recruited en bloc in a DNAJB6-independent manner. PSMB2 localization to dualPIM aggregates; however, depended on PSMC5 (Fig. 3g), but that of PSMC5 did not require PSMB2, suggesting that 20S CPs do not localize to the dualPIM aggregates without 19S RPs. When 19S RP and 20S CP localization was assessed simultaneously, we found that numerous dualPIM puncta were positive for both 19S RPs and 20S CPs, but a great proportion was also 19S RP-positive only (Extended Data Fig. 5g), emphasizing the key 19S RP role in lysosomal dualPIM aggregate turnover. Recruitment of DNAJB6 to the dualPIM puncta did not require 19S RPs, as ectopically expressed DNAJB6[H31Q] could still associate with dualPIM puncta in PSMC5-depleted cells (Extended Data Fig. 5h), indicating DNAJB6 and 19S RPs independently associate with these aggregates.

The 19S RPs have three main functions within proteasomes: binding Ub moieties of target proteins, deubiquitinating them to recycle Ub, and unfolding of target proteins by its AAA-ATPase activities[10]. Treating cells with PYR-41, an inhibitor of the Ub-activating enzyme E1 (ref. [64]) indeed blocked proteasomal degradation as expected (Extended Data Fig. 6a,b), but did not impact dualPIM aggregate fragmentation (Extended Data Fig. 6c), implying that ubiquitination as such is not required for fragmentation. DualPIM turnover was nonetheless blocked, probably because SARs like NDP52 and TAX1BP1 were less

efficiently recruited (Extended Data Fig. 6d). Similarly, dualPIM fragmentation was not blocked when the proteasomal Ub-binding receptor PSMD16/Rpn13/ADRM1 was inhibited using RA190[65] (Extended Data Fig. 6e), confirming our siRNA result (Fig. 3a). To determine whether the DUB activity plays a role in fragmentation, cells were incubated with capzimin (CZM) or b-AP15, which inhibit PSMD14/Rpn11 (proteasomal subunit), and USP14 and UCHL5 DUBs (DUBs associated with the proteasome), respectively[66,67]. While these treatments led to a proteasomal impairment (Extended Data Fig. 6a,b,f), dualPIM aggregate fragmentation was unaffected upon b-AP15 treatment, whereas CZM led to a fragmentation inhibition, confirming the knockdown results (Fig. 3a and Extended Data Fig. 6e). Together these data imply that 19S RPs are required for dualPIM fragmentation and suggest that their ATPase subunits play a key role in this.

## Fragmentation is required for the formation of protein aggregate-sequestering autophagosomes

Selective sequestration by autophagy is mediated by the elongation of nascent autophagosomes, known as phagophores, around the cargo[12]. To determine whether fragmentation is important to initiate phagophore formation, we examined dualPIM structures at the ultrastructural level in three-dimensions (3D) by electron tomography in control, SAR405-treated (to inhibit phagophore formation[68]), DNAJB6-depleted or PSMC5-depleted cells. Correlative light-electron tomography (CLET) was used to distinguish between (yellow) cytoplasmic and lysosomal dualPIM aggregates (Extended Data Fig. 7a). Consistent with our immunofluorescence (IF) data, immunogold labelling showed that amorphous dualPIM aggregates coalesced into large inclusions in DNAJB6-depleted or PSMC5-depleted cells in comparison to control and SAR405-treated cells (Fig. 4a and Extended Data Fig. 7b). Phagophores sequestering the cytoplasmic dualPIM aggregates were frequently detected in the control, whereas no phagophores were observed adjacent to the dualPIM inclusions formed in the absence of DNAJB6 or PSMC5, or the presence of SAR405 (Fig. 4a, Extended Data Fig. 7b and Supplementary Videos 3–10). Of note, dualPIM inclusions in DNAJB6- or PSMC5-depleted cells seemed less compact in comparison with the fragmented ones in control and SAR405-treated cells (Fig. 4a, Extended Data Fig. 7b and Supplementary Videos 3–10). This was confirmed by analysing the grey intensity, which is proportional to the protein concentration, of a randomly selected cross-section within the different aggregates and inclusions showing that dualPIM aggregates in control and SAR405-treated cells were ten times more intense compared with DNAJB6- or PSMC5-depleted cells (Fig. 4a,b and Extended Data Fig. 7b,c). Normalizing the grey values to the mean value of each cross-section (Fig. 4c and Extended Data Fig. 7d) shows that the grey value spread is much broader when DNAJB6 or PSMC5 were depleted (Fig. 4c and Extended Data Fig. 7d), further indicating that under these conditions the inclusions were indeed less compact. To corroborate 3D-CLET observations, we quantitated dualPIM

**Fig. 4 | Autophagosomal sequestration of dualPIM aggregates depends on PSMC5- and DNAJB6-mediated fragmentation. a**, DualPIM cells were transfected with siCtr, siDNAJB6 or siPSMC5, and aggregation was induced for 5 h before being fixed and processed for 3D-CLET. Cells transfected with siCtr were either left untreated or treated with 10 μM SAR405 during aggregation induction. Tomograms from tilt series of images from the generated sections (Supplementary Videos 3–6), and representative images from single tomogram slices and their visual models are shown. Additional representative examples are depicted in Extended Data Fig. 7b. Aggregate, green; phagophore, pink; M, mitochondria; N, nucleus; PM, plasma membrane. Scale bars, 200 nm.
**b**, The intensity profile of a random chosen cross-sections within the dualPIM aggregates, highlighted in **a** with yellow lines, were measured and the values are shown. The black lines indicate the mean intensity value. **c**, The intensity values of the graphs from **b** were normalized for each condition against the mean intensity value for each aggregate and the spreads of the differences compared with the

mean are depicted with histograms. **d**, Aggregation was induced in dualPIM cells for 5 h plus or minus 10 μM SAR405 before being fixed and imaged by fluorescence microscopy. Representative images are presented. **e,f**, The average size of the dualPIM puncta (**e**) and the signal intensities of GFP in the dualPIM aggregate (**f**) ROIs in the experiment shown in **d** were quantified. **g**, DualPIM cells were transfected with siCtr, siDNAJB6 or siPSMC5, and aggregation was induced for 5 h, before being fixed and imaged by fluorescence microscopy. Representative images at 5 h are shown. **h,i**, The average size of the dualPIM puncta ($P < 0.001$ (siDNAJB6 versus siCtr); $P < 0.001$ (siPSMC5 versus siCtr)) (**h**) and the signal intensities of GFP in the dualPIM aggregate ($P < 0.001$ (siDNAJB6 versus siCtr); $P < 0.001$ (siPSMC5 versus siCtr)) (**i**) ROIs in the experiment illustrated in **g** were quantified. Images in **d** and **g** were acquired using the SP8X DLS confocal/light-sheet microscope. Data are mean ± s.d. Error bars represent the s.d. of three independent experiments. ***$P < 0.001$. $P$ values were calculated by a two-tailed $t$-test with Bonferroni correction. Scale bars, 20 μm.

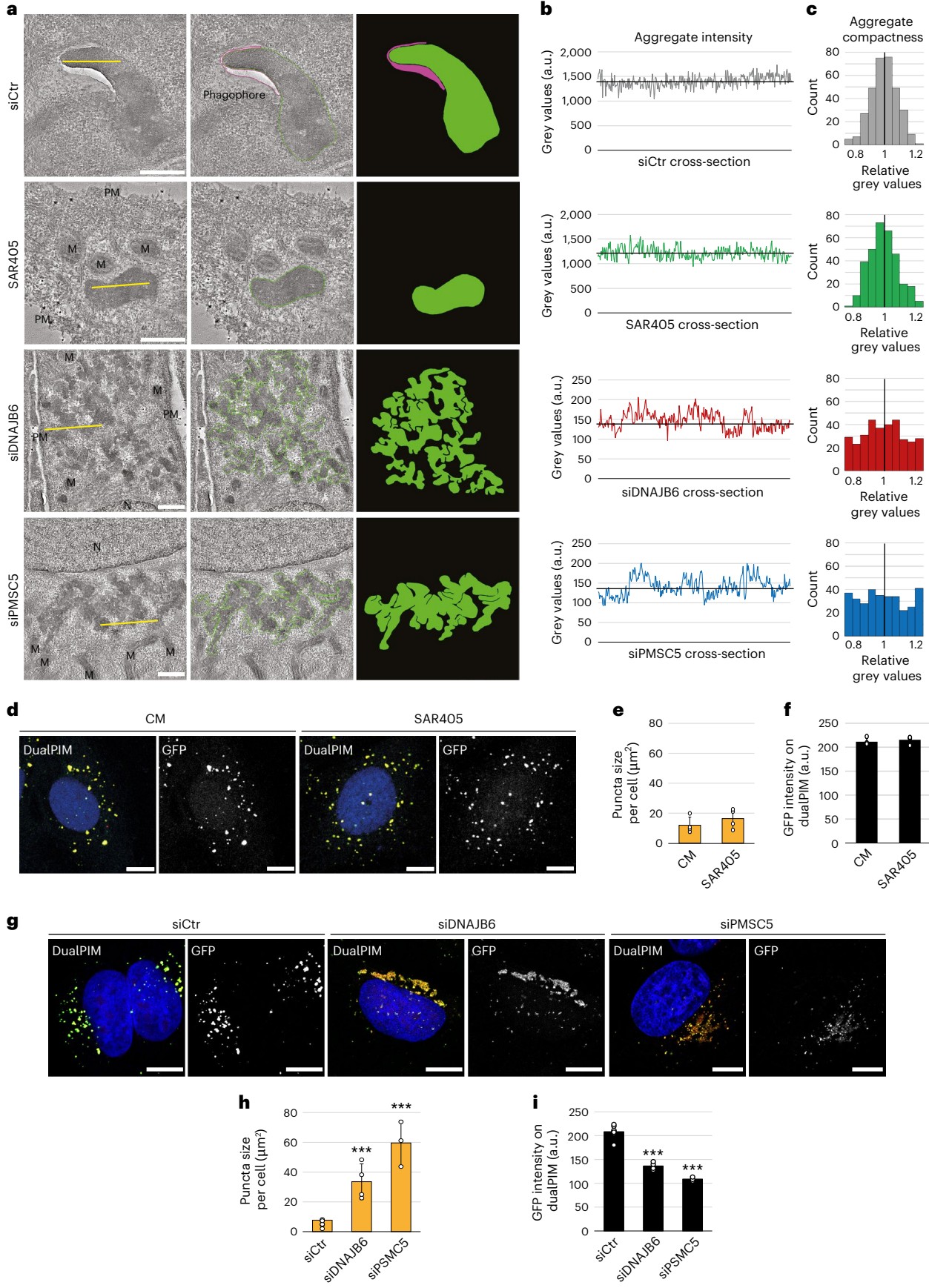

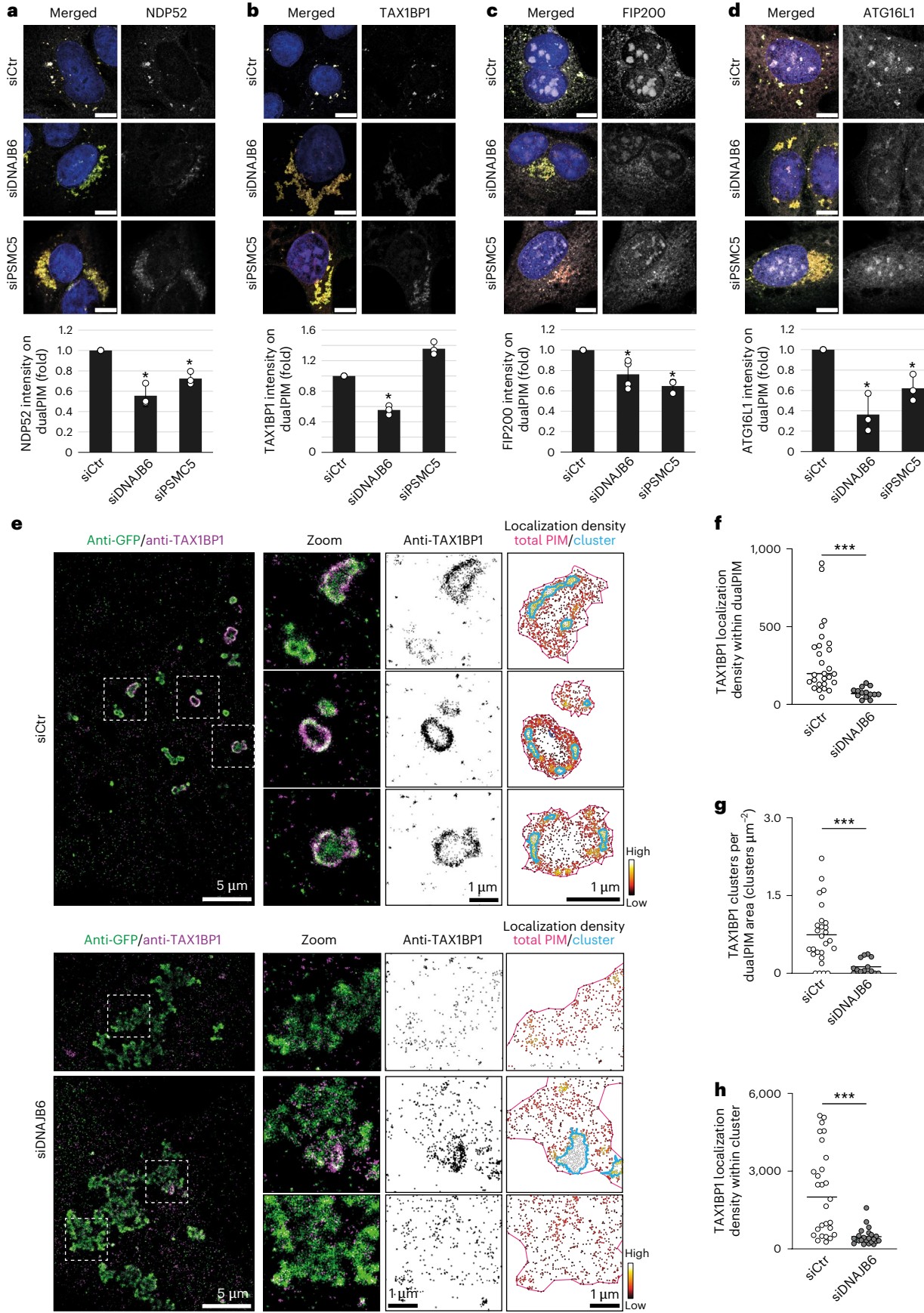

**Fig. 5 | Inhibition of aggregate compaction and fragmentation prevents local clustering of SARs. a–d,** DualPIM cells were transfected with siCtr, siDNAJB6 or siPSMC5, and aggregation was induced for 5 h, fixed and stained with antibodies against NDP52 (P = 0.038 (siDNAJB6 versus siCtr); P = 0.035 (siPSMC5 versus siCtr)) (**a**), TAX1BP1 (P = 0.012 (siDNAJB6 versus siCtr); P = 0.034 (siPSMC5 versus siCtr)) (**b**), FIP200 (P = 0.028 (siDNAJB6 versus siCtr); P < 0.001 (siPSMC5 versus siCtr)) (**c**) or ATG16L1 (P = 0.002 (siDNAJB6 versus siCtr); P = 0.003 (siPSMC5 versus siCtr)) (**d**) before to quantify the IF signal intensities of the indicated proteins in the dualPIM aggregate ROIs and express them as fold change relative to siCtr. Representative images at 5 h are depicted. **e,** Left: DualPIM cells were transfected with siCtr or siDNAJB6, and aggregation was induced for 5 h, before being fixed and stained with antibodies against the indicated proteins and prepared for single-molecule localization imaging. Representative images of SMLM reconstructions of dualPIM aggregates (green) stained for TAX1BP1 (magenta) are shown. Right: localizations of TAX1BP1 at dualPIM aggregates colour coded for local density (gradient). Lines mark the boundary of single aggregates (pink lines) or TAX1BP1 clusters (cyan). **f–h,** TAX1BP1 density on dualPIM aggregates (P < 0.001) (**f**), the number of TAX1BP1 clusters per dualPIM aggregate (P < 0.001) (**g**) and the localization density of TAX1BP1 within clusters (P < 0.001) (**h**) in the experiment shown in **b** were quantified (52 clusters from 33 aggregates within 4 cells for siCtr conditions and 23 clusters from 14 aggregates within 5 cells for siDNAJB6 conditions were quantified). Images were acquired using the SP8X DLS confocal/light-sheet microscope (**a–d**) and using the Nikon Ti microscope (**e**). Data are mean ± s.d. Error bars represent the s.d. of three independent experiments. *P < 0.05 and ***P < 0.001. P values were calculated by a two-tailed t-test with Bonferroni correction (for **a–d**) or a Mann–Whitney non-parametric two-sided test (**f–h**). Scale bars, 20 µm (or as indicated).

puncta GFP intensity and size by IF. While the GFP signal intensity and size of puncta was unchanged between control and SAR405-treated cells (Fig. 4d–f), signal intensity was decreased in the (larger) puncta in DNAJB6- or PSMC5-depleted cells (Fig. 4g–i). The morphological changes in the DNAJB6- or PSMC5-depleted cells, however, did not entail major changes in the biophysical properties of dualPIM inclusions as FRAP analyses showed the same GFP signal recovering profiles as in the control cells (Extended Data Fig. 7e,f). These results imply that both aggregate compactness and fragmentation depend on the action of DNAJB6 and 19S RPs, and this is a prerequisite for induction of autophagosome formation during aggrephagy.

## SAR clustering induced by aggregate fragmentation and compaction triggers phagophore formation

The current view about aggrephagy is that aggregates are modified by Ub moieties that are recognized by SARs, which in turn recruit the ATG machinery to mediate formation of the sequestering autophagosome around the cargo[12]. Consistently, dualPIM puncta in control cells were decorated with NDP52 and TAX1BP1, but also components of the ATG machinery such as FIP200 and ATG16L1 (Fig. 5a–d). Notably, we still observed recruitment of these proteins to dualPIM puncta in the absence of DNAJB6 or PSMC5 (Fig. 5a–d). However, the surface intensities of NDP52, TAX1BP1, FIP200 and ATG16L1, on dualPIM inclusions of DNAJB6 or PSMC5-depleted cells were significantly decreased (Fig. 5a–d). This was not due to decreased protein levels (Extended Data Fig. 8a) and thus indicated a reduced clustering of SARs and ATG proteins on dualPIM inclusions in the absence of DNAJB6 or PSMC5. To corroborate these findings, we employed single-molecule localization microscopy (Fig. 5e) and found that TAX1BP1 localization was less clustered on dualPIM inclusions in DNAJB6-depleted cells (Fig. 5e), as also confirmed by the quantitative analysis of TAX1BP1 distribution density on dualPIM aggregates (Fig. 5f). Moreover, more TAX1BP1 clusters were found on dualPIM aggregates in control cells, and these had a higher TAX1BP1 density (Fig. 5g,h). These results suggest that

SARs and ATG machinery recruitment alone is insufficient to initiate phagophore formation nearby aggregates. Rather, DNAJB6 and 19S RP action is needed to increase aggregate compactness for SAR clustering and phagophore nucleation.

Next, we tested whether the impairment in dualPIM inclusion clearance that accumulated in DNAJB6- or PSMC5-depleted cells could be overcome by nutrient starvation-induced bulk autophagy. Although bulk autophagy progresses normally in DNAJB6- or PSMC5-depleted cells (Extended Data Fig. 8b,c), dualPIM inclusion fragmentation and lysosomal turnover defects in these cells were not bypassed (Extended Data Fig. 8d–f). These data show that DNAJB6 and PSMC5, and by extension the fragmentation machinery, are specifically required for aggrephagy. In line with the notion that the ATG machinery functions downstream of aggregate fragmentation, we also observed that dualPIM puncta did not increase in size upon FIP200 knockdown (Extended Data Fig. 8g). When FIP200 and DNAJB6 or PSMC5 were depleted simultaneously, however, dualPIM inclusions had similar sizes as the ones of DNAJB6- or PSMC5-depleted cells (Extended Data Fig. 8g). Finally, we generated a dualPIM FIP200KO cell line and treated them with 100 nM BTZ for 6 h to accumulate dualPIM inclusions before BTZ washout. While dualPIM inclusion size significantly decreased with concomitant increase in the number of aggregates, they remained mCherry–GFP-positive because they were not delivered into lysosomes (Extended Data Fig. 8h). This shows that the aggrephagic clearance of dualPIM aggregates occurs after or eventually concomitantly with their fragmentation.

## Fragmentation is required to target cellular protein aggregates to lysosomal degradation

To corroborate the findings obtained with the dualPIM system, we next assessed the clearance of puromycin-induced protein aggregates, which mainly consist of defective ribosomal products[69] and are known to be cleared by aggrephagy[70–73]. As expected, puromycin exposure for 2 h led to the accumulation of Ub- and p62-positive puncta, which

**Fig. 6 | The fragmentase is required to clear protein aggregates via aggrephagy. a,** U2OS cells were left untreated (CM) or exposed to 5 µg ml⁻¹ puromycin (puro) for 2 h before being washed and incubated for 3 h just in medium (wash) or in the presence of 100 nM BAF (wash + BAF) or 40 µM VER (wash + VER). Cells were fixed and stained with anti-Ub and p62 antibodies. The number of Ub/p62 double-positive puncta per cell was determined and expressed relative to CM sample (P = 0.009 (puro versus CM); P = 0.001 (wash + BAF versus CM); P = 0.036 (wash + VER versus CM)). **b,** U2OS cells were transfected with siCtr, siDNAJB6, siHSP110 (combination of siHSPH1, siHSPH2, siHSPH3), siFIP200, siTAX1BP1, siBAG3, siPSMB2, siPSMB5, siPSMC1 or siPSMC5, before being treated as in **a**. Representative images are shown. **c,** The number of Ub/p62 double-positive puncta per cell was determined for each condition and the ratio of remaining aggregates was calculated by dividing the number of Ub/p62 double-positive puncta after puro washout with the one before. The quantification of the Ub/p62 double-positive puncta in CM and upon puro

treatment is shown in Extended Data Fig. 9c. P values versus siCtr: P < 0.001 (siDNAJB6); P = 0.002 (siFIP200); P = 0.004 (siTAX1BP1); P < 0.001 (siHSP110); P < 0.001 (siCtr + VER); P < 0.001 (siPSMC1); P < 0.001 (siPSMC5). **d,** Tandem-p62 cells were treated for 5 h with puro and imaged during the washout period. Time-lapse images showing multiple fragmentation (white arrowheads) events. See also Supplementary Video 11. **e,** TEM analysis of U2OS cells left untreated, treated with 5 µg ml⁻¹ puro for 2 h or analysed after 3 h washout in the presence or absence of 10 µM SAR405 or 100 nM BTZ is shown. Arrows and yellow dashed lines highlight aggregates/inclusions; DGC, degradative compartment; E, endosome; ER, endoplasmic reticulum; G, Golgi; LD, lipid droplet. Images in **a** and **b** were acquired using the DeltaVision microscope and in **d** using the IncuCyte S3 system. Data are mean ± s.d. Error bars depict the s.d. of three independent experiments. *P < 0.05, **P < 0.01 and ***P < 0.001. P values were calculated by a two-tailed t-test with Bonferroni correction (**c**) or one-way ANOVA following Dunnett's post hoc test (**a**). Scale bars, 20 µm or 1 µm for **e**.

were rapidly cleared after puromycin washout (Fig. 6a). This clearance was indeed mediated by aggrephagy as BAF treatment blocked their disappearance (Fig. 6a) and a fraction of them colocalized with LAMP1 (Extended Data Fig. 9a). Accordingly, TAX1BP1 or FIP200 knockdown

impaired puromycin-induced aggregate turnover and lysosomal delivery also in the presence of BAF[70] (Fig. 6b and Extended Data Fig. 9a). Transport and clearance of puromycin-induced aggregates in lysosomes was impaired when either DNAJB6 or HSP110 were depleted,

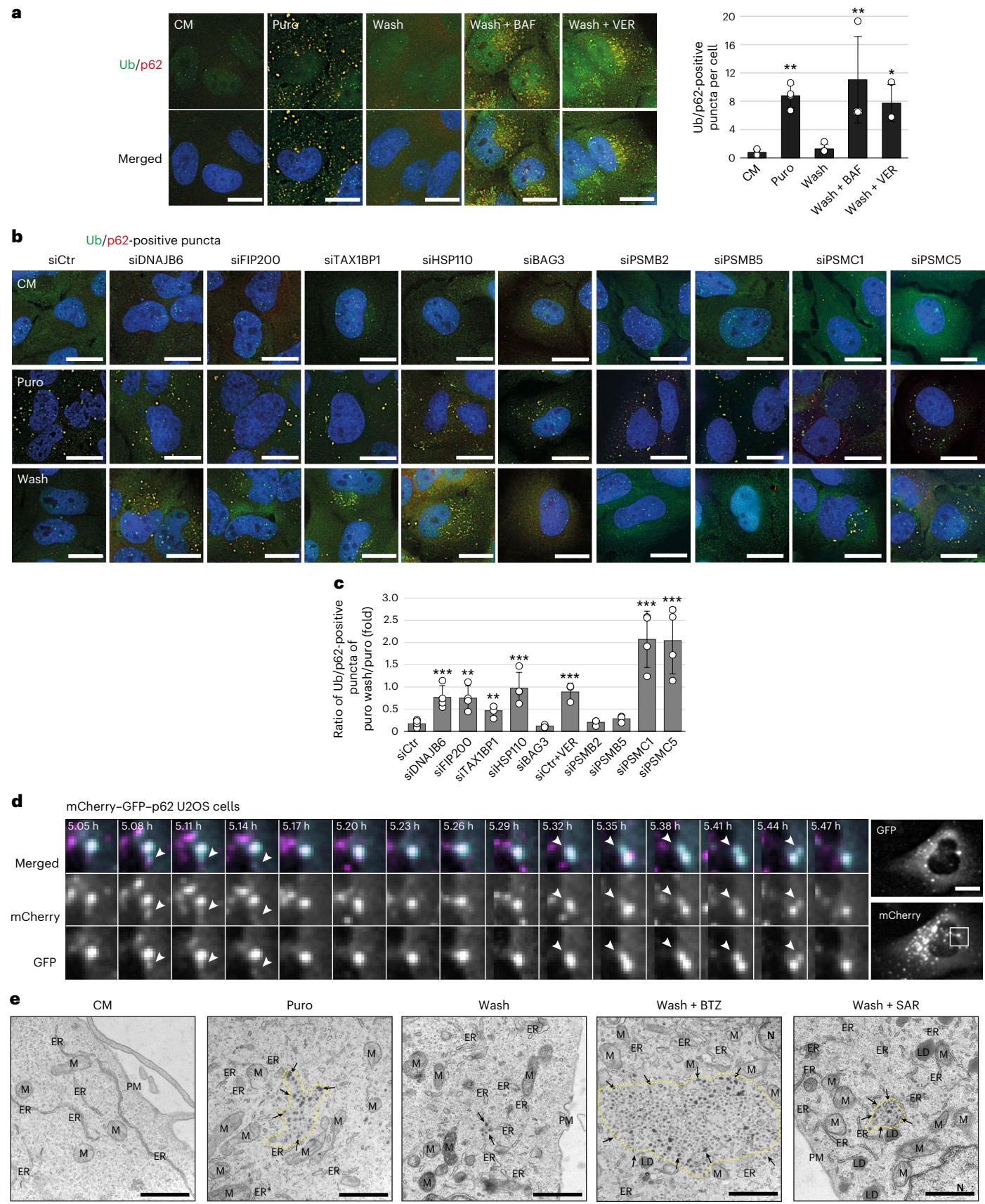

or HSP70 was inhibited with VER, whereas BAG3 knockdown did not (Fig. 6b,c and Extended Data Fig. 9b,c). Furthermore, PSMC1 or PSMC5 depletion also impaired the lysosomal clearance of puromycin-induced aggregates, whereas PSMB2 or PSMB5 knockdown did not (Fig. 6b,c and Extended Data Fig. 9b,c). The depletion of PMSB5, PMSC1 and PMSC5 even increased the number of Ub- and p62-positive puncta already without puromycin (Extended Data Fig. 9c), probably due to lack of functional proteasomes[70]. Similar results were obtained in A549 cells (Extended Data Fig. 10a–c). We also observed puromycin-induced p62 aggregates undergoing fragmentation during washout when single aggregates were imaged over time (Fig. 6d). To validate our IF observations, we examined U2OS cells by transmission electron microscopy (TEM). Puromycin treatment led to the formation of amorphous aggregates composed of clusters of small dark patches surrounded by a lighter area (Fig. 6e), which may correspond to the p62 filaments characterized by cryo-EM[74]. Puromycin washout led to the disappearance of these amorphous aggregates, which was accompanied with the infrequent detection of single dark patch in the cytoplasm (Fig. 6e). These cytoplasmic structures increased when autophagy was blocked with SAR treatment but did not dissolve in cells treated with 100 nM BTZ, they rather accumulated as big cytoplasmic inclusions (Fig. 6e). This indicates that puromycin-induced p62-positive aggregates are also fragmented before lysosomal delivery.

## 19S RPs reduce the accumulation of disease-associated huntingtin aggregates

Partial clearance of disease-associated aggregation-prone proteins, such as mutant huntingtin with an expanded stretch of glutamines (HTT-polyQ), has been attributed to autophagy[9,70,75–77]. Moreover, specific SARs and adaptor proteins such as TAX1BP1, ALFY and CCT2 have been linked to the aggrephagic turnover of HTT-polyQ variants[14,70,77]. It is noteworthy that HTT-polyQ inclusions are very heterogeneous and consist of an amyloid core surrounded by an amorphous rim[78], which makes them structurally different from the dualPIM and puromycin-induced aggregates.

DNAJB6 was already shown to be a crucial factor in counteracting HTT-polyQ aggregation by functioning before amyloids are formed[58,79,80] and its expression levels tightly correlate with pathogenic HTT-polyQ aggregation onset[80]. To study the role of the DNJAB6-HSP70-HSP110 module and 19S RPs in disease-associated inclusion degradation, we generated HEK cells with TTC-inducible expression of a HTT variant with 119 glutamines (HTT-polyQ119). In these cells, we depleted DNAJB6, HSP110, PSMC1, PSMC5, PSMB2 or PSMB5 using siRNA, or blocked autophagy with SAR405, before inducing HTT-polyQ119 expression. The number of HTT-polyQ119-EGFP puncta increase over time (Fig. 7a,b) and the impact of each depletion was quantified by calculating the area under the measurement curves (AUC) and measuring puncta size (Fig. 7b–d). These quantifications confirmed that autophagy inhibition (SAR405 treatment) enhanced HTT-polyQ119 inclusion accumulation and size (Fig. 7a–d). DNAJB6- or HSP110-depleted cells showed 1.4- and 1.8-fold increase in HTT-polyQ119 puncta per cell, respectively (Fig. 7a–c), as expected[37,58]. Knockdown of 19S RP subunits, but not the ones of 20S CP, led to an even larger increase of HTT-polyQ119 puncta and size (Fig. 7a–d). Both DNAJB6 and PSMC5 also localize to HTT-polyQ119 puncta (Extended Data Fig. 10d). DNAJB6-, HSP110-, PSMC1- or PSMC5-depleted cells also showed increased SDS-insoluble HTT-polyQ119 material using a filter trap assay[79] (Fig. 7e). HTT-polyQ119 inclusions were also more prominent in PSMB2 or PSMB5 knockdown cells in comparison with the control. This is consistent with proteasomal inhibition impairing degradation of soluble HTT species, thereby increasing the concentration required for aggregation and also impairing protein homeostasis, which ultimately accelerate disease-related protein aggregation[81–83]. However, even more aggregates accumulated when 19S RP subunits were silenced (Fig. 7f). Indeed, comparing depletion of all proteasomal subunits shows that knockdown of some 20S CP subunits moderately increases HTT-polyQ119 aggregation whereas depletion of the 19S RP subunit overall had more pronounced effects (Extended Data Fig. 10e,f). Moreover, while the depletion of 20S CP subunits did not increase the HTT-polyQ119 puncta size, 19S RP subunit knockdown did, highlighting the importance of the fragmentase activity in delaying HTT-polyQ119 inclusion accumulation.

## Discussion

This study reveals the existence of a cellular fragmentation machinery required for autophagic disposal of aggregates that comprises the HSP70–HSP110–DNAJB6 module and 19S RPs (Fig. 7g). In yeast, the main disaggregase/fragmentase consists of Hsp104 and the Hsp70 machinery. As such, Hsp104 supports the disaggregation of both disordered (heat-induced)[84] and amyloid-type aggregates (reviewed in ref. 21). Expression of Hsp104 and potentiated variants increases the disentangling of several types of protein aggregates in metazoa[85]. Although metazoa lack Hsp104 homologues, HSP70-linked disaggregase activities have been described in vivo[29,86], but initially without any insight in the underlying mechanism or required partners. More recently, however, as shown in vitro metazoan HSP70 proteins together with HSP110 and selected JDPs such as DNAJA2 and one DNAJB1 can effectively disassemble protein aggregates (for example refs. 22, 31–34,37,87). For example, preformed aggregates of heat-unfolded firefly luciferase can be disassembled into functional proteins by an HSP70–HSP110 module that requires a complex composed of one DNAJA2 and one DNAJB1 dimer[22,36]. Likewise, preformed amyloids[30–33,37] can also be disassembled in vitro by the action of an HSP70–HSP110–DNAJB1 module. The latter, however, results in fragments that can act as new seeds[33,35], accelerating amyloid/fibril formation, and potentially enhancing disease pathogenesis in vivo through prion-like spreading[30].

**Fig. 7 | The cellular fragmentation machineries reduce the accumulation of disease-associated HTT-polyQ119 aggregates. a**, HEK-HTT-polyQ119-EGFP cells were transfected with the indicated siRNAs for 48 h before inducing the expression of HTT-polyQ119-EGFP using TTC for 24 h and simultaneously monitoring aggregate formation/dissipation by live-cell microscopy using the IncuCyte S3 system. Then, 10 μM SAR405 was added to siCtr-treated cells during the TTC treatment. Representative images at 24 h post-TTC treatment are shown. **b–d**, The number of HTT-polyQ119-EGFP puncta per image was measured every 2 h using the IncuCyte S3 system (**b**) and the aggregate levels were quantified as AUC to monitor aggregate levels over time (**c**) (*P* values versus siCtr: *P* < 0.001 (siCtr + SAR); *P* = 0.014 (siDNAJB6); *P* = 0.006 (siHSP110); *P* = 0.002 (siPSMC5)), and the puncta sizes were determined (**d**) (*P* values versus siCtr: *P* = 0.015 (siCtr + SAR); *P* = 0.001 (siPSMC1); *P* < 0.001 (siPSMC5)). **e,f**, HEK-HTT-polyQ119-EGFP cells were transfected with the indicated siRNAs for 48 h before inducing HTT-polyQ119 expression with TTC for 24 h and examining HTT-polyQ119 inclusions amounts using the filter trap assay. *P* values versus siCtr: *P* < 0.001 (siPSMC1); *P* < 0.001 (siPSMC5); *P* = 0.06 (siDNAJB6); *P* = 0.001 (siHSP110). A representative filter trap image is presented (**e**) as well as the quantification of the biological repeats (**f**). **g**, Working model for the role of DNAJB6-HSP70-HSP110 module and the 19S RP in aggrephagy. Together these two fragmentation machineries mediate compaction and fragmentation of large aggregates or inclusions, which lead to the clustering of SARs. Increase local binding avidity of the SARs trigger the assembly of the ATG machinery. This coordinated action promotes phagophore formation around the fragmented aggregates and their subsequent delivery into lysosome for degradation. In contrast, in the absence of either the DNAJB6-HSP70-HSP110 module or the 19S RP, aggregates are not processed and remain cytosolic as inclusions. Images in **a** were acquired using the IncuCyte S3 system. Data are mean ± s.d. Error bars represent the s.d. of four independent experiments. *P < 0.05, **P < 0.01 and ***P < 0.001. *P* values were calculated by a two-tailed *t*-test with Bonferroni correction (**c,f**) or one-way ANOVA following Dunnett's post hoc test (**d**). Scale bar, 20 μm.

The fragmentase activity of the HSP70–HSP110–DNAJB6 module that we describe differs from the one of HSP70–HSP110–DNAJB1 in several ways, including the dependence on 19S RPs. Moreover, the fragmentation action of the HSP70–HSP110–DNAJB6 module is coordinated with subsequent autophagic degradation and therefore seems not to be harmful, but rather protective. The apparent difference in JDP requirements can be multi-faceted. It is possible that the amorphous

aggregates that we investigated cannot be recognized by previously studied JDPs such as DNAJB1, DNAJA1 or DNAJA2. Although their combined depletion did not impact the fragmentase activity, a possibility is that they act redundantly on amorphous aggregates and therefore single deletions do not lead to a phenotype. One major difference could be that unlike in vitro where aggregates are preformed and homotypic, aggregate generated in living cells interact with other proteins forming

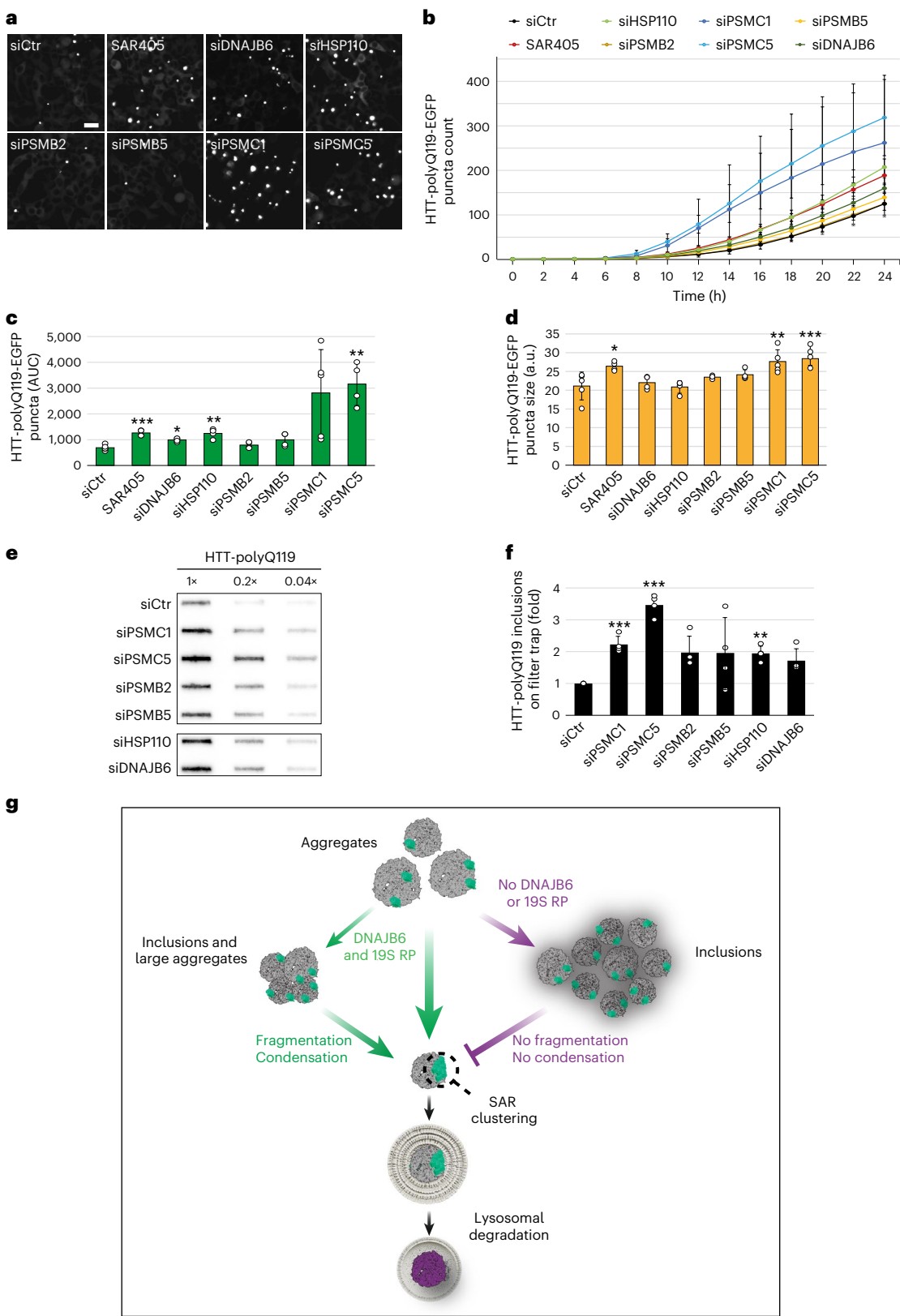

heterotypic aggregates. Additionally, the presence of chaperones, particularly DNAJB6, during this process may not only affect aggregate arrangements into ordered fibrils[22,58,79,80], but also block such heterotypic interactions affecting their further disposal. This combinatorial effect may explain why DNAJB6 decreases toxicity and increases life span in polyQ-disease-related animal models[58,88].

To our surprise, fragmentation of the aggregates studied here also requires 19S RPs, which are best known for their function as part of the 26S proteasome, where they recognize, deubiquitinate and unfold proteins channelled into 20S CPs for degradation[89]. For this function, 19S RPs and 20S CPs assemble in single-capped or double-capped 26S proteasomes[90,91]. The 19 RP fragmentase activity appears to not rely on 20S CPs, inferring that their ATPases are required for fragmentation. As 19S RP depletions had overall somewhat stronger effects on the proteasomal degradation activity, however, it cannot be excluded that 20S CPs may also play a partial role in dualPIM aggregate fragmentation. 19S RPs having a 20S CP-independent function is not unprecedented. Both 19S RPs and 20S CPs have autonomous roles in cellular and synaptic communication, as chaperones and degradation units, respectively[92–95]. In vitro experiments indicated that 26S proteasomes are able to fragment Tau fibrils into smaller pieces and that this depends on ATP, suggesting that 19S ATPases may be key for this function[62]. This notion is indirectly supported by the observation that α-synuclein aggregates entering cells are surrounded by proteasomes that is associated with their size reduction[96]. As we still observed 20S CP recruitment to aggregates, albeit in a 19S-dependent manner, it seems that at least in some cases entire 26S proteasomes associate to aggregates for 19S RP-mediated fragmentation. The precise recruitment mechanism is, however, still unresolved.

This combined action of the DNAJB6–HSP70–HSP110 module and 19S RPs on protein aggregates is reminiscent of the synergy between the yeast Hsp70 system and Hsp104 (refs. 27–29) or the bacterial DNAK-J-E system and Clp-B[97,98]. Metazoan HSP70 chaperones seem to be able to act in concert with AAA-ATPase as well, including VCP, to deal with protein aggregates[60,61] or Torsins, for nuclear pore quality control[99,100]. Thus, we hypothesize that metazoa can use combinations of JDP proteins and AAA-ATPases to specify the functionality of the otherwise promiscuous HSP70 proteins.

Both the DNAJB6–HSP70–HSP110 module and 19S RPs are also essential for the fragmentation and autophagic turnover of puromycin-induced aggregates and for prevention of accumulation of HTT-polyQ119 amyloids. This result might explain why proteasomes are recruited to HTT-polyQ aggregates, albeit not for direct degradation[81,101]. Based on previous results[58,102], we speculate that the fragmentation machinery targets the HTT-polyQ entities that are not amyloids yet. This is supported by findings showing that HTT-polyQ amyloids are not an aggrephagy substrate[78]. HTT-polyQ proteins form condensates before becoming amyloids[103], which start to assemble into fibrils from the inner core, with the rims remaining fluid or gel-like[78]. These 'gel-like' amyloid precursors are preferentially targeted by autophagy[78].

In the current model of aggrephagy, protein aggregates are ubiquitinylated and then recognized by SARs, which in turn recruit the ATG machinery to initiate the sequestration by a phagophore[12]. Our data show that SARs and at least parts of the ATG machinery localize to the aggregates in the absence of the fragmentation machinery, but this is insufficient to trigger local autophagosome biogenesis. Our working hypothesis is that fragmentase components and SARs are independently recruited to aggregates. Compaction by the fragmentase, however, will cluster the already recruited SARs to increase binding avidity and concentration of the ATG machinery, which triggers phagophore formation (Fig. 7g). In contrast, aggregates coalescence into bigger inclusions and autophagic degradation is inhibited when the fragmentation is blocked (Fig. 7g). Similar piecemeal digestions have been observed in other selective types of autophagy, including mitophagy, ER-phagy, lipophagy, nucleophagy and pexophagy[104–109]. Additionally,

SAR clustering seems to drive organelle degradation in ER-phagy and a selective autophagy system relying on artificial cargo[110–112].

Overall, we suggest that aggregate fragmentation is per se not harmful, but necessary for their successful degradation by aggrephagy. If the aggrephagic clearance is disturbed, however, the fragmented material may become cytotoxic, especially if it has seeding propensity. Notably, as autophagy induction alone is insufficient to enhance the disposal of inclusions, understanding the functional interplay between the fragmentation and ATG machineries is critical. Aggrephagy stimulation is considered a potential therapeutic approach removal of disease-associated aggregates[113,114]. Therefore, this knowledge may also suggest how to optimally modulate aggrephagy to benefit human health.

## Online content

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

## Methods

### Antibodies and reagents

The following primary antibodies (and dilutions) were used: rabbit anti-BAG3 (Abcam, ab47124, 1:1,000); mouse anti-GAPDH (Fitzgerald Industries International, 10R-G109a, 1:1,0000); rabbit anti-RB1CC1/ FIP200 (Proteintech, 17250-1-AP, 1:1,000 WB, 1:50 IF); rabbit anti vinculin (Cell Signaling, 13901, 1:1,000); rabbit anti-TAX1BP1 (Millipore, HPA024432, 1:1,000 WB, 1:100 IF); rabbit anti-CALCOCO2/NDP52 (Millipore, HPA023195, 1:1,000 WB, 1:100 IF); rat anti-HSC70 (Enzo Life Sciences, ADI-SPA-815, 1:1,000 WB); mouse anti-HSP70 (Enzo Life Sciences, ADI-SPA-810, 1:1,000 WB, 1:100 IF); mouse anti-APG2 (Santa Cruz Biotechnology, sc-365366, 1:1,000 WB); mouse anti HSPBP1 (Novus Biologicals, NBP 2-01168, 1:1,000 WB); mouse anti-HSPB1 (Enzo Life Sciences, ADI-SPA-800, 1:1,000 WB); mouse anti-HSPB7 (Abnova H00027129-M01, 1:1,000 WB); rabbit anti-DNAJA1 (Abcam, ab126774, 1:1,000 WB); mouse anti DNAJA2 (Sigma-Aldrich, WH0010294M1, 1:1,000 WB); rabbit DNAJB1 (Atlas Antibodies, HPA063247, 1:1,000 WB); rabbit anti-DNAJB2 (Atlas Antibodies, HPA036268, 1:1,000 WB); rabbit anti-DNAJB6 (a kind gift from Ineke Braakman, Utrecht University, 1:1,000 WB, 1:50 IF); rabbit anti-ATG16L (MBL International, PM040, 1:1,000 WB, 1:50 IF); rabbit anti-LC3 (Novus Biologicals, NB600-1384, 1:1,000 WB); mouse anti-tubulin (Sigma-Aldrich, T5168, 1:10,000 WB); mouse anti-actin (Merck, MAB1501, 1:10,000 WB); guinea pig anti-p62/SQSTM1 (Progen, GP62-C, 1:200 IF); mouse anti-p62/SQSTM1 (Abcam, ab56416, 1:2,000 WB); mouse anti-LAMP1 (BP Biosciences, 555798, 1:100 IF); mouse anti-RFP (ChromoTek, 6g6, 1:2,000 WB); mouse anti-V5 (Thermo Fisher Scientific, R960-25, 1:2,000 WB, 1:200 IF); rabbit anti-PSMC1 (Merck, HPA000872, 1:1,000 WB, 1:50 IF); rabbit anti-PSMC2 (Cell Signaling, 14395S, 1:1,000 WB); rabbit anti-PSMC3 (Cell Signaling, 13923S, 1:1,000 WB); rabbit anti-PSMC4 (Proteintech, 11389-1-AP); rabbit anti-PSMC5 (Merck, HPA064293, 1:1,000 WB, 1:50 IF); rabbit anti-PSMC6 (Proteintech, 15839-1-AP, 1:1,000 WB); rabbit anti-PSMB5 (Merck, HPA049518, 1:1,000 WB, 1:50 IF); rabbit anti-PSMB2 (Merck, HPA026324, 1:1,000 WB, 1:50 IF); mouse anti-PSMA7 (Enzo Life Sciences, BML-PW8110, 1:50 IF); mouse anti-PSMC2 (Enzo Life Sciences, BML-PW8825, 1:3,000 WB). The following secondary antibodies were used for the visualization of the primary antibodies: AlexaFluor488-conjugated goat anti-mouse (Invitrogen, Invitrogen, A-11001, 1:250 IF); AlexaFluor568-conjugated goat anti-mouse (Invitrogen, A-11031, 1:250 IF); AlexaFluor568-conjugated goat anti-rabbit (Invitrogen, A-11011, 1:250 IF); goat anti-guinea pig (Invitrogen, A-11075, 1:250 IF), AlexaFluor647-conjugated goat anti-mouse (Invitrogen, A-21235, 1:250 IF); AlexaFluor680-conjugated goat anti-mouse (Invitrogen, A-21058, 1:5,000 WB, 1:250 IF); AlexaFluor680-conjugated goat anti-rabbit (Invitrogen, A-21109, 1:5,000 WB, 1:250 IF); IRDYE 800-conjugated goat anti-mouse (Rockland, 610-132-121, 1:5,000 WB); AlexaFluor647-conjugated donkey anti-rat (Jackson Immuno-Research, 712-605-153, 1:250 IF); goat anti-chicken AF647 (Invitrogen, A-21449, 1:250 IF); and horse radish peroxidase (HRP)-conjugated sheep anti-mouse (GE Healthcare, NXA931, 1:5,000 WB). EBSS (Earle's balanced salt solution) (24010-043), LysoTracker red (L7528) and puromycin (A1113803) were from Invitrogen/Thermo Fisher Scientific. Hoechst33342 (B2261), RPN13 inhibitor RA190 (5303410001), capzimin dimer (SML1995), nocodazole (M1404) and cytochalasin D (C8273) were from Sigma-Aldrich, Bafilomycin A₁ was from BioAustralis (BIA-B1012). B/B Homodimerizer (rapalog2) was from Takara Bio (635059). BTZ (S1013), SAR405 (S7682), VER-155008 (S7751) and b-AP15 (S4920) were from Selleck Chemicals and MG132 (474790) was from Sigma-Aldrich. JG-98 was a kind gift from J. Gestwicki.

### Plasmids

The pcDNA5–FRT-TO–mCherry–GFP–PIM plasmid was generated by subcloning the mCherry–GFP–PIM fragment from pBA–mCherry–EGFP–PIM[38] into an empty pcDNA5–FRT-TO (Invitrogen) using the BamHI and NotI restriction enzymes. The pcDNA5–FRT-TO–mCherry–APEX2–PIM plasmid was generated by first inserting the APEX2 sequence into an mCherryPIM construct using the AscI restriction enzyme. Subsequently, the mCherry–APRX2–PIM construct were amplified by PCR and subcloned into an empty pcDNA5–FRT-TO plasmid using the BamHI and NotI restriction enzymes. The pcDNA5–FRT-TO–mCherry–EGFP–p62 plasmid was generated as follows: an empty pcDNA5–FRT-TO plasmid was digested using restriction enzymes KpnI and BamHI. PCR-amplified DNA fragments corresponding to mCherry, EGFP and p62 were inserted in the cut pcDNA5–FRT-TO plasmid backbone by Gibson assembly (NEBuilder Assembly Tool v.1.12.15 was used to generate annealing primers). To have an optimal Kozak sequence, the nucleotides preceding the first Met of mCherry were ccagcc. Two *SGSGSGSG* codon-optimized linker sequences (agcggttcaggatcaggttcagga and tctgggtccggcagtgggagcggt) were added between mCherry and EGFP, and EGFP and p62, respectively. The remaining plasmids have been described elsewhere: pcDNA5–FRT-TO–V5–DNAJB6-wt[79], pcDNA5–FRT-TO–V5–DNAJB6–H31Q mutant[79] and pcDNA5–FRT-TO–V5 (ref. [79]), pEGFP–N1 Ub-R–GFP plasmid (a kind gift from N. Dantuma, Addgene plasmid #11939)[59].

### Cell lines and cell culture

U2OS (a kind gift from G. Strous), A549 (a kind gift from A. Huckriede), HeLa cells (a kind gift from J. Klumperman), Flp-In U2OS (Invitrogen, K6500-01), FLIP-IN dualPIM U2OS (dualPIM), FLIP-IN mCherryPIM U2OS (mCherryPIM), Ub-R–GFP HEK293 (Ub-R–GFP), FIP200KO dualPIM, mCherry–GFP–p62 U2OS (tandem-p62) and stable TTC-inducible HTT-polyQ119-EGFP-expressing HEK293T (HEK-HTT-polyQ119-EGFP)[115] cells were cultured in Dulbecco's modified Eagle medium (DMEM; Gibco, 31966-021) supplemented with 100 U ml⁻¹ penicillin, 100 µg ml⁻¹ streptomycin (Gibco, 15140-122) and 10% fetal calf serum (FCS) at 37 °C in a 5% $CO_2$ humidified atmosphere. The culture medium of the dualPIM U2OS and HEK-HTT-polyQ119-EGFP cells were additionally supplemented with 15 µg ml⁻¹ blasticidin (Invivogen, ant-bl-1) and 100 µg ml⁻¹ hygromycin B (Thermo Fisher Scientific, 10687010).

The FLP-IN mCherry–EGFP–PIM U2OS cell line (dualPIM cells) and the FLP-IN mCherry–APEX2–PIM U2OS cell line (mCherryPIM cells) were generated using the Flp-In T-REx Core kit (Invitrogen, K6500-01) according to the manufacturer's protocol. In brief, Flp-In T-Rex U2OS cells were co-transfected with the pOG44 vector and the pcDNA5–FRT-TO–mCherry–GFP–PIM plasmid (for dualPIM) or the pcDNA5–FRT-TO–mCherry–APEX2–PIM (for mCherryPIM) in a 10:1 ratio before adding 15 µg ml⁻¹ blasticidin and 100 µg ml⁻¹ hygromycin B to the culture medium at 24 h post-transfection for single colony selection. For the mCherryPIM cells, single clones were further selected based on their mCherry expression using fluorescence-activated cell sorting (FACS) (Bigfoot cell sorter) with a 100-µm nozzle. PIM construct expression was induced by adding 1 mg ml⁻¹ TTC (Sigma-Aldrich, 87128) for 24 h and validated by western blot analysis. PIM aggregation was triggered by treating TTC-induced dualPIM cells with 500 nM rapalog2 (ref. [38]). Similarly, mCherry–GFP–p62 cells (tandem-p62) were generated by co-transfecting pcDNA5–FRT-TO–mCherry–EGFP–p62 and the pOG44 recombinase plasmid (Invitrogen) in a 1:9 ratio into the U2OS Flp-In T-REx cells (kindly provided by S. Blacklow), following the manufacturer's procedure. After selection with 5 µg ml⁻¹ blasticidin and 100 µg ml⁻¹ hygromycin B for 2 weeks, a pool population of mCherry–EGFP–p62-expressing U2OS cells was collected.

The Ub-R-GFP HEK293 cell line was generated by transfecting Flp-In T-REx HEK293 cells (Invitrogen, R78007) with the pEGFP–N1 Ub-R–GFP plasmid using polyethylenimine (PEI)[116]. At 48 h post-transfection, cells were selected with 1 mg ml⁻¹ G418 (Gibco, 11811031). The selected and expanded cell population was sorted by FACS (Sony SH800S) with a 130-µm nozzle and a 488 nm laser and was single-cell seeded and expanded. Expanded clones were treated with 5 µM MG132 for 6 h and Ub-R–GFP expression was validated by western blot analysis.

The FIP200 knockout dualPIM cell line (dualPIM FIP200KO) was generated using CRISPR/Cas9 gene editing. Sets of two single guide RNAs (CACCAGGTGCTGGTGGTCAA and GAGTGTGTACCTACAGTGCT) were designed using the online tool http://crispr.tefor.net/. Guides were cloned into the pX458 plasmid (Addgene, #48138), which allows the simultaneous expression of Cas9 and GFP. DualPIM cells were seeded in six-well plates at a density of 150,000 per well, and transiently transfected using the XfectTM transfection kit (Takara) as per the manufacture's protocol. Cells were trypsinized 48 h post-transfection, and suspended in the FACS buffer (PBS, 1% FBS, 1% pen−strep, 5 mM EDTA and 25 mM HEPES). Single cells were sorted for GFP expression using a SH800S cell sorter (Sony Biotechnology). Single-cell clones were then expanded before examining the modification of the *FIP200* gene by DNA sequencing and the lack of FIP200 expression by immunoblot.

### DualPIM aggregate induction
DualPIM cells were either transfected with the indicated siRNAs for 48 h or were seeded without transfection. At 24 h before the analyses the cells were treated with 1 mg ml[−1] TTC to induce the dualPIM expression. To induce dualPIM aggregate formation, cells were treated with 500 nM rapalog2 for either the whole experiment (Incucyte S3 analysis) or until cells were fixed or lysed. While dualPIM aggregates are formed and then turned over by aggrephagy under control conditions, a fragmentation block leads to their coalescence in much larger aggregates, which we call dualPIM inclusions.

### Western blot analyses
Cells grown in six-well or 24-well plates were washed with PBS and collected in 100 µl of lysis buffer (Tris-buffered saline, pH 7.6, 1% Triton X-100 and Complete protease inhibitor (Roche, 11836170001)). Lysates were then incubated on ice for 30 min, vortexed and centrifuged at 14,000$g$ for 10 min at 4 °C. The supernatants were collected and mixed with the Laemmli loading buffer[117]. Alternatively, cells were directly lysed in Laemmli loading buffer and sonicated for 1 min. Equal protein amounts were separated by SDS−PAGE and after standard western blot, proteins were detected using specific antibodies and the Odyssey Imaging System (LI-COR Biosciences). Densitometric values were determined and quantified on western blots at non-saturating exposures using the ImageJ software[118], and normalized against signal intensities of the tubulin or the actin loading control. For the filter trap assay and the 6xHis-Ub pulldown the following western blot detection protocol was used. Then, 30 µg of each sample were loaded onto 10% SDS−PAGE gels (TGX Stain-Free FastCast system, Bio-Rad), before proteins were transferred to nitrocellulose membranes (Schleicher and Schuell, PerkinElmer), followed by primary antibody incubation and specific horseradish peroxidase-conjugated secondary antibodies. Chemiluminescent reactions were performed with the Pierce ECL Western Blotting Substrate kit (Thermo Fisher Scientific) and detected using the ChemiDoc Touch Imaging System (Bio-Rad). Bands were quantified with the Image Lab software v.6.0 (Bio-Rad) and protein levels were expressed as relative to GAPDH. DualPIM degradation was determined by dividing the lysosomal dualPIM degradation product (dualPIM$^d$) by the amount of full-length dualPIM (dualPIM$^{fl}$).

### Filter trap assay
Detection of HTT-polyQ119 aggregates by filter trap assay was performed as described previously[119], with a modified lysis protocol that substantially improved assay reproducibility. HEK-HTT-polyQ119-EGFP cells were grown in six-well plates and transfected with 20 nM siRNAs targeting the proteins indicated in the figure legends for 48 h. Expression of the HTT-polyQ119 fusions was induced using 1 mg ml[−1] TTC for 24 h. Cells were washed twice in ice-cold PBS and scraped in 200 µl lysis buffer (100 mM NaCl, 50 mM Tris-HCl, pH 7.4, 1 mM $MgCl_2$, 0.5% SDS, 0.15 U µl[−1] benzonase nuclease (Merck), EDTA-free complete protease inhibitor cocktail (Merck)). Samples were vortexed for 20 s and

incubated on ice for 30 min (with 5 s vortexing every 10 min). Subsequently, the final SDS concentration was adjusted to 2% using a solution of 20% SDS, before protein concentration was measured with the DC protein assay (Bio-Rad) and samples stored at −80 °C until further use. Serial dilutions (1.2, 0.24 and 0.048 µg µl[−1]) of each sample were prepared in FTA buffer (150 mM NaCl, 10 mM Tris-HCl, pH 8.0, 2% SDS and 50 mM dithiothreitol), boiled for 5 min and stored at −20 °C. Finally, 100 µl of each sample were used for the filter trap assay[119].

### Native gel analysis
Cell pellets were suspended in TSDG buffer (10 mM Tris-HCl pH 7.4, 25 mM KCl, 10 mM NaCl, 1.1 mM $MgCl_2$, 0.1 mM EDTA, 10% glycerol and 1 mM ATP freshly added) and lysed by five freeze−thaw cycles in liquid nitrogen. Lysates were then clarified by centrifugation at 14,000$g$ for 15 min at 4 °C and protein concentration in the supernatants was determined by Bradford protein assay (Serva). After the addition of 5× native sample buffer (20 mM Tris pH 8.0, 50% glycerol and bromophenol blue) samples were separated for 2 h at 150 V on a 3–12% NativePAGE Novex Bis-Tris gels (Invitrogen). Gels were imaged in a Typhoon Biomolecular Imager (GE Healthsciences) and then transferred onto PVDF membranes (Millipore) using the TransBlot Turbo transfer system (Bio-Rad). After blocking in 5% milk, The PVDF membranes were first incubated with an anti-PSMA6 a (kindly provided by S. Murata, University of Tokyo) or an anti-PSMC2 antibody and finally visualized and analysed with the Odyssey Imaging System.

### Immunofluorescence and fluorescent microscopy
Cells were fixed with 3.7% paraformaldehyde (PFA; Sigma-Aldrich, 441244), 3.7% formaldehyde (Merck, 1040031000) or 100% methanol, washed and blocked with blocking buffer (PBS, 1% bovine serum albumin and 0.1% saponin). Primary and secondary antibodies were diluted in the blocking buffer and incubated for 1 h and cells were mounted in ProLong Gold (Thermo Fisher Scientific, P-36931) with 4,6-diamidino-2-phenylindole (DAPI) to stain the nuclei. Fluorescent microscopy was performed at room temperature (RT) using the DeltaVision RT fluorescence microscope (Applied Precision) equipped with a CoolSNAP HQ camera (Photometrix). Images were generated by collecting a stack of 6–16 images with focal planes 0.20 µm apart using a PLAPON ×60 oil, 1.42, WD 0.15 mm (Olympus) objective, and subsequently deconvolved using the SoftWoRx software (Applied Precision). Alternatively, confocal images were acquired using a SP8X DLS confocal/light-sheet microscope (Leica microsystems) with a HC PL APO CS2 ×63/1.4 oil objective. Confocal images were generated by collecting a stack of four images with focal planes 0.20 µm apart. Quantification of puncta number and size, and colocalization events of the acquired images (at least 20 cells per condition per independent experiment) was performed using Icy software (http://icy.bioimageanalysis.org) using either the spot detector plugin or the ImageJ software. For all analyses, representative single plan images are presented. The sensitivity and segmentation of the Icy spot detection analysis module was adjusted in each experiment. ImageJ software was used to determine the signal intensity of each examined protein in the region of interest (ROI), the imaged dualPIM aggregate. In brief, the mCherry channel was used to define the ROIs before examining the distribution and quantifying the signal intensity of the analysed proteins within the ROIs. For live-cell imaging, HeLa cells were incubated for 1 h with 500 nM rapalog2. Cells were mounted in a metal imaging ring (Invitrogen, A7816) in DMEM containing 10% FCS and penicillin−streptomycin without phenol red (Lonza, BE12-917F). Imaging started 6−7 h after rapalog2 addition. Cells were maintained at 37 °C with 5% $CO_2$ and imaged every 5−10 min for 10−15 h. Imaging was performed with a Nikon Eclipse TE2000E equipped with ×40 oil immersion objective (Plan Fluor, NA 1.3 Nikon), an incubation chamber (Tokai Hit; INUG2-ZILCS0H2) and motorized stage. Illumination was performed using LED illumination (pE-4000; CoolLED) using 470 nm and 550 nm excitation and ET−GFP

(49002, Chroma) and ET–mCherry (49008, Croma) filters. Images were processed using ImageJ software.

## Fluorescence recovery after photo-bleaching

DualPIM cells were transfected with either control, PSMC5 or DNAJB6 siRNA for 48 h, TTC-treated for 24 h and incubated with 500 nM rapalog2 for 5 h. Cells were imaged using a Zeiss LSM 780 laser-scanning confocal microscope coupled to a XL S1 Dark incubation chamber (Carl Zeiss Microscopy), with a ×63/1.3 NA glycerine immersion objective. For the FRAP analyses, circular ROIs of the same area were bleached with a 488 nm laser and a He-Ne laser (594 nm) at maximum power, and recovery/loss of fluorescence intensity was recorded within the ROIs. In total, 80 images per inclusion were acquired every 10 s, with two images being taken before the bleaching. The images were then analysed using ImageJ and the fluorescence recovery profiles (at least 15 per condition) of the mCherry signal were normalized to the average pre-bleaching values.

## siRNA and DNA transfections

U2OS cells were transfected for 48 h with 20 nM siRNA using 0.1 µl or 0.5 µl of Lipofectamine RNAiMAX (Thermo Fisher scientific, 13778150) for 96- or 24-wells plate cultures, respectively, according to the manufacturer's protocol. Supplementary Table 1 provides the list of all used siRNAs. Of note, the siRNA against DNAJB6 is targeting its 5'UTR region and consequently it does not affect the expression of DNAJB6 variants from plasmids. For DNA transfection, plasmids were transfected using either the Fugene HD (Promega, E2311), Fugene 6 (Promega, E2691) or the Xfect (TakaraBio, 631318) transfection reagent, according to the manufacturer's protocol.

## RNA isolation, cDNA synthesis and RT–qPCR

The Power SYBR Green Cells-to-CT kit (Thermo Fisher Scientific) was used according to manufacturer's protocol to isolate RNA, reverse transcribe the RNA and synthesis cDNA from cells. Quantitative PCR was performed in a SureCycler 8800 (Agilent Technologies) using specific primers for DNAJB4 (forward: CGCGGTGATGCTCCTGAAAA; reverse: TGCTCCTCCTTTCAACCCCTG) and GAPDH (forward: GGGAACGCATT-GACTGTTTT; reverse: CTCGGGCTTCTCAAAGTCAC). Knockdown efficiency was determined by calculating the $\Delta$CT score and comparing the control with DNAJB4-depleted conditions.

## Automated image acquisition

An IncuCyte S3 Live-Cell Analysis System (Essen BioScience) was used to acquire live-cell images in the phase contrast, GFP (green) and mCherry (red) channels using a ×20 lens for automated fluorescence signal acquisition for dualPIM cells, tandem-p62 cells, Ub-R–GFP cells and HTT-polyQ cells. These analyses were performed in 96-well plates and for each condition, 3–4 images per well (approximately 100–200 cells per image; total of at least 1,000 cells per experiment) were acquired in 2–3 replicate wells per experimental condition. Cells were imaged every 30 min with an exposure time of 300 ms and 400 ms for the green and the red channels, respectively. The basic analyser function of the IncuCyte software was set up using the TopHat segmentation method to examine the acquired images in an automated manner, and determine the intensity and shape of the fluorescent puncta (the aggregates) in the green and the red (displayed in magenta) channels, and their overlap/white (colocalization). For the green and red (displayed in magenta) channel, the TopHat parameter was set to 10 and 100 µm radius, respectively, and the colour unit threshold was set to 0.7 for both channels. The overlap/white puncta were then calculated based on the green and magenta overlaying signals (Extended Data Fig. 1c). Raw values, generated using Incucyte software, were used to determine the average size (average object size option) of the dualPIM puncta/aggregates. The percentage of overlay/white puncta was determined by dividing the overlap/white area by the magenta area. The collected

data from the IncuCyte software were exported to Excel for summarizing the results and statistical analysis. DualPIM aggregate turnover was monitored as the percentage of the white puncta/aggregates over time. To determine the degradation rate for each experiment, raw values were converted to logarithmic values (natural logarithm) and the slopes of the obtained curves were calculated using the SLOPE function in Microsoft Excel and multiplied by −1 to have positive values. HTT-polyQ119–EGFP puncta were quantified using the basic analyser function of the IncuCyte software. The TopHat segmentation method was used to examine the acquired images in an automated manner and to determine the intensity, size and shape of the fluorescent puncta (the aggregates) in the green channel. The formation of the HTT-polyQ puncta was quantified over time (0–24 h) and expressed as AUC or as HTT-polyQ119 puncta size. The block in Ub-R–GFP turnover was also quantified using the basic analyser function of the IncuCyte software as follows. Signals for the cell were measured using the AI confluence segmentation features in the phase-contrast channel, while the green channel segmentation was carried out by surface fit to measure the Ub-R–GFP signal. To determine the GFP-positive cell areas, the total surface areas of green fluorescence were divided by the cell surface area and finally expressed as fold in comparison with the control.

## Single-molecule localization microscopy

Samples were fixed using prewarmed 4% PFA and permeabilized using 0.1% Triton X-100 in PBS for 10 min. Subsequently, cells were washed in PBS and blocked for 30 min in blocking buffer (3% BSA in PBS) and stained with the indicated primary antibodies over night at 4 °C. Cells were then washed three times in PBS for 10 min before being incubated with the secondary antibodies for 1 h at RT. All SMLM imaging was performed in an imaging buffer containing an oxygen scavenger as previously described[120]. In short, stocks of 1 M mercaptoethylamine (MEA; Sigma; 3000-10G) in 250 mM HCl and 70 mg ml$^{-1}$ glucose oxidase (Sigma; G2133-10KU) plus 4 mg ml$^{-1}$ catalase (Sigma; C40-100MG) were prepared and kept at −80 °C. Before imaging, the imaging buffer was freshly prepared by diluting MEA, glucose oxidase with catalase and glucose in 50 mM Tris-HCl, pH 8.0 to final concentrations of 100 mM MEA, 5% glucose, 700 µg ml$^{-1}$ glucose oxidase and 40 µg ml$^{-1}$ catalase.

Samples were mounted on slides containing a concave cavity (Sigma, BR475505). Then, 90 µl of imaging buffer was placed in the cavity before placing the coverslips on top. The coverslips were tightly sealed by removing the surplus buffer using a vacuum pump. The resulting configurations avoid gaseous exchanges and oxygen-induced deterioration of the imaging buffer. Cells were imaged in the same buffer for a maximum of 1 h.

For the SMLM detection and localization time-lapse images of samples were first processed using fast temporal median filter to remove the background fluorescence[121]. Images were acquired using a Nikon Ti microscope with a ×100 Apo TIRF objective (NA 1.49; oil immersion). The images were then analysed using the custom ImageJ plugin DoM (Detection of Molecules; https://github.com/ekatrukha/DoM_Utrecht), as previously described[120]. Reconstructed images were generated by plotting localizations as a two-dimensional Gaussian distribution with s.d. equal to the localization error. Drift correction was performed using spatial cross-correlation between two intermediate reconstructions. For the cluster analysis quantification, reconstructed images were used to perform density-based clustering (DBscan) implemented in a custom MATLAB script. Before clustering, an ROI was drawn based on the manually determined outlines of each single dualPIM aggregate. An aggregate was included only when it had <100 localizations per aggregate. The resulting cluster shape was used for subclustering and visualization. For each aggregate, TAX1BP1 molecules were plotted and colour-coded for local density[122]. Localization density of a single molecule is defined as the number of other localizations within a radius of five times the mean nearest neighbour distance of all molecules within an aggregate. Molecules with a local density value > 40 were

counted as part of a TAX1BP1 cluster. Local density analysis of individual TAX1BP1 molecules yielded localization density of TAX1BP1 on total aggregates and in dense clusters, number of clusters per aggregate and cluster size. In total, 33 aggregates (containing 52 clusters) from four different control cells and 14 aggregates (containing 23 clusters) from five siDNAJB6-treated cells were included in the analysis. Visualization and statistical analyses (Mann–Whitney $U$-Test) were performed in GraphPad Prism (v.9.4.0).

### Immunoelectron microscopy and correlative light-electron tomography

For the IEM analyses, dualPIM cells were grown and treated as indicated, before being fixed by adding an equal volume of culture medium and freshly prepared double-strength fixative (4% formaldehyde, 0.4% glutaraldehyde in 0.1 M phosphate buffer, pH 7.4) for 5 min at RT. This mixture was then replaced by one volume of single-strength fixative (2% formaldehyde, 0.2% glutaraldehyde in 0.1 M phosphate buffer, pH 7.4) and cells were incubated for an additional 3 h at RT. Cells were then embedded following the Tokuyasu procedure before cutting ultrathin cryosections, at −110 °C, using a dry diamond knife (Diatome AG) and a FC7 cryo-ultramicrotome (Leica), as previously described[123]. Ribbons of 4–6 cryosections of 60 nm each were shifted from the knife-edge using an eyelash and picked up in a wire loop filled with a pre-cooled (4 °C) drop of 1% (w/v) methylcellulose (Sigma) and 1.15 M sucrose in 0.1 M phosphate buffer, pH 7.4. Cryosections were thawed on the pick-up droplet and transferred, sections downwards, onto Formvar (Formvar 15/95 resin, Electron Microscopy Sciences) carbon-coated 100 mesh copper grids (type 0100-CU, EMS). Then, the grids were transferred over a series of droplets for washing, blocking and antibody and protein A-gold (CMC, University Medical Center Utrecht) incubation, following a routine labelling procedure[123]. After a final wash in distilled water, cryosections were stained with uranyl acetate to contrast membranes. Detection of the dualPIM aggregates by immunogold labelling was carried out using an anti-GFP polyclonal antibody (ab290, Abcam). Detection of Lamp2 was performed using an anti-Lamp2 antibody (BD biosciences, 555803) follow by a rabbit anti-mouse (Rockland, 610-4120) and protein A-conjugated gold (GFP: 15 nm; LAMP2: 10 nm). The ultrathin cryosections were finally viewed in a Talos 200Fi (FEI). For 3D-CLET, cells were also processed following the Tokuyasu procedure as described above and thick cryosections of 400–450 nm, cut at −80 °C, were collected, sections downwards, on Formvar carbon-coated finder grids (type EF-100-CU, EMS). Cryosection were processed for immunogold labelling as above using the anti-GFP polyclonal antibody, with the exception that the incubation time with the primary antibody at RT was of 1.5 h. Nuclei were stained with Hoechst 33342 for 20 min at RT. The GFP, mCherry and Hoechst 33342 signals were captured with the DeltaVision RT microscope before the addition of the fiducial gold (protein A-conjugated 15 nm gold, CMC, University Medical Center Utrecht), required for electron tomography, and the membrane staining as previously described[124]. For standard electron tomography, Hoechst 33342 staining and imaging were omitted for the above-described procedure. Dual tilt series images of the thick cryosections were recorded using a 200 kV Tecnai T20 (FEI) using an angular range of typically −55° to +55° with 1° increments, and aligned using at least 20 fiducial gold particles, using the IMOD programme package (University of Colorado). IMOD software was also used to create double tilt tomograms by combining two R-weighted back-projection tomograms. Filtering options in the IMOD package (median and smooth algorithms) were used to 'smooth' the tomograms. The tomograms had a final lateral resolution of approximately 4 nm based on the Crowther criterion. Correlation between the fluorescence and the electron microscopy images was performed using eC-CLEM[125]. Videos were obtained from the tomograms using QuickTime player (Apple). Features of interest were contoured manually in one slice extracted from each tomogram, using IMOD software.

The intensity profiles of the dualPIM aggregates were measured by drawing randomly chosen cross-sections (indicated with a yellow line) within the dualPIM aggregates and using the ImageJ software. The intensity profiles were performed on extracted images from the original tomograms before contrast adjustments. The raw grey values were shown in a line graph for part of each aggregate and the mean values of these cross-sections are indicated as a black horizontal line in each graph. To determine the spreads of the measured values through the cross-section, each measured value in the cross-section was normalized against the mean value of each aggregate and the frequency of values were depicted in a histogram with equally sized bins.

### Transmission electron microscopy

For conventional TEM, cells were fixed using a volume equal to the culture medium of the double-strength fixative (4% paraformaldehyde, 5% glutaraldehyde in 0.1 M sodium cacodylate buffer, pH 7.4) for 20 min at RT. The fixative was replaced with single-strength fixative (2% paraformaldehyde and 2.5% glutaraldehyde in 0.1 M sodium cacodylate buffer, pH 7.4) for 3 h at RT to further fix the cells. After five washes with 0.1 M sodium cacodylate buffer, pH 7.4, cells were post-fixed with 1% $OsO_4$ and 1% potassium ferricyanide (III) in 0.1 M sodium cacodylate buffer, pH 7.4 for 1 h on ice. After five washes with distilled water, cells were processed for dehydration and embedded in Epon resin[126]. During the first step of dehydration, 0.5% uranyl acetate was added to the 70% ethanol solution and cells were incubated overnight at RT. After dehydration, embedding and polymerization in Epon resin, 70-nm ultrathin sections were cut using a Leica EM UC7 ultramicrotome (Leica Microsystems) and stained with lead citrate for 2 min as previously described[126]. Cell sections were analysed using a 120-kV transmission electron microscope JEOL JEM-1400 equipped with a digital camera.

### Statistical analyses

Statistical significance was evaluated using a two-tailed heteroscedastic $t$-test before calculating the $P$ values. Individual data points from each independent experiment are shown in the graphs and were used for the calculation of significance. When multiple groups were compared within one experiment, a one-way analysis of variance (ANOVA) test followed by Tukey's multiple comparison test were performed. A one-way ANOVA with the Dunnett's post hoc test or correcting the $P$ values using the Bonferroni correction was used when multiple treatments were compared with the same controls. The number of independent experiments is indicated in each figure legend. For the analysis of the super-resolution images, a Mann–Whitney non-parametric test was used to compare the differences between the different conditions. In the figures, significant differences are indicated by $*P < 0.05$, $**P < 0.01$ and $***P < 0.001$. Non-significant differences are not indicated. Data distribution was assumed to be normal but this was not formally tested. Data collection was not performed blind to the conditions of the experiments, but analysis always was. No statistical methods were used to predetermine sample sizes but our sample sizes are similar to those reported in previous publications[127–129].

### Reporting summary

Further information on research design is available in the Nature Portfolio Reporting Summary linked to this article.

## Data availability

All constructs used in this study are available from the corresponding authors upon reasonable request. The uncropped western blot images are provided in the source data. A detailed list of reagents, including antibodies, plasmids and cell lines, is provided in Methods. All other data supporting the findings of this study are available from the corresponding authors upon reasonable request. Source data are provided with this paper.

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

## Acknowledgements

We thank J. Klumperman (UMCU), G. Strous (Utrecht University), S. Blacklow (Harvard Medical School) and A. Huckriede (University Medical Center Groningen; UMCG) for cell lines, N. Dantuma (Karolinska Institute) for sharing plasmids, and S. Murata (University of Tokyo) and I. Braakman (Utrecht University) for antibodies. We thank A. Sanz for technical assistance with the native gel analysis and D. Jurriens for guidance during STORM imaging. Fluorescence microscopy analyses were performed at the Microscopy & Imaging Centre of the UMCG. FACS analyses and sorting were performed at the FACS Core Facility, Aarhus University, Denmark and the Flow Cytometry Unit of the UMCG. IEM analyses were performed at the Department of Biomolecular Sciences of Cells and Systems of the UMCG and at the Department of Biomedicine of the Aarhus University. Tomography imaging was carried out at the electron microscopy facility of the Faculty of Science and Engineering of the University of Groningen. We thank P. Verlhac for setting up the eC-CLEM (EM/IF overlap) procedure and J. Lykkegaard Karlsen for helping to install and use IMOD-eTomo on their workstation. M. Mauthe, L.C.K., H.H.K. and F.R. are supported by a ZonMW TOP grant (91217002). F.R. is also supported by Open Competition ENW-KLEIN (OCENW.KLEIN.118), SNSF Sinergia (CRSII5_189952), Novo Nordisk Foundation (0066384) and Lundbeck Foundation (R383-2022-180) grants. H.H.K. is also supported by an EU-JPND consortium grant PROTEST (Dutch part funded by the ZonMW Memorable programme), a grant of the Campaign Team Huntington in the Netherlands (project no. 95169). H.H.K., P.N., M.S.H. and S.S.-K. received support from an NWO (De Nederlandse Organisatie voor Wetenschappelijk Onderzoek)-funded NWA-ORC project (Cure-Q). L.C.K. and G.K. are also supported by a consolidator grant from the European Research Council (819219). L.R.d.l.B. and A.S. are supported by the Research Council of Norway through its Centers of Excellence funding scheme (grant no. 262652) and FRIPRO (grant nos 249753 and 314684). This paper is dedicated to the memory of our dear colleague and friend Catherine Rabouille.

## Author contributions

M. Mauthe, F.R. and H.H.K. designed research. M. Mauthe, N.v.d.B., I.O., J.L.D. and E.P.d.M. performed experiments and analysed data. G.K. performed live-cell imaging and SMLM imaging and analysis. M. Mari and K.B.C. carried out the ultrastructural sample preparation, analyses, data recording and processing, and quantifications. S.S.-K. performed native gel experiments and their analysis. M.R.M., P.N. and L.R.d.l.B. generated cell lines and performed corresponding quality control experiments. L.C.K., M.S.H. and A.S. provided advice and guidance on live-cell imaging and cell line generation, and contributed reagents. M. Mauthe, F.R. and H.H.K. wrote the initial paper. All authors contributed critical feedback and input to the writing. H.H.K. and F.R. supervised the study.

## Competing interests

The authors declare no competing interests.

## Additional information

**Extended data** is available for this paper at https://doi.org/10.1038/s41556-025-01747-1.

**Correspondence and requests for materials** should be addressed to Mario Mauthe, Harm H. Kampinga or Fulvio Reggiori.

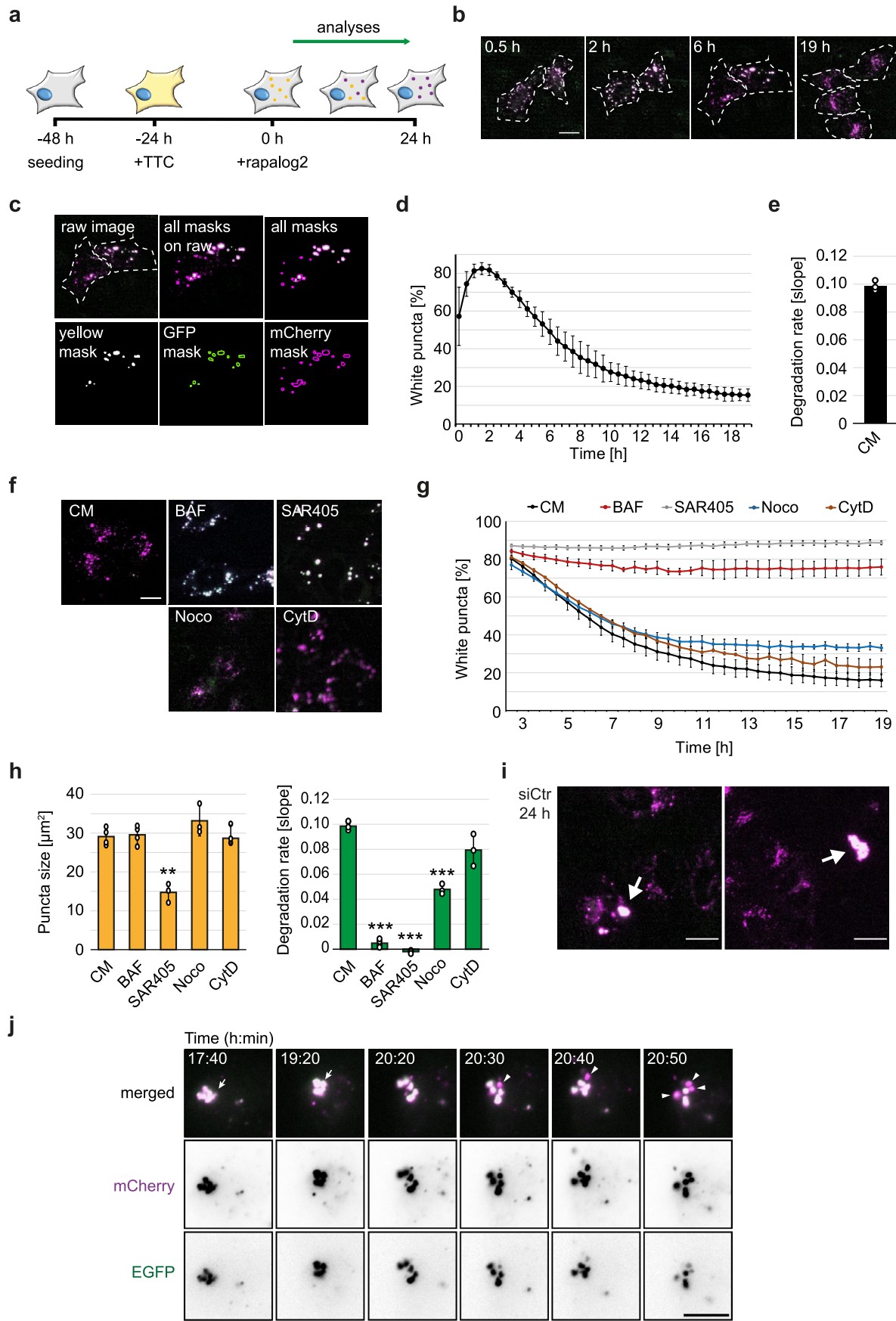

**Extended Data Fig. 1 | See next page for caption.**

**Extended Data Fig. 1 | Image-based quantification of lysosomal dualPIM aggregate turnover. a**, The dualPIM aggregate experiment schematic shows when cells were seeded, TTC-treated, rapalog2-treated and examined. **b**, Representative fluorescence images of dualPIM cells examined by IEM in Fig. 1b. **c**, Automated dualPIM aggregate turnover analysis using the Incucyte system. Separate segmentation masks for GFP, mCherry (magenta) and overlay/white signals as well as their overlay and the overlay with the raw images for the 6 h time point depicted in panel **b** are shown. DualPIM aggregate degradation is quantified by calculating the percentage of magenta puncta (mCherry mask) that are also GFP-positive (overlay mask) over time (white aggregates [%]). The results are displayed either as graphs **(d)** or by calculating curve slopes (degradation rate) **(e)**. **f-h**, Aggregation was induced in dualPIM cells treated with either 100 nM BAF, 10 µM SAR405, 1 µg/ml nocodazol (Noco), 2 mM cytochalasin D (CytD), or left untreated (CM) and imaged for 19 h. **f**, Representative images are shown, and **g**, their degradation rate and **h**, dualPIM puncta average size were quantified. Degradation rate: p = 0.004(SAR vs CM); puncta sizes: p < 0.001(BAF vs CM); p < 0.001(SAR vs CM); p < 0.001(Noco vs CM). **i**, Examples of large, not degraded dualPIM puncta (white arrows) detected at 24 h are shown. **j**, Time-lapse images of HeLa cells showing fragmentation (white arrows) and partial degradation (white arrowheads) of a large dualPIM aggregate. Additional examples are shown in Fig. 1d and Supplementary Video 1. Images in **a** were acquired using the Nikon Eclipse microscope while in **b**, **c**, **f** and **i** using the IncuCyte S3 system. Data are mean values +/- SD. Error bars represent the SD of 3 or 4 (**h**, CM and BAF condition) independent experiments. p-values were calculated by a two-tailed t-test with Bonferroni correction. Scale bars, 20 µm.

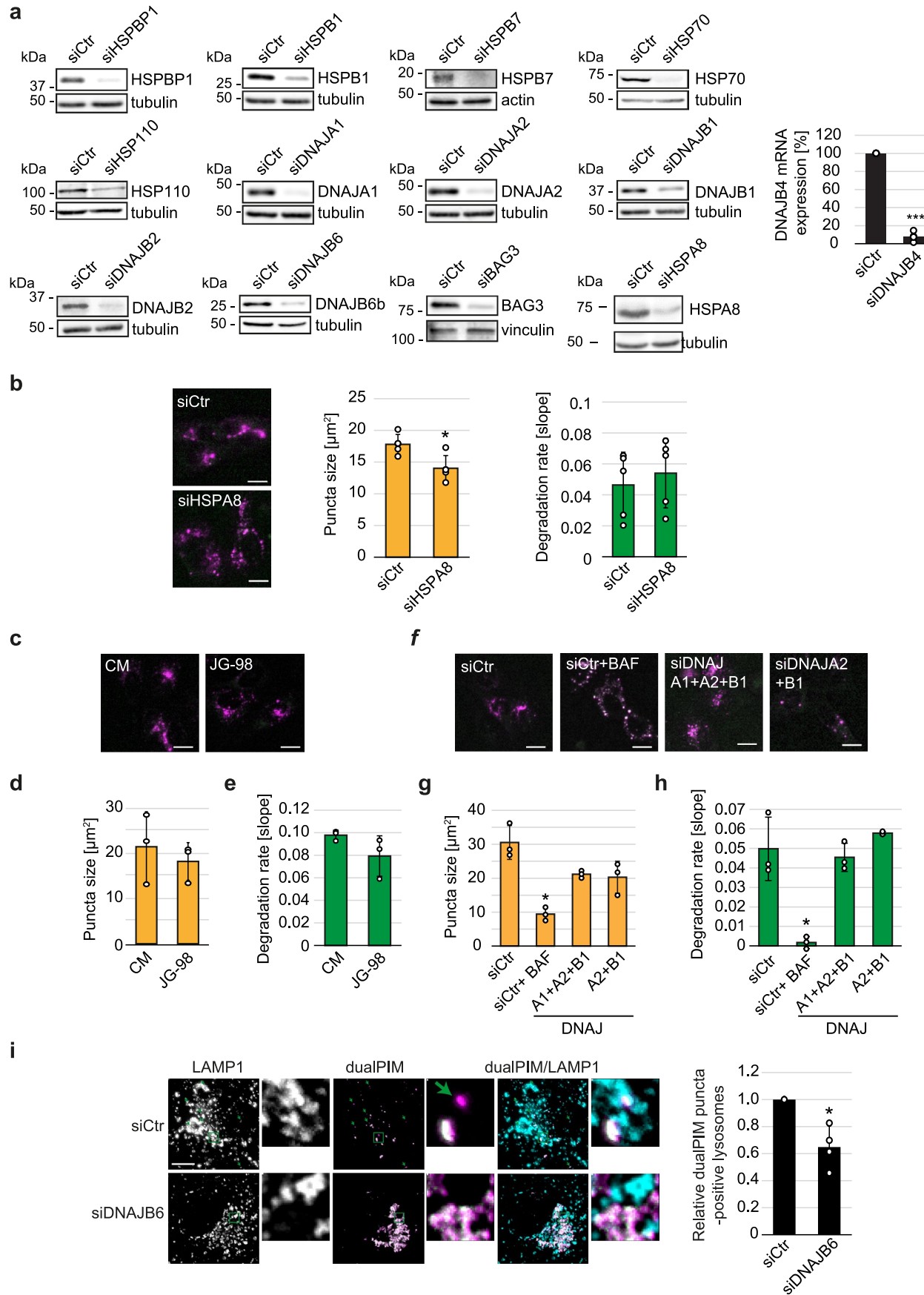

**Extended Data Fig. 2 | See next page for caption.**

**Extended Data Fig. 2 | DNAJB6 depletion blocks dualPIM aggregate transport to lysosomes. a**, DualPIM cells were transfected with the indicated siRNAs as Fig. 1g, to determine the knockdown efficiency by immunoblot with antibodies against the downregulated proteins, except for DNAJB4, which was assessed by RT–qPCR (p < 0.001). Tubulin, actin and vinculin served as loading controls. **b**, HSPA8 (HSC70)-depleted in dualPIM cells were imaged for 24 h. Representative images at 24 h are presented as well as the dualPIM puncta average size at 24 h (p = 0.011) and their degradation rate. **c**, Aggregation was induced in dualPIM cells treated plus or minus 20 nM JG-98 for 19 h, and dualPIM puncta average size (**d**) at 19 h and **e**, their degradation rate is shown. **f**, The indicated genes were depleted in dualPIM cells, which were then imaged for 24 h. Representative images at 24 h are presented as well as the dualPIM puncta average size at 24 h

(**g**) (p = 0.03(siCtr vs siCtr+BAF) and their degradation rate (**h**) (p = 0.02(siCtr vs siCtr+BAF). **i**, DualPIM cells were transfected with siCtr or siDNAJB6 and aggregation was induced for 5 h before being IF with anti-LAMP1 antibodies. Representative images and zoom-in are shown, as well as the colocalization degree of dualPIM (based on the mCherry signal) and LAMP1 relative to the siCtr-treated dualPIM cells (p = 0.02). The overlapping mCherry and LAMP1 signals are highlighted with green arrows. Images in **a** were acquired using the Nikon Eclipse microscope while in **c** and **f** with the IncuCyte S3 system and in i using the DeltaVision microscope. Data are mean values +/- SD. Error bars represent the SD of 3 (**b-h**) and 4 (i) independent experiments. p-values were calculated by a two-tailed t-test with Bonferroni correction. Scale bars, 20 µm, except in panel **a** (scale bars: 5 µm).

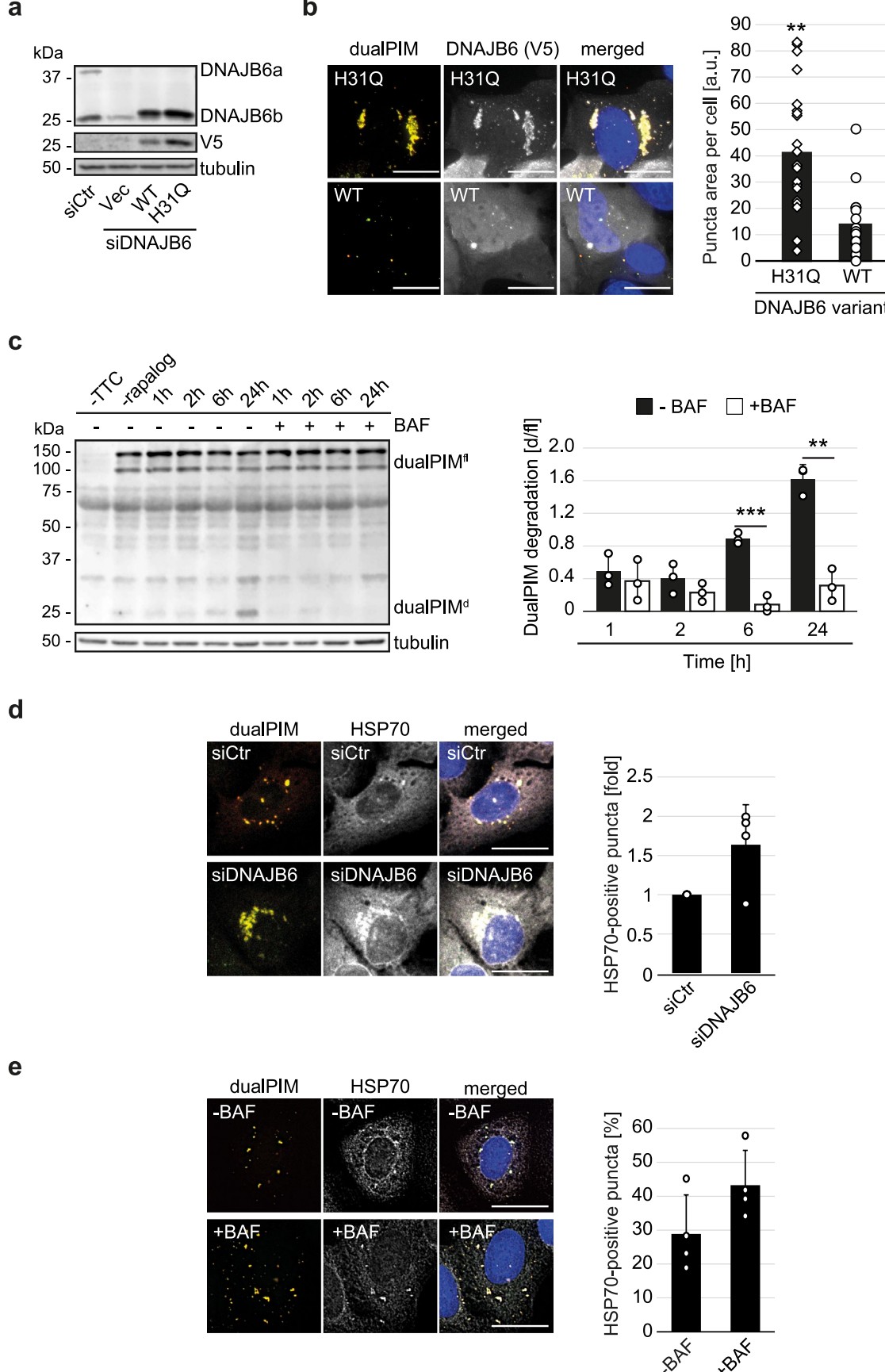

**Extended Data Fig. 3 | See next page for caption.**

**Extended Data Fig. 3 | HSP70 binding-deficient DNAJB6 is unable to promote dualPIM degradation. a**, DualPIM cells were transfected with both siCtr or siDNAJB6 and Vec or a plasmid expressing either V5-DNAJB6$^{WT}$ (WT) or V5-DNAJB6$^{H31Q}$ (H31Q), for 24 h before assessing DNAJB6 knockdown and plasmid expression by immunoblot. Tubulin served as loading control. The anti-DNAJB6 antibody also detects V5-DNAJB6$^{WT}$ (WT) or V5-DNAJB6$^{H31Q}$ (H31Q) (lane 3 and 4). **b**, DualPIM cells were transfected with a plasmid expressing V5-DNAJB6$^{WT}$ (WT) or V5-DNAJB6$^{H31Q}$ (H31Q) for 24 h before aggregation induction (as in Fig. 2c) for 6 h and IF with anti-V5 antibodies. Representative images are shown, and the dualPIM signal area per cell was expressed relative to WT (p < 0.001). **c**, DualPIM cells were left untreated or TTC-treated for 24 h, and incubated with 500 nM rapalog2 for the indicated times plus or minus 100 nM BAF. DualPIM degradation was determined by dividing the lysosomal dualPIM

degradation product (dualPIM$^d$) by the amount of full length dualPIM (dualPIM$^{fl}$) (p < 0.001(6hBAF vs 6hCM); p = 0.001(24hBAF vs 24hCM). **d**, DualPIM cells were transfected with siCtr or siDNAJB6 and aggregation was induced for 6 h, before IF with anti-HSP70 antibodies. Representative images are shown, as well as the relative degree of HSP70 colocalization with the dualPIM puncta. **e**, Aggregation was induced in dualPIM cells for 6 h plus or minus 100 nM BAF, before IF with anti-HSP70 antibodies. Representative images are shown and the colocalization percentage between dualPIM puncta and HSP70 was determined. Images in **d** and **e** were acquired using the SP8X DLS confocal/light sheet microscope. Data are mean values +/- SD. Error bars show the SD of 3 (**b**, **c**) or 4 (**d**, **e**) independent experiments or of 25 individual cells (**b**). p-values were calculated by a two-tailed t-test with Bonferroni correction. Scale bars, 20 μm.

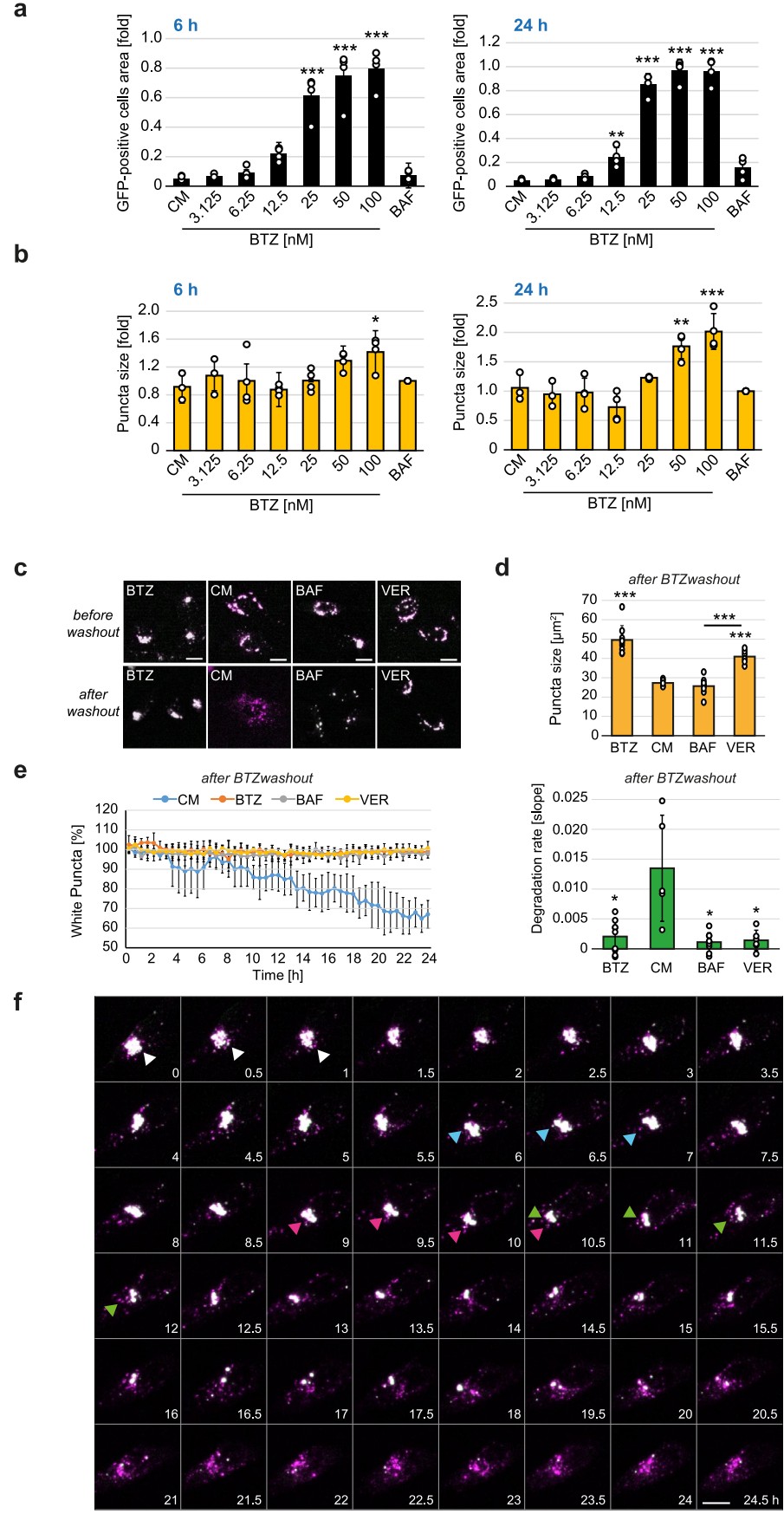

**Extended Data Fig. 4 | See next page for caption.**

**Extended Data Fig. 4 | Fragmentation is a prerequisite for lysosomal dualPIM aggregate delivery.** Ub-R-GFP (**a**) or dualPIM (**b**) cells were treated with the indicated concentration of BTZ, or 100 nM BAF or left untreated (CM), before the GFP-positive cell areas (**a**) (p-values vs CM: $p < 0.001$(100nM-BTZ;6 h); $p < 0.001$(50nM-BTZ;6 h); $p < 0.001$(25nM-BTZ;6 h); $p < 0.001$(100nM-BTZ;24 h); $p < 0.001$(50nM-BTZ;24 h); $p < 0.001$(25nM-BTZ;24 h); $p = 0.005$(12.5nM-BTZ;24 h)) or dualPIM puncta size (**b**) (p-values vs CM: $p = 0.02$(100nM-BTZ;6 h); $p < 0.001$(100nM-BTZ;24 h); $p = 0.001$(50nM-BTZ;24 h)) were quantified at 6 h and 24 h. **c-e**, Aggregation was induced for 6 h in dualPIM cells in the presence of 100 nM BTZ before being washed out and imaged over time. During the washout phase, cells were left untreated (CM), or incubated with 100 nM BTZ, 100 nM BAF or 40 μM VER. Representative IF images 17 h after BTZ washout are shown

(**c**), as well as quantification of the dualPIM puncta average size (**d**) (p-vs CM: $p < 0.001$(BTZ); $p < 0.001$(BAF); $p = 0.001$(VER)) and the degradation rate after BTZ washout (**e**) ($p < 0.001$(BTZ to CM); $p < 0.001$(VER to CM); $p < 0.001$(VER to BAF)). **f**, A dualPIM puncta accumulated in a dualPIM cell upon BTZ treatment was imaged after BTZ washout for a period of 24.5 h. Arrowheads of different colours highlight different fragmentation events. The complete video of this analysis is shown in Supplementary Video 2. Images in **c** and **f** were acquired using the IncuCyte S3 system. Data are mean values +/- SD. Error bars represent the SD of 3 (**a**, **b**) and 5 (**c-e**) independent experiments. p-values were calculated by a two-tailed t-test with Bonferroni correction (**a**, **b**) or with a one-way ANOVA test followed by the Tukey's multiple comparison test (**d**, **e**). Scale bars, 20 μm.

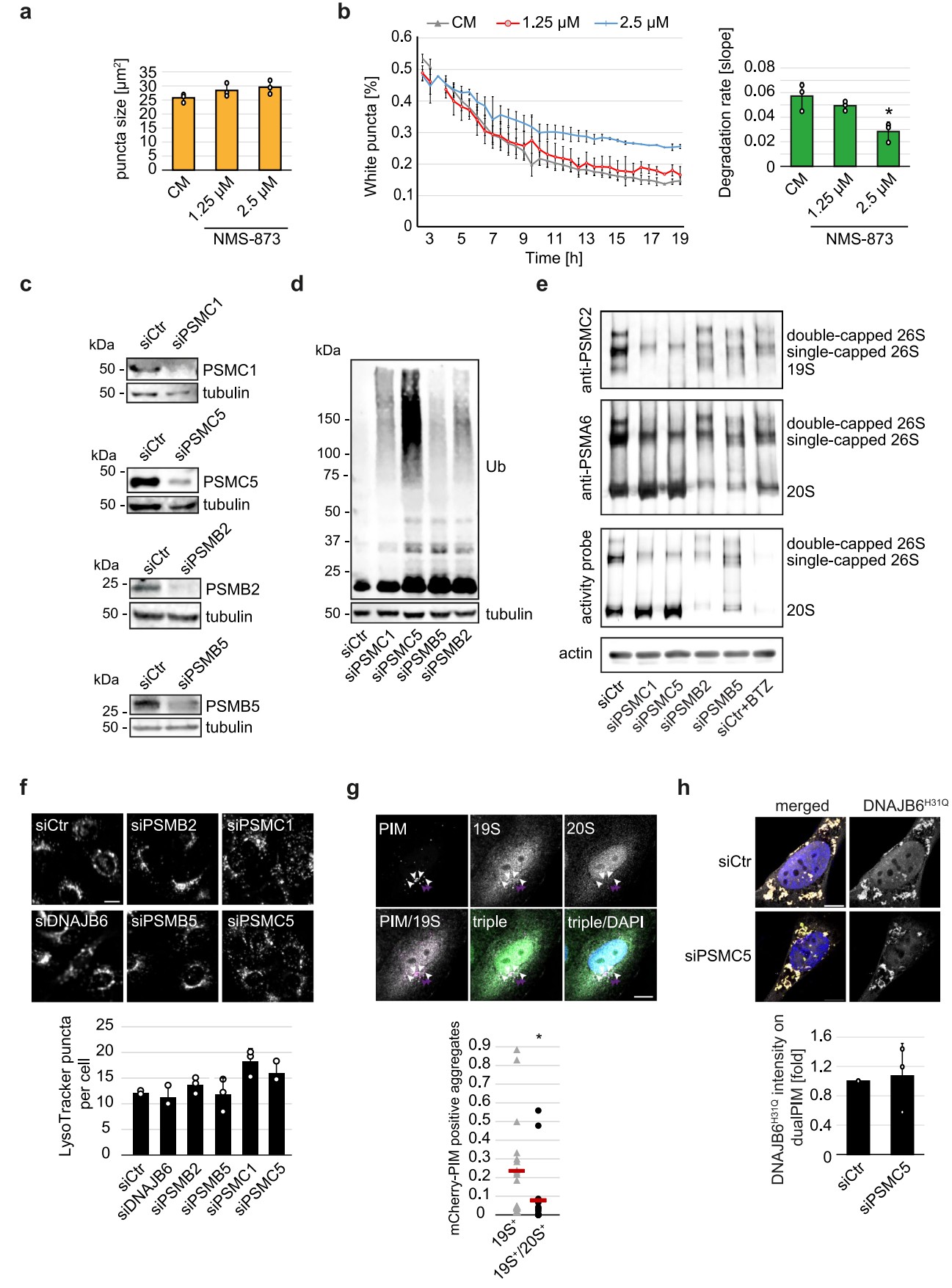

**Extended Data Fig. 5 | See next page for caption.**

**Extended Data Fig. 5 | 19S RP subunit depletion does not inhibit the catalytic activity of 20S CPs. a**, Aggregation was induced in dualPIM cells plus or minus 1.25 µM or 2.5 µM NMS-873 before imaging for 19 h. DualPIM aggregate average size at 19 h and **b**, degradation rate are shown (p = 0.049 NMS2.5 µM vs CM). Of note, higher NMS-873 concentrations enhanced lysosomal clearance but also had toxic effects from 10 h. **c**, PSMB2, PSMB5, PSMC1 or PSMC5 knockdown efficiency in dualPIM cells by western blot. **d**, DualPIM cell transfected with siCtr, siPSMB2, siPSMB5, siPSMC1, or siPSMC5 were analysed by immunoblot with anti-Ub and tubulin antibodies. Tubulin served as loading control. **e**, Proteasomes from dualPIM cells depleted of PSMC1, PSMC5, PSMB2 or PSMB5, or treated with 100 nM BTZ were separated by native gel electrophoresis, and decorated with anti-PSMA6 (20 CP) or PSMC2 (19 RP) antibodies. Proteasomal activity was measured using the ABP4 probe. **f**, U2OS cells were transfected with the indicated siRNAs and LysoTrackerRed-positive puncta number per cell was counted. **g**, Aggregation was induced in mCherryPIM (magenta) cells for 5 h and the percentage of only 19S RP (white arrows) and double 19S RP/20S CP-positive (purple arrows) puncta was determined. **h**, DualPIM cells were transfected with both siCtr or siPSMC5 and a plasmid expressing V5-DNAJB6$^{H31Q}$ for 24 h, before aggregation induction for 5 h and IF staining using anti-V5 antibodies. Representative images are shown and the signal intensity of V5-DNAJB6$^{H31Q}$ in the dualPIM aggregate ROIs was quantified. Images in **g** and **h** were acquired using the SP8X DLS confocal/light sheet microscope and in **f** with the IncuCyte S3 system. Data are mean values +/- SD. Error bars represent the SD of 3 (4 for **g**) independent experiments. p-values were calculated by a two-tailed t-test with Bonferroni correction. Scale bars, 20 µm.

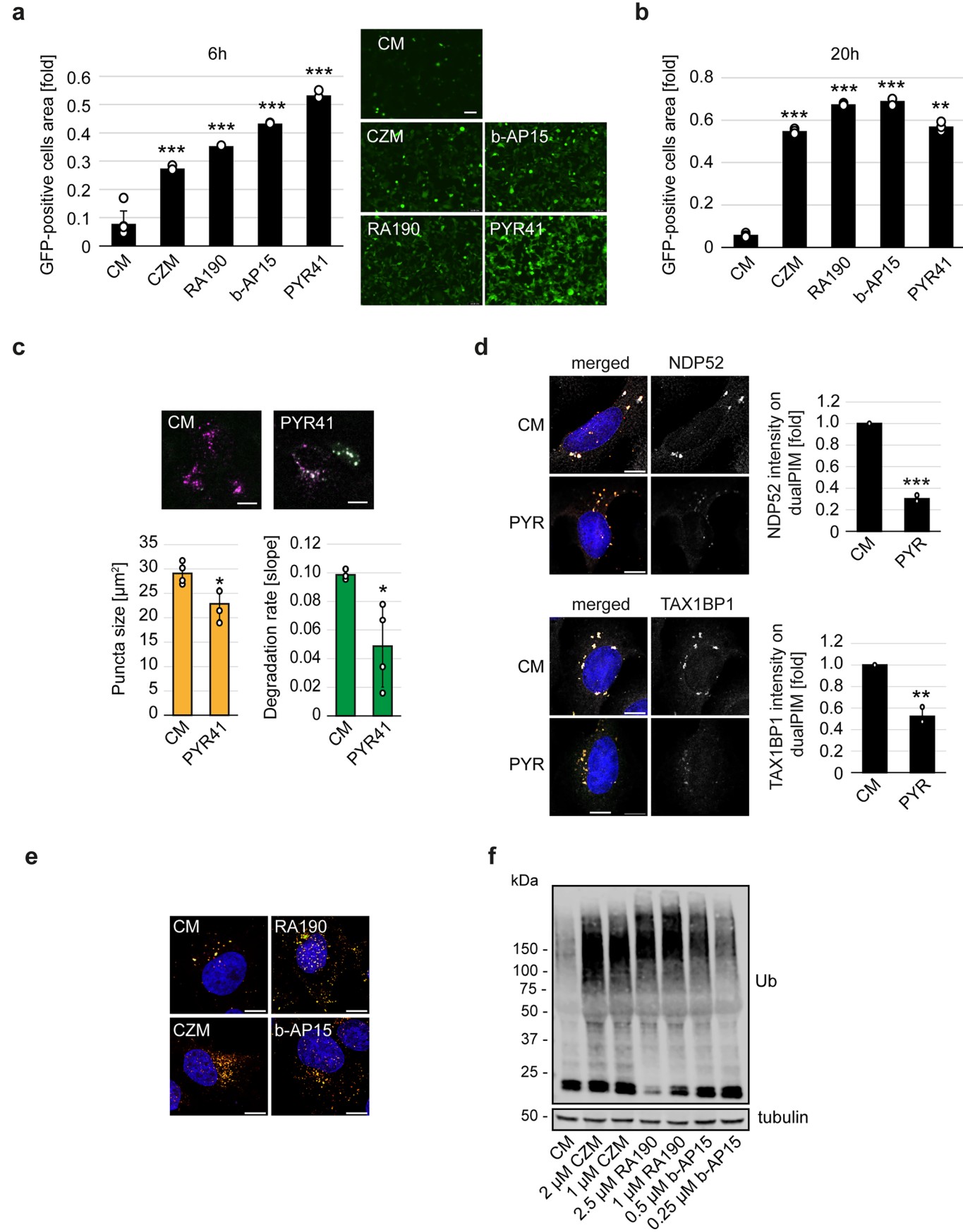

**Extended Data Fig. 6 | See next page for caption.**

**Extended Data Fig. 6 | Ubiquitination of the dualPIM aggregates is required for SAR recruitment.** Ub-R-GFP cells were treated with 1 μM CZM, 1 μM RA190, 0.5 μM b-AP15 or 5 μM PYR-41 and the percentage of the cell area positive for GFP was measured at 6 h (**a**) (p-values to CM: $p < 0.001$ (CZM); $p < 0.001$ (RA190); $p < 0.001$ (b-AP15); $p < 0.001$ (PYR-41)) and 20 h (**b**) (p-values to CM: $p < 0.001$ (CZM); $p < 0.001$ (RA190); $p < 0.001$ (b-AP15); $p = 0.001$ (PYR-41)) post-treatment. **c**, Aggregation was induced in dualPIM cells in the absence (CM) or the presence of 5 μM PYR-41 (PYR) and imaged for 19 h. Representative images are shown as well as dualPIM puncta average size at the 19 h time point ($p = 0.022$) and their degradation rate ($p = 0.037$). **d**, Aggregation was induced in dualPIM cells plus or minus 5 μM PYR-41 (PYR) for 5 h. Cells were processed for IF with antibodies against either NDP52 (left panel) ($p < 0.001$) or TAX1BP1 (right panel) ($p = 0.008$) before to quantify the signal intensities in the dualPIM aggregate ROIs. **e**, Aggregation was induced in dualPIM cells plus or minus 1 μM RA190 (RA190), 1 μM (CZM) or 0.5 μM b-AP15 for 5 h. Representative images at 5 h are shown. **f**, DualPIM cells were treated as indicated for 15 h and cell extracts probed with anti-Ub and anti-tubulin antibodies. Tubulin served as the loading control. Images in **d** and **e** were acquired using the SP8X DLS confocal/light sheet microscope and in **a** and **c** with the IncuCyte S3 system. Data are mean values +/- SD. Error bars represent the SD of 3 independent experiments. p-values were calculated by a two-tailed t-test with Bonferroni correction. Scale bars, 20 μm.

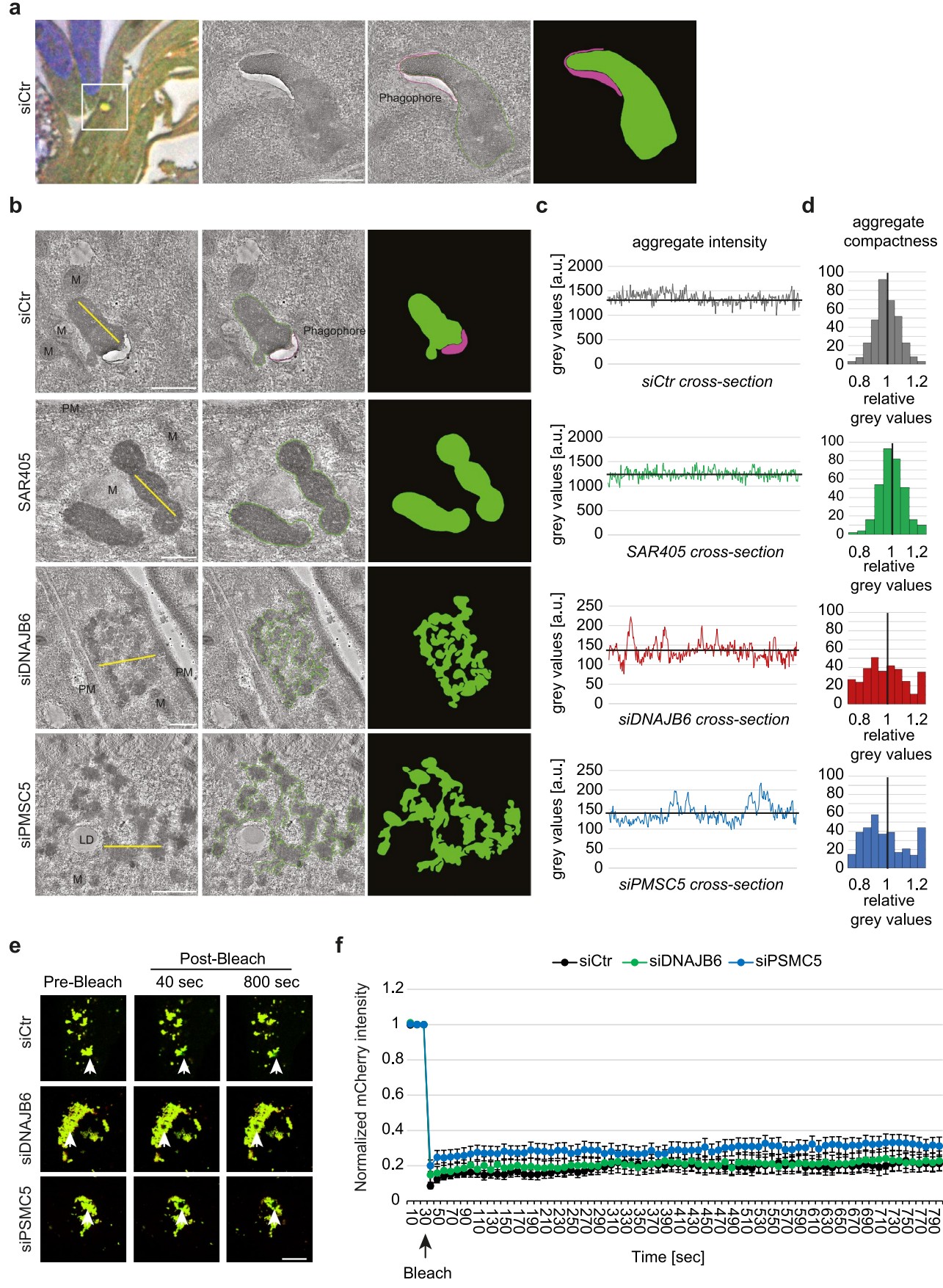

**Extended Data Fig. 7 | See next page for caption.**

**Extended Data Fig. 7 | Phagophore formation adjacently to dualPIM aggregates is absent in DNAJB6-or PSMC5-depleted cells. a**, CLET image provided to the siCtr-treated cells provided in Fig. 4a. **b**, Additional examples of the cryosections analysed in Fig. 4a. Representative images from single slices and visual models are presented and the corresponding tomograms are shown in Supplementary Videos 7–10. Aggregate, green; phagophore, pink; LD, lipid droplet; M, mitochondria; N, nucleus; PM, plasma membrane. Scale bars: 200 nm. **c**, The intensity profiles were generated as in Fig. 4b. Black lines indicate the mean grey value. **d**, The intensity values of the line graphs from panel **c** were normalized for each condition and the spreads of the differences compared to the mean are depicted with histograms. **e**, dualPIM inclusions accumulated in dualPIM cells depleted of DNAJB6 or PSMC5 were bleached (white arrow). **b**, Fluorescent recovery of the bleached area was measured for 13 min (80 images in 10 s intervals).

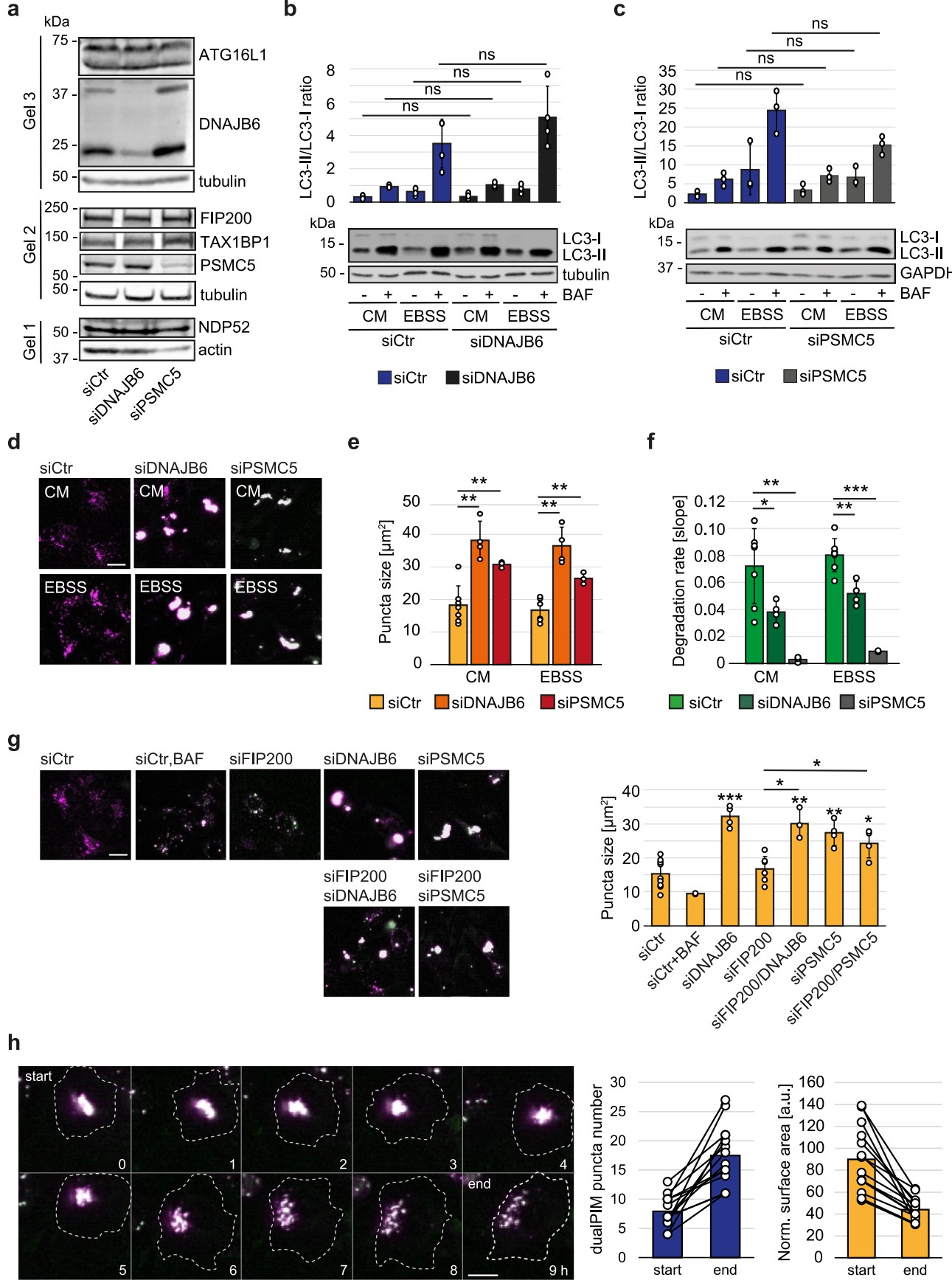

**Extended Data Fig. 8 | See next page for caption.**

**Extended Data Fig. 8 | Bulk autophagy induction cannot rescue the fragmentation defect caused by DNAJB6 or 19 RP deficiency. a**, DualPIM cells were transfected with siCtr, siDNAJB6 or siPSMC5 and aggregation was induced for 5 h. Cell lysates were probed with the indicated antibodies. Tubulin or actin were loading controls. DualPIM cells were transfected with siCtr or siDNAJB6 **(b)** or siCtr or siPSMC5 **(c)**, and treated with starvation medium (EBSS) or not (CM), for 2 h plus or minus 100 nM BAF. Cell lysates were probed with antibodies against LC3 and tubulin or GAPDH. Tubulin and GAPDH were loading controls. Autophagy induction was determined by LC3-II/LC3-I ratio measurement. **d**, DualPIM cells were transfected with siCtr, siDNAJB6 or siPSMC5 and aggregation was induced in control (CM) or starvation medium (EBSS) for 24 h. Representative images before and after EBSS treatment. **e**, Aggregate average size at 24 h (p-values vs siCtr-CM: p = 0.002(siDNAJB6-CM); p = 0.002(siPSMC5-CM); vs siCtr-EBSS: p = 0.004(siDNAJB6-EBSS); p = 0.002(siPSMC5-EBSS)) and **f**, the degradation rates after transfer to EBSS in Panel **d**'s experiment were quantified (p-values vs siCtr-CM: p = 0.017(siDNAJB6-CM); p < 0.001(siPSMC5-CM); P-values vs siCtr-EBSS: p = 0.002(siDNAJB6-EBSS); p < 0.001(siPSMC5-EBSS)). **g**, DualPIM cells were transfected with siCtr, siDNAJB6, siFIP200, siPSMC5, or siFIP200 in combination with siDNAJB6 or siPSMC5, and aggregation was induced for 24 h before dualPIM puncta average size quantification (p < 0.001(siCtr vs siDNAJB6); p < 0.001(siCtr vs siDNAJB6/FIP200); p < 0.001(siCtr vs siPSMC5); p = 0.008(siCtr vs siPSMC5/FIP200); p < 0.001(siFIP200 vs siDNAJB6/FIP200); p = 0.048(siFIP200 vs siPSMC5/FIP200). Representative images at 24 h. **h**, FIP200KO dualPIM cells were pre-treated with 100 nM BTZ for 6 h, and dualPIM puncta number (p < 0.001) and size (p < 0.001) was assessed before and after washout. Images in **d**, **g** and **h** were acquired using the IncuCyte S3 system. Data are mean values +/- SD. Error bars show the SD of 3 (**e**, **f**) or 4 (**c**, **d** and **g**) independent experiments or 14 cells (**h**). p-values were calculated by a two-tailed t-test with Bonferroni correction (**b**, **c**, **e**, **f**, **h**) or with a one-way ANOVA test followed by the Tukey's multiple comparison test (**g**). Scale bars, 20 μm.

**a**

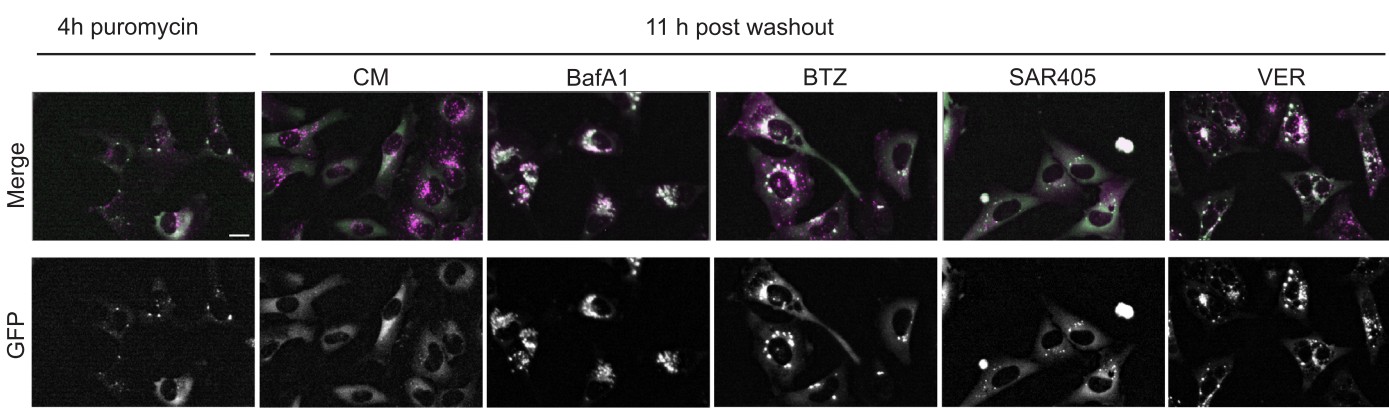

**b**

**c**

*untreated (CM)*

*puromycin-treated (puro)*

**d**

4h puromycin | 11 h post washout

**Extended Data Fig. 9 | See next page for caption.**

**Extended Data Fig. 9 | The DNAJB6-HSP70-HSP110 machinery inhibits puromycin-induced aggregate clearance in multiple cell lines. a**, U2OS cells were transfected with siCtr, siDNAJB6 or siFIP200, and processed as for the wash+BAF treatment in Fig. 6a. Quantification of the number of Ub/p62 double-positive puncta that colocalize with LAMP1-positive compartments and expressed relative to the siCtr control (p = 0.011(siCtr vs siDNAJB6); P = 0.026(siCtr vs siFIP200)). **b**, Knockdown efficiency of HSP110, DNAJB6, FIP200, TAX1BP1, BAG3, PSMB2, PSMB5, PSMC1 and PSMC5 in U2OS cells was assessed by western blot. **c**, The number of Ub/p62 double-positive puncta per cell of untreated (CM) or puromycin-treated U2OS cells (puro) in the experiment shown in Fig. 6c are shown and expressed relative to siCtr. The quantification of the Ub/p62 double-positive puncta after washout and representative images are shown in Fig. 6c. **d**, mCherry-GFP-p62 cells were pre-treated with puro for 4 h and imaged during the washout-out period in the presence of 100 nM Baf, 10 μM SAR, 100 nM BTZ or 40 μM VER, or no treatment. Representative images are shown. Images in **a** were acquired using the DeltaVision microscope and in **d** with the IncuCyte S3 system. Data are mean values +/- SD. Error bars represent the SD of 3 independent experiments. p-values were calculated by a two-tailed t-test with Bonferroni correction. Scale bars, 20 μm.

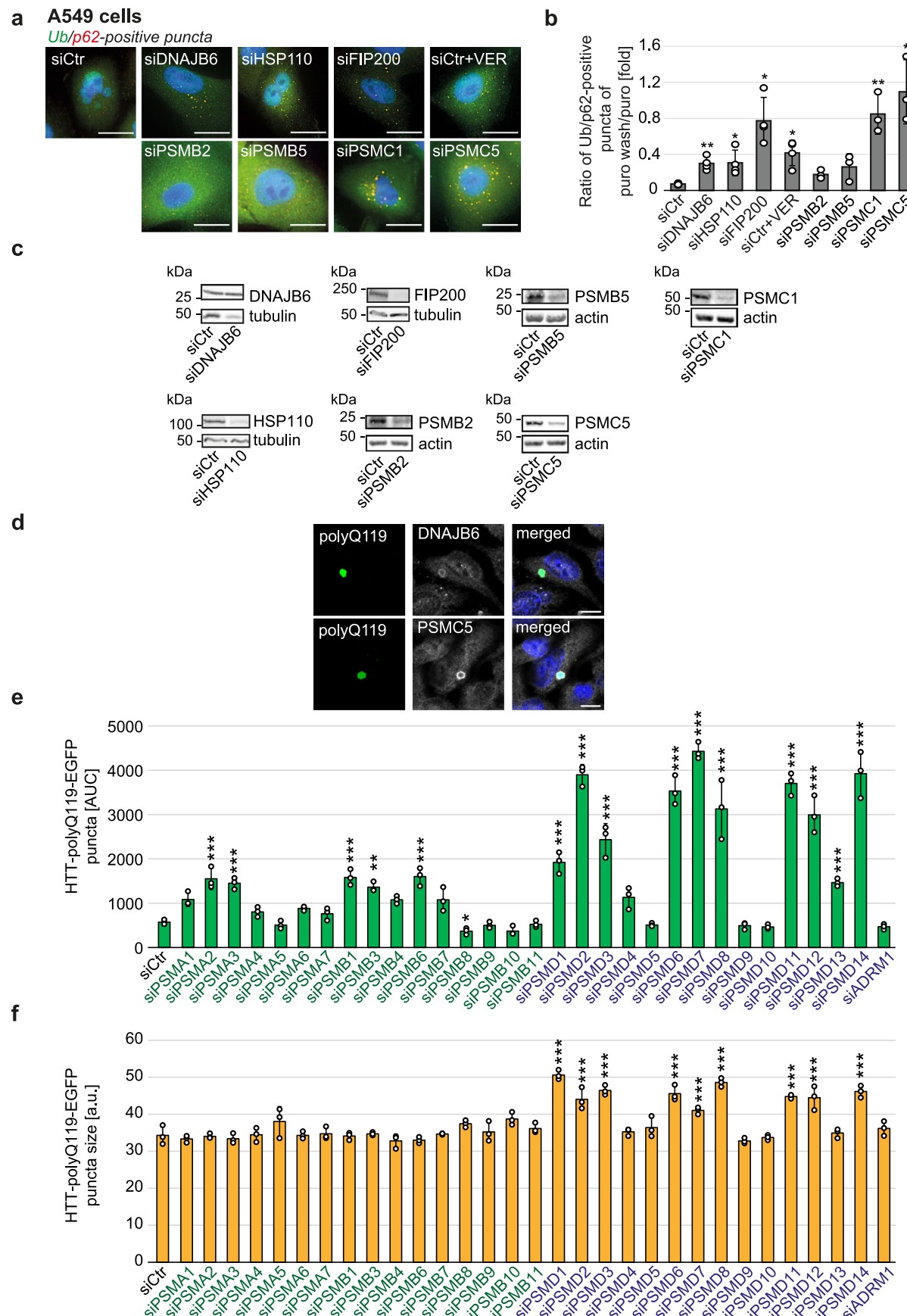

**Extended Data Fig. 10 | See next page for caption.**

**Extended Data Fig. 10 | Depletion of components of the fragmentation machinery inhibits puromycin-induced aggregate clearance and accelerates HTT-polyQ aggregate formation. a**, A549 cells were transfected with siCtr, siDNAJB6, siHSP110 (combination of siHSPH1, siHSPH2, siHSPH3), siFIP200, siPSMB2, siPSMB5, siPSMC1 or siPSMC5 and treated as indicated in Fig. 6a. 40 μM VER was added during the washout period in siCtr cells. **b**, The number of Ub/p62 double-positive puncta per cell was determined for each condition and the ratio of remaining aggregates after puro washout was evaluated as in Fig. 6d. p-values vs siCtr: p = 0.007(siDNAJB6); p = 0.013(siFIP200); p = 0.022(siCtr+VER); p = 0.006(siPSMC1); p = 0.015(siPSMC5). **c**, Knockdown efficiency of DNAJB6, HSP110, FIP200, PSMB2, PSMB5, PSMC1 and PSMC5 in A549 cells was determined by western blot using specific antibodies. Tubulin and actin served as the loading control. **d**, HTT-polyQ119 aggregates were induced for 24 h with TTC and cells were then stained with anti-DNAJB6 or anti-PSMC5 antibodies. HTT-polyQ119 puncta amount (expressed as AUC) (**e**) (p-values vs siCtr: p < 0.001(siPSMA2);

p < 0.001(siPSMA3); p < 0.001(siPSMB1); p < 0.001(siPSMB6); p < 0.001(siPSMD1); p < 0.001(siPSMD2); p < 0.001(siPSMD3); p < 0.001(siPSMD6); p < 0.001(siPSMD7); p < 0.001(siPSMD8); p < 0.001(siPSMD11); p < 0.001(siPSMD12); p < 0.001(siPSMD13); p < 0.001(siPSMD14)) and the average size of the individual aggregates (**f**) (p-values vs siCtr: p < 0.001(siPSMD1); p < 0.001(siPSMD2); p < 0.001(siPSMD3); p < 0.001(siPSMD6); p < 0.001(siPSMD7); p < 0.001(siPSMD8); p < 0.001(siPSMD11); p < 0.001(siPSMD12); p < 0.001(siPSMD14)) upon treatment with the indicated siRNAs was assessed after 24 h of expression. Images in **a** were acquired using the DeltaVision microscope and in **d** with the SP8X DLS confocal/light sheet microscope. Data are mean values +/- SD. Error bars represent the SD of 3 independent experiments. p-values were calculated by a two-tailed t-test with Bonferroni correction (**b**) or one-way ANOVA following Dunnett's post hoc test (**e**, **f**). Scale bars, 20 μm.

Harm Kampinga
Mario Mauthe

# Reporting Summary

## Statistics

For all statistical analyses, confirm that the following items are present in the figure legend, table legend, main text, or Methods section.

| n/a | Confirmed | |
|---|---|---|
| ☐ | ☒ | The exact sample size (*n*) for each experimental group/condition, given as a discrete number and unit of measurement |
| ☐ | ☐ | A statement on whether measurements were taken from distinct samples or whether the same sample was measured repeatedly |
| ☐ | ☒ | The statistical test(s) used AND whether they are one- or two-sided<br>*Only common tests should be described solely by name; describe more complex techniques in the Methods section.* |
| ☒ | ☐ | A description of all covariates tested |
| ☒ | ☐ | A description of any assumptions or corrections, such as tests of normality and adjustment for multiple comparisons |
| ☐ | ☒ | A full description of the statistical parameters including central tendency (e.g. means) or other basic estimates (e.g. regression coefficient) AND variation (e.g. standard deviation) or associated estimates of uncertainty (e.g. confidence intervals) |
| ☐ | ☒ | For null hypothesis testing, the test statistic (e.g. *F*, *t*, *r*) with confidence intervals, effect sizes, degrees of freedom and *P* value noted<br>*Give P values as exact values whenever suitable.* |
| ☒ | ☐ | For Bayesian analysis, information on the choice of priors and Markov chain Monte Carlo settings |
| ☒ | ☐ | For hierarchical and complex designs, identification of the appropriate level for tests and full reporting of outcomes |
| ☒ | ☐ | Estimates of effect sizes (e.g. Cohen's *d*, Pearson's *r*), indicating how they were calculated |

*Our web collection on statistics for biologists contains articles on many of the points above.*

## Software and code

Policy information about availability of computer code

| | |
|---|---|
| Data collection | Fluorescence microscopy: DeltaVision Elite RT microscope system (GE Healthcare, Applied Precision) with SoftWoRx software (Applied Precision; version 7.0.0 Software Suite). Raw microscope images were deconvolved with SoftWorx software (Applied Precision; version 7.0.0 Software Suite). SP8X DLS confocal/light sheet microscope (Leica microsystems) with a LAS X acquisition software 5.3.0; Nikon Eclipse TE2000E ; Zeiss LSM 780 laser scanning confocal microscope with a ZEN black acquisition software 2.3.; IncuCyte S3 Live-Cell Analysis System (Essen BioScience) with an IncuCyte software version 2023A.<br>Electron microscopy: Tecnai T20 (FEI);  JEOL JEM-1400<br>Western blotting (protein detection): Odyssey® XF Imaging System (LI-COR Biosciences) with Empiria Studio software (LI-COR Biosciences; version 2.3). ChemiDoc Touch Imaging System (Bio-Rad) with Image Lab software v. 6.0 (Bio-Rad) |
| Data analysis | Data analysis:<br>Western blotting (protein detection): ImageJ (1.54f)<br>Electron microscopy: IMOD package 5.1.<br>Fluorescence microscopy: SoftWoRx software (Applied Precision; version 7.0.0 Software Suite); ImageJ (1.54f), using the Imagej plugin DoM (Detection of Molecules, https://github.com/ekatrukha/DoM_Utrecht) for SMLM.<br>Icy version 2.5.2.0 using the spot detection algorithm.<br> IncuCyte software version 2023A<br>Data preparation: Microsoft Office Excel (Microsoft; version 2016).<br>Data representation: Microsoft Office Excel (Microsoft; version 2016) or GraphPad Prism 10 (GraphPad Software Inc.; version 10.3.1 (464)).<br>Statistical analysis: Microsoft Office Excel (Microsoft; version 2016) or GraphPad Prism 10 (GraphPad Software Inc.; version 10.3.1 (464)) or SPSS statistics 28. |

Images and figures preparation: Adobe Illustrator (Adobe; version 2024 28.5 (64-bit)).

For manuscripts utilizing custom algorithms or software that are central to the research but not yet described in published literature, software must be made available to editors and reviewers. We strongly encourage code deposition in a community repository (e.g. GitHub). See the Nature Portfolio guidelines for submitting code & software for further information.

## Data

Policy information about availability of data

All manuscripts must include a data availability statement. This statement should provide the following information, where applicable:
- Accession codes, unique identifiers, or web links for publicly available datasets
- A description of any restrictions on data availability
- For clinical datasets or third party data, please ensure that the statement adheres to our policy

All constructs used in this study are available from the corresponding authors upon reasonable request. The uncropped western blot images are provided in the source data. A detailed list of reagents, including antibodies, plasmids and cell lines, is provided in the Methods. All other data supporting the findings of this study are available from the corresponding authors upon reasonable request. Source data are provided with this paper.

## Research involving human participants, their data, or biological material

Policy information about studies with human participants or human data. See also policy information about sex, gender (identity/presentation), and sexual orientation and race, ethnicity and racism.

| | |
|---|---|
| Reporting on sex and gender | N/A |
| Reporting on race, ethnicity, or other socially relevant groupings | N/A |
| Population characteristics | N/A |
| Recruitment | N/A |
| Ethics oversight | N/A |

Note that full information on the approval of the study protocol must also be provided in the manuscript.

# Field-specific reporting

Please select the one below that is the best fit for your research. If you are not sure, read the appropriate sections before making your selection.

☒ Life sciences  ☐ Behavioural & social sciences  ☐ Ecological, evolutionary & environmental sciences

For a reference copy of the document with all sections, see nature.com/documents/nr-reporting-summary-flat.pdf

# Life sciences study design

All studies must disclose on these points even when the disclosure is negative.

| | |
|---|---|
| Sample size | No statistical methods were used to pre-determine sample sizes but our sample sizes are similar to those reported in previous publications (Claude-Taupinet al, 2021; Mauthe et al, 2016; Mauthe et al, 2018). |
| Data exclusions | No data were excluded from the analyses, except for clear technical failures. |
| Replication | All the experiments were performed at least in triplicate (biological replicates) as indicated in each figure legend. |
| Randomization | Samples collection were randomly collected along the development of this study. |
| Blinding | Data collection was not performed blind, but random, to the conditions of the experiments, but analysis always was. |

# Reporting for specific materials, systems and methods

We require information from authors about some types of materials, experimental systems and methods used in many studies. Here, indicate whether each material, system or method listed is relevant to your study. If you are not sure if a list item applies to your research, read the appropriate section before selecting a response.

## Materials & experimental systems

| n/a | Involved in the study |
|-----|----------------------|
| ☐ | ☒ Antibodies |
| ☐ | ☒ Eukaryotic cell lines |
| ☒ | ☐ Palaeontology and archaeology |
| ☒ | ☐ Animals and other organisms |
| ☒ | ☐ Clinical data |
| ☒ | ☐ Dual use research of concern |
| ☒ | ☐ Plants |

## Methods

| n/a | Involved in the study |
|-----|----------------------|
| ☒ | ☐ ChIP-seq |
| ☒ | ☐ Flow cytometry |
| ☒ | ☐ MRI-based neuroimaging |

## Antibodies

| | |
|---|---|
| Antibodies used | The following primary antibodies were used: rabbit anti-BAG3 (Abcam, ab47124, 1:1000), mouse anti-GAPDH (Fitzgerald Industries International, 10R-G109a, 1:10000), rabbit anti-RB1CC1/FIP200 (Proteintech, 17250-1-AP, 1:1000 WB, 1:50 IF), rabbit anti vinculin (E1E9V) (Cell Signaling, 13901, 1:1000), rabbit anti-TAX1BP1 (Millipore, HPA024432, 1:1000 WB, 1:100 IF), rabbit anti-CALCOCO2/NDP52 (Millipore, HPA023195, 1:1000 WB, 1:100 IF), mouse anti-Ub (FK2) (Enzo Life Sciences, LSI-AB-0120, 1:2000 WB, 1:100 IF), rat anti-HSC70, (1B50 (Enzo Life Sciences, ADI-SPA-815, 1:1000 WB), mouse anti-HSP70 (Enzo Life Sciences, ADI-SPA-810, 1:1000 WB, 1:100 IF), mouse anti-APG2 (Santa Cruz Biotechnology, sc-365366, 1:1000 WB), mouse anti HSPBP1 (OTI1D5)(Novus Biologicals, NBP 2-01168, 1:1000 WB), mouse anti-HSPB1 (G3.1)(Enzo Life Sciences, ADI-SPA-800, 1:1000 WB), mouse anti-HSPB7 (3E11)(Abnova H00027129-M01, 1:1000 WB), rabbit anti-DNAJA1 (Abcam, ab126774, 1:1000 WB), mouse anti DNAJA2 (Sigma-Aldrich, WH0010294M1, 1:1000 WB), rabbit DNAJB1 (Atlas antibodies, HPA063247, 1:1000 WB), rabbit anti-DNAJB2 (Atlas antibodies, HPA036268, 1:1000 WB), rabbit anti-DNAJB6 (a kind gift from Ineke Braakman, Utrecht University, 1:1000 WB, 1:50 IF), rabbit anti-ATG16L (MBL International, PM040, 1:1000 WB, 1:50 IF), rabbit anti-LC3 (Novus Biologicals, NB600-1384, 1:1000 WB), mouse anti-tubulin (Sigma-Aldrich, T5168, 1:10000 WB), mouse anti-actin, C4 (Merck, MAB1501, 1:10000 WB), guinea pig anti-p62/SQSTM1 (Progen, GP62-C, 1:200 IF), mouse anti-p62/SQSTM1 (Abcam, ab56416, 1:2000 WB), mouse anti-LAMP1 (BP Biosciences, 555798, 1:100 IF), mouse anti-RFP (ChromoTek, 6g6, 1:2000 WB), mouse anti-V5 (Thermo Fisher Scientific, R960-25, 1:2000 WB, 1:200 IF), rabbit anti-PSMC1 (Merck, HPA000872, 1:1000 WB, 1:50 IF), rabbit anti-PSMC2 (Cell signaling, 14395S, 1:1000 WB), rabbit anti-PSMC3 (Cell signaling, 13923S, 1:1000 WB), rabbit anti-PSMC4 (Proteintech, 11389-1-AP), rabbit anti-PSMC5 (Merck, HPA064293, 1:1000 WB, 1:50 IF), rabbit anti-PSMC6 (Proteintech, 15839-1-AP, 1:1000 WB), rabbit anti-PSMB5 (Merck, HPA049518, 1:1000 WB, 1:50 IF), rabbit anti-PSMB2 (Merck, HPA026324, 1:1000 WB, 1:50 IF), mouse anti-PSMA7 (Enzo Life Sciences, BML-PW8110, 1:50 IF), anti-rabbit PSMA6 (a kindly provided by Shigeo Murata, University of Tokyo, 1:1000 WB) and mouse anti-PSMC2 (MSS1-104) (Enzo Life Sciences, BML-PW8825, 1:3000 WB). The following secondary antibodies were used for the visualization of the primary antibodies: AlexaFluor488-conjugated goat anti-mouse (Invitrogen, Invitrogen, A-11001, 1:250 IF), AlexaFluor568-conjugated goat anti-mouse (Invitrogen, A-11031, 1:250 IF) or goat anti-rabbit (Invitrogen, A-11011, 1:250 IF) or goat anti-guinea pig (Invitrogen, A-11075, 1:250 IF), AlexaFluor647-conjugated goat anti-mouse (Invitrogen, A-21235, 1:250 IF), AlexaFluor680-conjugated goat anti-mouse (Invitrogen, A-21058, 1:5000 WB, 1:250 IF) or AlexaFluor680-conjugated goat anti-rabbit (Invitrogen, A-21109, 1:5000 WB, 1:250 IF), IRDYE 800-conjugated goat anti-mouse (Rockland, 610-132-121, 1:5000 WB), AlexaFluor647-conjugated donkey anti-rat (Jackson ImmunoResearch, 712-605-153, 1:250 IF), goat anti-chicken AF647 (Invitrogen, A-21449, 1:250 IF), horse radish peroxidase (HRP)-conjugated sheep anti-mouse (GE Healthcare, NXA931, 1:5000 WB). |
| Validation | rabbit anti-BAG3 (Abcam, ab47124, 1:1000) https://www.abcam.com/en-us/products/primary-antibodies/bag3-antibody-ab47124 <br> mouse anti-GAPDH (Fitzgerald Industries International, 10R-G109a, 1:10000) https://www.biosynth.com/p/10R-G109A/gapdh-antibody <br> rabbit anti-RB1CC1/FIP200 (Proteintech, 17250-1-AP, 1:1000 WB, 1:50 IF) https://www.ptglab.com/products/RB1CC1-Antibody-17250-1-AP.htm?srsltid=AfmBOoqBR81AvbectmN_nb2eFX5j5kZqaz9qwwH4wKJNW_S89_67xd6m <br> rabbit anti vinculin (E1E9V) (Cell Signaling, 13901, 1:1000) https://www.cellsignal.com/products/primary-antibodies/vinculin-e1e9v-xp-rabbit-mab/13901?srsltid=AfmBOop7Wj2O_HTymYKJ9vBWdNESgP0PijDNcJHl_Gdj7d297cpTHrpl <br> rabbit anti-TAX1BP1 (Millipore, HPA024432, 1:1000 WB, 1:100 IF) https://www.sigmaaldrich.com/NL/en/product/sigma/hpa024432?srsltid=AfmBOop2LeFGGx_R00ucdZKSSE9cZG33xR1e-huqZuru-rLCiDvSs2PS <br> rabbit anti-CALCOCO2/NDP52 (Millipore, HPA023195, 1:1000 WB, 1:100 IF) https://www.sigmaaldrich.com/NL/en/search/hpa023195?focus=products&page=1&perpage=30&sort=relevance&term=HPA023195&type=product <br> mouse anti-Ub (FK2) (Enzo Life Sciences, LSI-AB-0120, 1:2000 WB, 1:100 IF) https://www.enzo.com/product/anti-ubiquitin-antibody-mab-fk2/ <br> rat anti-HSC70 (Enzo Life Sciences, ADI-SPA-815, 1:1000 WB) https://www.enzo.com/product/hsc70-hsp73-monoclonal-antibody-1b5/ <br> mouse anti-APG2 (Santa Cruz Biotechnology, sc-365366, 1:1000 WB) https://www.scbt.com/p/apg-2-antibody-a-7?srsltid=AfmBOoqodNXe8P69hJsfEgfVbLtKpNcmsbaG3PR6Diz4XU051ZGY-2_j <br> mouse anti HSPBP1 (Novus Biologicals, NBP 2-01168, 1:1000 WB) https://www.novusbio.com/products/hspbp1-antibody-oti1d5_nbp2-01168 <br> mouse anti-HSPB1 (Enzo Life Sciences, ADI-SPA-800, 1:1000 WB) https://www.enzo.com/product/hsp27-monoclonal-antibody-g3-1/ <br> mouse anti-HSPB7 (Abnova H00027129-M01, 1:1000 WB) https://www.abnova.com/en-global/product/detail/H00027129-M01 <br> rabbit anti-DNAJA1 (Abcam, ab126774, 1:1000 WB) https://www.abcam.com/en-us/products/primary-antibodies/dnaja1-antibody-epr7248-ab126774?srsltid=AfmBOoqCv1MFIRLh7JlEaF24-4319CRodb4LQl25sAl3b4Qehyvm1g_E <br> mouse anti DNAJA2 (Sigma-Aldrich, WH0010294M1, 1:1000 WB) https://www.sigmaaldrich.com/NL/en/product/sigma/wh0010294m1 <br> rabbit DNAJB1 (Atlas antibodies, HPA063247, 1:1000 WB) https://www.sigmaaldrich.com/NL/en/product/sigma/hpa063247 <br> rabbit anti-DNAJB2 (Atlas antibodies, HPA036268, 1:1000 WB) https://www.sigmaaldrich.com/NL/en/product/sigma/hpa036268 <br> rabbit anti-DNAJB6 (a kind gift from Ineke Braakman, Utrecht University, 1:1000 WB, 1:50 IF) PMID: 36302971 <br> rabbit anti-ATG16L (MBL International, PM040, 1:1000 WB, 1:50 IF) https://www.mblbio.com/bio/g/dtl/A/?pcd=PM040 <br> rabbit anti-LC3 (Novus Biologicals, NB600-1384, 1:1000 WB) https://www.novusbio.com/products/lc3b-antibody_nb600-1384 |

mouse anti-tubulin (Sigma-Aldrich, T5168, 1:10000 WB) https://www.sigmaaldrich.com/NL/en/product/sigma/t5168
mouse anti-actin, C4 (Merck, MAB1501, 1:10000 WB) https://www.sigmaaldrich.com/NL/en/product/mm/mab1501
guinea pig anti-p62/SQSTM1 (Progen, GP62-C, 1:200 IF) https://www.progen.com/anti-p62-SQSTM1-C-terminus-guinea-pig-polyclonal-serum/GP62-C
mouse anti-p62/SQSTM1 (Abcam, ab56416, 1:2000 WB) https://www.abcam.com/en-us/products/primary-antibodies/sqstm1-p62-antibody-2c11-bsa-and-azide-free-ab56416
mouse anti-LAMP1 (BP Biosciences, 555798, 1:100 IF) https://www.bdbiosciences.com/en-nl/products/reagents/flow-cytometry-reagents/research-reagents/single-color-antibodies-ruo/purified-mouse-anti-human-cd107a.555798?tab=product_details
mouse anti-RFP (ChromoTek, 6g6, 1:2000 WB) https://www.ptglab.com/products/RFP-antibody-6G6.htm?srsltid=AfmBOopuTE7hEfv2xCqUOdUW4U9V-Vz_MTjhvTNwYKMkDdz4uQ8uUDbe
mouse anti-V5 (Thermo Fisher Scientific, R960-25, 1:2000 WB, 1:200 IF) https://www.thermofisher.com/antibody/product/V5-Tag-Antibody-clone-SV5-Pk1-Monoclonal/R960-25
rabbit anti-PSMC1 (Merck, HPA000872, 1:1000 WB, 1:50 IF) https://www.sigmaaldrich.com/NL/en/product/sigma/hpa000872
rabbit anti-PSMC2 (Cell signaling, 14395S, 1:1000 WB) https://www.cellsignal.com/products/primary-antibodies/psmc2-d5t1t-rabbit-mab/14395
rabbit anti-PSMC3 (Cell signaling, 13923S, 1:1000 WB) https://www.cellsignal.com/products/primary-antibodies/psmc3-tbp1-antibody/13923
rabbit anti-PSMC4 (Proteintech, 11389-1-AP) https://www.ptglab.com/products/PSMC4-Antibody-11389-1-AP.htm
rabbit anti-PSMC5 (Merck, HPA064293, 1:1000 WB, 1:50 IF) https://www.sigmaaldrich.com/NL/en/product/sigma/hpa064293
rabbit anti-PSMC6 (Proteintech, 15839-1-AP, 1:1000 WB) https://www.ptglab.com/products/PSMC6-Antibody-15839-1-AP.htm
rabbit anti-PSMB5 (Merck, HPA049518, 1:1000 WB, 1:50 IF) https://www.sigmaaldrich.com/NL/en/product/sigma/hpa049518
rabbit anti-PSMB2 (Merck, HPA026324, 1:1000 WB, 1:50 IF) https://www.sigmaaldrich.com/NL/en/product/sigma/hpa026324
mouse anti-PSMA7 (Enzo Life Sciences, BML-PW8110, 1:50 IF) https://www.enzo.com/product/proteasome-20s-%ce%b17-subunit-monoclonal-antibody-mcp72/
mouse anti-PSMC2 (Enzo Life Sciences, BML-PW8825, 1:3000 WB) https://www.enzo.com/product/proteasome-19s-rpt1-s7-subunit-monoclonal-antibody-mss1-104/
AlexaFluor488-conjugated goat anti-mouse (Invitrogen, Invitrogen, A-11001, 1:250 IF) https://www.thermofisher.com/antibody/product/Goat-anti-Mouse-IgG-H-L-Cross-Adsorbed-Secondary-Antibody-Polyclonal/A-11001
AlexaFluor568-conjugated goat anti-mouse (Invitrogen, A-11031, 1:250 IF) https://www.thermofisher.com/antibody/product/Goat-anti-Mouse-IgG-H-L-Highly-Cross-Adsorbed-Secondary-Antibody-Polyclonal/A-11031
AlexaFluor568-conjugated goat anti-rabbit (Invitrogen, A-11011, 1:250 IF) https://www.thermofisher.com/antibody/secondary/query/*A-11011
AlexaFluor568-conjugated goat anti-guinea pig (Invitrogen, A-11075, 1:250 IF) https://www.thermofisher.com/antibody/product/Goat-anti-Guinea-Pig-IgG-H-L-Highly-Cross-Adsorbed-Secondary-Antibody-Polyclonal/A-11075
AlexaFluor647-conjugated goat anti-mouse (Invitrogen, A-21449, 1:250 IF) https://www.thermofisher.com/antibody/product/Goat-anti-Chicken-IgY-H-L-Secondary-Antibody-Polyclonal/A-21449
AlexaFluor680-conjugated goat anti-mouse (Invitrogen, A-21058, 1:5000 WB, 1:250 IF) https://www.thermofisher.com/antibody/product/Goat-anti-Mouse-IgG-H-L-Highly-Cross-Adsorbed-Secondary-Antibody-Polyclonal/A-21058
AlexaFluor680-conjugated goat anti-rabbit (Invitrogen, A-21109, 1:5000 WB, 1:250 IF) https://www.thermofisher.com/antibody/product/Goat-anti-Rabbit-IgG-H-L-Highly-Cross-Adsorbed-Secondary-Antibody-Polyclonal/A-21109
IRDYE 800-conjugated goat anti-mouse (Rockland, 610-132-121, 1:5000 WB) https://www.rockland.com/categories/secondary-antibodies/mouse-igg-hl-antibody-dylight-800-conjugated-pre-adsorbed-610-145-121/?
AlexaFluor647-conjugated donkey anti-rat (Jackson ImmunoResearch, 712-605-153, 1:250 IF) https://www.jacksonimmuno.com/catalog/products/712-605-153
goat anti-chicken AF647 (Invitrogen, A-21449, 1:250 IF) https://www.thermofisher.com/antibody/product/Goat-anti-Chicken-IgY-H-L-Secondary-Antibody-Polyclonal/A-21449
horse radish peroxidase (HRP)-conjugated sheep anti-mouse (GE Healthcare, NXA931V, 1:5000 WB). https://www.sigmaaldrich.com/NL/en/product/sigma/gena931100ul

# Eukaryotic cell lines

Policy information about cell lines and Sex and Gender in Research

| | |
|---|---|
| Cell line source(s) | U2OS (a kind gift from Ger Strous); A549 (a kind gift from Anke Huckriede); HeLa cells (a kind gift from Judith Klumperman); Flp-In™ U2OS (Invitrogen, K6500-01) was used to generate the following cell lines in this study: FLIP-IN dualPIM U2OS (dualPIM), FLIP-IN mCherryPIM U2OS (mCherryPIM), FIP200KO dualPIM, mCherry-GFP-p62 U2OS (tandem-p62); Flp-In™ T-REx™ HEK293 cells (Invitrogen, R78007) were used to generate the Ub-R-GFP HEK293 (Ub-R-GFP)cell line; stable TTC-inducible HTT-polyQ119-EGFP-expressing HEK293T (HEK-HTT-polyQ119-EGFP) were described in  Fan, S. et al. ACS Cent Sci (2023). |
| Authentication | The different cell lines were distinguished by their morphology and growth rate differences. |
| Mycoplasma contamination | All the employed cell lines were tested negative for mycoplasma. |
| Commonly misidentified lines (See ICLAC register) | No commonly misidentified lines were used. |

## Plants

Seed stocks
N/A

Novel plant genotypes
N/A

Authentication
N/A

