## [Peer Review File · Nature Cell Biology]

A chaperone-proteasome-based fragmentation machinery essential for aggrephagy

Corresponding Author: Professor Fulvio Reggiori

Version 0:

Decision Letter:

Dear Professor Reggiori,

Thank you again very much for submitting your manuscript "A chaperone/proteasome-based fragmentation machinery essential for aggrephagy", to Nature Cell Biology. Your manuscript has now been seen by 3 referees, who are experts in proteasome, proteostasis (Referee #1); autophagy, proteostasis (Referee #2); and chaperones, proteostasis (Referee #3), whose comments are pasted below. In light of their advice, we regret that we cannot offer to publish the study in Nature Cell Biology.

As you will see, although the reviewers found the work interesting, they raised serious concerns that question the strength of the data and of the novel conclusions that can be drawn at this stage. Their points undermined the strength of the model, and we overall concluded that the dataset is too preliminary to pursue at this stage.

We are very sorry that we could not be more positive on this occasion, but we thank you for the opportunity to consider this work and hope you find the reviews useful as you define the next steps for the manuscript.

With kind regards,
Melina

Melina Casadio, PhD
Senior Editor, Nature Cell Biology
ORCID ID: <https://orcid.org/0000-0003-2389-2243>

Although we cannot publish your paper, it may be appropriate for another journal in the Nature Portfolio. If you wish to explore the journals and transfer your manuscript please use our manuscript transfer portal. You will not have to re-supply manuscript metadata and files, unless you wish to make modifications. For more information, please see our http://www.nature.com/authors/author_resources/transfer_manuscripts.html?WT.mc_id=EMI_NPG_1511_AUTHORTRANSF&WT.ec_id=AUTHOR manuscript transfer FAQ page.

Reviewers' comments:

Reviewer #1 (Remarks to the Author):

Mauthe and colleagues present in this interesting manuscript a comprehensive study that unveils novel cellular aggregate fragmentation pathways linking disaggregation machinery comprising the HSP70-HSP110-DNAJB6 module and the 19S proteasomal regulatory particle of the ubiquitin-proteasome system. Using an inducible aggregate reporter system they've developed, they show that this pathway is crucial for the fragmentation of protein aggregates that necessarily precedes their clearance by aggrephagy. The study employs a range of techniques, including semi-quantitative live-cell imaging, electron tomography, and biochemical methods, to elucidate the mechanism and physiological relevance of this process. The authors have effectively used this combination of approaches to build a convincing case for their proposed model, and the extension of their findings to puromycin-induced aggregates and disease-associated huntingtin aggregates strengthens the physiological relevance of their work.

This is a well-structured manuscript with thoughtfully designed experiments that provide significant insights into how aggregated protein deposits may be cleared. It is particularly exciting to read a systematic examination of potential synergistic actions of disaggregases, proteasome and autolysosomal systems. The findings presented are novel and impactful, making the work suitable for the wider readership of Nature Cell Biology. I would definitely cite this work.

I noticed a couple of experimental variations that may benefit from further validation and examination; perhaps the authors may find these points useful:

1. The authors report in the main figures changes to PIM-foci (aggregate) area. As most results are performed by diffraction-limited microscopy, this is not the most suitable analysis. The authors should consider counting the total aggregate numbers, especially as fragmentation is in the focus, likely changing aggregate numbers. There should be mention of how these aggregates and aggregate areas defined.
2. Only in Figure 7 is aggregate counted, but there aggregate areas not reported - may it interpreted it this figure that knock-down of 19S subunits increases aggregate fragmentation as the aggregate number increases?
3. All knock-down experiments are on base subunits of the 19S RP; the authors may wish to knock-down lid subunits to show whether the observed effect is limited to the ATPase subunits. The authors may cite Braun et al., NSMB 1999 which reported the 19S base with putative chaperone activity.
4. I fail to appreciate the experiment that demonstrate aggregate condensation, as claimed in Figure 7, please clarify how aggregate condensation with fragmentation was demonstrated.
5. It is unclear how cellular proteasome and lysosomal activities are affected by the different knock-down experiments. Especially with proteasome knock-down experiments, it may be worthwhile to demonstrate the integrity by native gel assessment of proteasomes (e.g. Elsasser et al. Methods Enzymol. 2005). The authors should also consider whether other 20S CP subunits should be knocked down, as it is possible to replace PSMB2 and PSMB5 with alternative subunits.
6. It is unclear whether 19S RP-dependent aggregate fragmentation is a direct effect or if it may be mediated through other paths. The authors may consider to co-stain to determine whether the 19S-mediated effect is directly related to free 19S RPs or if the proteasome holoenzyme, which is the most prominent form, is involved. Proteasome-mediated fragmentation/reduction of aggregate size in cells has been observed before (Morten et al., PNAS 2022) and should be considered for comparative discussions.
7. Proteasome nomenclature should be consistent for the wider readership, please use either PSMC1/PSMC2 etc. or human Rpt2/Rpt3 etc... and either PSMD14, ADRM1 or human Rpn11, Rpn13. It may be worthwhile to indicate the commonly used nomenclatures on first mention e.g. PMSB5/ β 5 subunit.
8. The dualPIM system is a specialist reporter system developed by the authors. It may be useful to describe this system (e.g. that it's based on FKBP, related to AgDD etc.) as I had to investigate through citations to appreciate this.

other notes:

9. The use of siRNA for knock-downs raises concerns about incomplete depletion and potential compensatory mechanisms. The authors might consider addressing this limitation in their discussion.
10. Figure legends slightly repetitive and reads like sentences from Materials and Methods sometimes. It may also be helpful to move control/validation experiments (e.g. of siRNA) to Supplementary Information.
11. Additionally, a more detailed definition of the amorphous aggregates used in this study would be helpful for readers to better understand the nature of the protein assemblies being investigated.

Reviewer #2 (Remarks to the Author):

The manuscript by Mauthe et al. attempts to answer the question of how large aggregates can be degraded by the lysosomal degradation pathway, in particular by autophagy, given the apparent size limitation of the latter. As a model for aggregation, they build on their previous research in which artificial aggregates produced by Particles Induced by Multimerization (PIMs) were shown to be degraded by autophagy and could be tracked with a dual GFP-mCherry tag that quenches GFP in the lysosome. Using this assay, they show that in this system, small pieces of aggregates are pinched off from large aggregates that are subsequently delivered to the lysosome.

This clearance required the HSP70/HSP110/DNAJB6 machinery and/or the proteasomal 19S RP. Interestingly, the core particle of the proteasome was not required for aggregate degradation, suggesting a function independent of the canonical proteasome role. When DNAJB6 or 19S RP was depleted, the aggregates were less compact as shown by CLEM and degraded less. Clearance is dependent on autophagy because when autophagy was blocked, smaller aggregates still formed but were not degraded. It is concluded that DNAJB6 and/or 19S RP form a fragmentation machinery for PIM aggregates, followed by autophagic degradation. Deletion of DNAJB6 or 19S RP impaired the assembly of the autophagic machinery at the aggregate surface.

In the last part, they extend this analysis to other aggregates. They show that the colocalization of p62 with UB in puromycin-induced aggregates depends on the HSP70/110/DNAJB6, and/or 19S RP machinery. Similarly for Htt, they show that deletion of chaperones increases Htt aggregation (previously shown) and that the 19S PR has a similar effect.

In summary, the study by Mauthe et al. addresses a very important question of how aggregates are removed by autophagy. Throughout the manuscript, the authors claim to have identified an unknown fragmentation machinery, consisting of the HSP70/110/DNAJB6, and/or 19S RP machinery, that acts directly on aggregates to fragment them into small digestible pieces for autophagic clearance. However, whether this is a direct or indirect effect is poorly characterized and not very convincing at this stage. Both chaperones and proteasomes are known components of the proteostasis network responsible for the removal and disaggregation of protein aggregates. Impairment of these networks has been associated with accumulation of protein aggregates established by many groups. Therefore their involvement is not surprising. Surprising, however, is that only the 19S RP and not the full proteasome is important for the subsequent autophagic removal of aggregates. However, mechanistic insights into the action of the 19S are lacking. Also lacking are any mechanistic insights

into how the HSP70/110/DNAJB6 fragment these aggregates into larger junks distinct from normal disaggregates as previously proposed. Overall, the authors provide interesting insights into the degradation of protein aggregates by autophagy, but at the moment most conclusions are not convincingly supported by the experimental data as described below and therefore it is not clear if the proposed fragmentation machinery exist as proposed.

Major concerns:

- 1.) Figure 1 is very exciting because it shows the piecemeal degradation of aggregates over time. Throughout the rest of the manuscript, the authors use endpoint imaging of aggregates to measure their average size. This is very indirect and could result from an increased aggregation rate due to general protein aggregation in proteasome and chaperone mutants. To measure fragmentation more directly, the authors should follow individual aggregates in time-lapse microscopy experiments to measure their size changes over time.
- 2.) Figure 4g: After short times of aggregate formation (compared to long times in Figure 1g). Why are the aggregates more diluted for the dual PIM signal? How can it be excluded that general protein aggregation leads to diluted PIM aggregates that completely change the identity of the aggregate itself, similar to the different packing of aggregates seen in the EM experiments?
- 3.) A key finding is that depletion of proteasome core particles does not affect autophagic degradation of PIM and other aggregates. However, these experiments should be complemented with additional controls to assess proteasome activity. The data in Ext. Data Figure 5c are only partially convincing, as for example PSMB5 has the weakest effect compared to the others, a rather unexpected finding. It is therefore possible that the seen effects correlate with the residual activity. It is standard practice to use model substrates such as ZsProSensor-1 (TaKaRa) to monitor degradation in these mutants.
- 4.) In line, complete inhibition of the proteasome by BTZ, which inhibits the catalytic activity of the 20S core particle, showed the opposite effect as the depletion of core particle subunits resulted in large protein aggregates. This is in stark contrast to the depletion of 20S core subunits, which resulted in a strong fragmentation phenotype. Similar experiments as shown in 1f should be performed with the proteasome inhibitor BTZ. In general, the BTZ should be used to impair proteasome inactivation throughout the manuscript. As a note from the Materials and Methods section, it is unclear how the slopes of the curves obtained in these experiments were calculated.
- 5.) If only the 19S RP is essential, the authors should provide experimental evidence for the statement in lines 289-290 that the ATPase function is required for fragmentation or remove the statement.
- 6.) In line 287 – 288 the authors state “While these treatments led to a proteasomal impairment (Extended Data Fig. 5h), the fragmentation of the dualPIM aggregate was unaffected, i.e., the dualPIM aggregates were small.” Inhibition of DUBs blocks deubiquitylation but does not necessarily inhibit proteasome function as the authors suggest. They should test degradation of model proteasome substrates to assess activity in these experiments.
- 7.) In Figure 5, the authors argue that autophagy receptors are less clustered on the aggregate surface. This could simply be due to the increased surface area of the aggregates themselves. The authors should compare the clustering with respect to the aggregate surface to rule out dilution effects.
- 8.) Since the authors suggest that fragmentation is upstream of the autophagy machinery, the authors should show by live cell microscopy in time-lapse experiments that large aggregates fragment over time into smaller ones that accumulate due to blocking autophagic degradation (e.g., by depletion of FIP200 or p62). This should lead to a concomitant increase in the number of small aggregates.
- 9.) In Figures 6 and 7, the authors attempt to extend their findings to other described protein aggregates. While they are able to show that HSP70/110/DNAJB6 and/or 19S have an effect on aggregate size, they do not provide experiments that really confirm the fragmentation of aggregates. It is essential to show that these aggregates are indeed degraded in a piecemeal fashion. Similar to point 1, the authors should try to follow the size of individual aggregates over time.
- 10.) Figure 7 appears to be very preliminary. For the disease-relevant polyQ aggregate, no experiments dealing with fragmentation or autophagic turnover are presented. Often the standard deviation is very high, questioning the significance of the experiments presented (Figure 7b).
- 11.) Figure 7d and e: In this case, all subunits of the proteasome, including the core particle subunits, have a significant effect on the aggregation of polyQ119 compared to the control. However, in Figure 7c, there is no obvious difference in polyQ119 aggregates.

Minor concerns:

- 12.) Line 135 – 136 page 5 (Extended Data Fig. 1i). Extended Data Fig. 1i does not exist. In contrast Figure e exists twice.
- 13.) Line 149 (Fig. 1F) should be (Fig. 1f) -> Figure 1f does not support the statement of lysosomal degradation should be double cited with Ext. Data Fig. 1f
- 14.) line 159: “While BAF treatment did not increase dualPIM aggregate size (but blocked

dualPIM turnover) (Fig. 1h, i),..." this sentence is misleading as it has an effect on aggregate size – it actually decreases it.

15.) Extended Data Fig. 1e (fluorescence images) and 1g does not correlate. In the images SAR405 shows the largest aggregates to the very small ones in Noco treated cells. It should be made clear what is actually measured in these experiments.

16.) From the legend it is not clear what Extended Data Fig. 1h is actually showing and for which experiment this is the control. I assume Figure 1g.

17.) Figure 2A should be marked that DNAJB6 is overexpressed as it clearly leads to reduction of aggregates in general.

18.) Figure 2c lacks a negative control like +BAF. More importantly why does WT DNAJB6 reconstitution not rescue to the full extent as WT, particularly as no aggregates were detected see Figure 2b.

19.) Figure 2h degradation rate of control cells is 20x lower for BTZ compared to normal PIM aggregates – why?

20.) Extended Fig. 2f. This Figure should show more clearly the overlap of the Lamp1 signal with mCherry only signal from the dualPIMs.

21.) Extended Figure 3c: The full length dualPIM is very low at the 24h timepoint after bafilomycin treatment. Does this argue for proteasomal degradation?

22.) Extended data Fig4 representation of data is not intuitive. It should be labelled that the washout phase contains still different drugs.

23.) Extended Data Figure 5e lacks a control that the E1 enzyme is really inhibited. Particularly since there is still, also less, clear targeting of Ub-dependent autophagy receptors to the aggregates (Ext. Data Fig. 5f). A Western blot as in Ext. Data Fig. 5h would clarify the extent of inhibition.

24.) Line 289 – 290: The authors should provide experimental evidence for the statement that the ATPase function is needed for fragmentation or remove the statement.

25.) Figure 4G: siDNAJB6 the contrasting of the GFP signal seems very strange and unnatural this should be checked and only the overall brightness contrast should be adjusted not the just one.

26.) Extended Data Figure 8b. What is the rationality that siPSMB5 but not siPSMB2 lead to an increase in Ub/p62-positive aggregates. Also why siPSMB5 shows now one of the strongest phenotypes. Significance should be indicated. In line why does KD in A549 cells show the opposite?

27.) Video 6 is not working

Reviewer #3 (Remarks to the Author):

The manuscript „A chaperone/proteasome-based fragmentation machinery essential for aggregate clearance“ by Mauthe et al. investigates cellular mechanisms of aggregate clearance (i.e. aggregate clearance by autophagy). They show that the 19S proteasomal regulatory particle and the chaperones DNAJB6, HSP70 and HSP110 are required for the delivery of aggregated proteins to the autophagy-lysosomal system.

Overall, the findings are very interesting and relevant to a wide audience, and the chemically inducible particle induction by multimerization (PIM) system they use is very elegant, but the data (although very extensive and of high quality) do not support all of the authors' claims. Therefore, the paper needs to be substantially revised.

Specific points

1. The authors imply that DNAJB6 participates in aggregate disaggregation and acts mechanistically similar to DNAJB1. However, there is not a single assay that directly examines the fragmentation of amorphous aggregates and in particular amyloid fibers. They only monitor the dynamics of fluorescent foci in cells. Upon KD of DNAJB6 these foci persist, no smaller foci are pinched off from the large foci, which would be directed to aggregate in the presence of DNAJB6. However, pinching off pieces from the foci does not necessarily mechanistically equate to the amorphous aggregate/amyloid fiber fragmentation mediated by the DNAJB1-HSP70-HSP110 disaggregation machinery. Such visible cellular foci contain a large number of individual aggregates/fibers brought into close proximity at these specific protein quality control regions in the cells for efficient refolding/degradation. The size of these foci does not necessarily correlate with the size of single aggregate particles within them. Therefore, the scale and nature of this type of 'disaggregation' is completely different from the disaggregation mediated by the DNAJB1-HSP70-HSP110 machinery.

To substantiate the claim that DNAJB6 is functionally similar to DNAJB1, the authors would need to show at least an effect on the size of individual aggregates in the cells, e.g. by using semidenaturing agarose gel electrophoresis. And even if DNAJB6 is shown to reduce aggregate size, it could just dissolve loosely or laterally associated fibers/aggregates. To demonstrate that DNAJB6 acts mechanistically similar to DNAJB1 would require an in vitro assay using purified components, showing that DNAJB6-HSP70-HSP110 cooperate and act specifically on aggregated substrates (also see below), reducing the size of pre-formed amorphous aggregates and shortening amyloid fibers. If such evidence cannot be

provided, the authors must clarify that DNAJB6 is important for the dissociation of foci, but that this activity is not comparable to the disaggregation activity of the DNAJB1-HSP70-HSP110 system.

2. The experimental setup does not distinguish whether DNAJB6 acts during aggregate formation or whether it acts on pre-existing aggregates. DNAJB6 is depleted by RNAi before aggregates are formed, so it is not clear whether DNAJB6 acts during aggregate formation or on pre-existing aggregates.

To distinguish between these two possibilities, the authors would need to first induce aggregation and then KD DNAJB6 and other chaperones, etc.

The effect they describe here can also be explained by the fact that DNAJB6 is not present during aggregate formation. The authors have previously shown that DNAJB6 co-condensates with its misfolded substrates during aggregate formation. Foci could consist of such condensates. It is possible that this co-condensation with substrates is necessary to keep the aggregated proteins in a specific state that facilitates disaggregation by the DNAJB1-HSP70-HSP110 machinery and the recruitment autophagy receptors, which subsequently facilitates the pinching off of smaller parts from the large foci. DNAJB6 may therefore only indirectly contribute to the breaking down of foci into smaller units of aggregated proteins by HSP70 disaggregation machinery and the autophagy system.

This would be consistent with the view of some co-authors, stated in a manuscript on bioRxiv: '...suggesting that DNAJB6 prevents polyQ condensates to convert from the soluble to the solid state. This in turn, may keep the polyQ peptides competent for (regulated) degradation and accessible to factors allowing its extraction from the condensed state'. The data shown here support exactly this interpretation. The authors provide no evidence here that DNAJB6 is directly involved in the disaggregation.

3. According to the materials and methods, the authors used only siRNA targeting HSPA1A, an inducible Hsp70, but not HSPA8, the cognate Hsc70. This is surprising as it has been shown that disaggregation activity is mediated by Hsc70 and not Hsp70. The authors should comment on this discrepancy.

4. Every figure legend indicates that a two-tailed T-test was used. I want to believe that this is a copy-paste error. The authors should make sure they correctly state the statistical test they have used, and the t-test is not the correct one for most of their assays.

5. Extended Figure 5: If the authors want to claim that inhibition of VCP has no effect, they should use the typical concentration of the drug, which is 10 μ M for U2OS cells (<https://www.abmole.com/products/nms-873.html>). The data differ significantly even when using low concentrations (2.5 μ M) of the inhibitor. I would assume that the effect is stronger when using the appropriate concentration.

6. Expanded Figure 7, panels b and c: The statistics should be based on the data sets that are compared and on which conclusions are drawn: Therefore, the statistics here should be done on the comparison of the effect of siRNAs versus control and not on the comparison of +/- BAF. Also, the LC3-II/LC3-I ratio of controls is quite different in panel b (CM+BAF \approx 1) compared to panel c (CM+BAF \approx 6), but very homogeneous for the replicates within each data set. Do the authors have an explanation for this?

Mislabeling in panel g: siFIP200 +siPSMC5 image is mislabeled, corresponding graph is correct.

7. The discussion lacks clarity. The authors do not mention the substrate specificity of DNAJs, something they should be well aware of. Consequently, they do not differentiate in the discussion between the observed effects on amorphous and fibrillar aggregates, even though studies have shown that there is a strong substrate specificity of DNAJs. DNAJB1 simply might not recognize amorphous dualPIM aggregates, but this does not mean that it is generally not involved in disaggregation of foci. Moreover, the effect of siDNAJB1 was not even tested on HTT-polyQ19-EGFP aggregates in Figure 6, therefore the authors cannot claim that it has no effect here. Because the Hsp70 disaggregation system is somewhat redundant regarding the collaborating DNAJs (DNAJB1, DNAJB4, DNAJA1 and DNAJA2 do all show disaggregation activity in conjunction with Hsp70 and Hsp110), it could be that a KD of a single DNAJ does not yet have a major effect because the others take over. The authors need to consider this in their discussion.

8. Figure 1c: last panel is wrong: no overlay of EGFP channel with merged channel

9. Figure 6b: Quantification of 'wash+VER' is missing.

10. Mislabeling in Extended data Figure 1: there are two 1e, but no 1i. References in text don't match.

11. Figure panels should match order of appearance. Fig. 2c is mentioned before figure 2a and b. Please change order.

12. Line 67: Typo: expressing aggregate-prone proteins

13. Line 80-82: The sentence 'So far, only DNAJA1 and DNAJB1 (or their combined action) have been shown to participate in the disaggregation of amorphous or amyloid fibril-containing aggregates' is not correct: also DNAJA2 and DNAJB4 have been shown to participate in disaggregation (see papers of Nillegoda and Nachman).

14. Line 112-116: 'From 2 h onwards, the first lysosomal degradation events, i.e., appearance of red-only puncta, were detected. These events progressively increased with time after rapalog2 treatment, with lysosomal localization being complete approximately one day post-rapalog2 treatment (Extended Data Fig. 1b)': This sentence should refer to Extended Data Fig. 1b-d to include quantification and not only 1b.

15. The statement in line 134-135 saying: '...supported by the observation that although almost all the cytoplasmic dualPIM aggregates are completely cleared within 19 h (Extended Data Fig. 1b),' is exaggerated, as it is not clear from Fig. 1b that the aggregates are cleared. This figure shows that most of the aggregates are in lysosomes (red color), cleared would mean that they are degraded, which is not the case.

16. Line 150-154: 'In contrast, hindering components of the cytoskeleton network such as the actin filaments and microtubules, which have been involved in the formation and disassembly of aggresomes, stress granules and Juxta Nuclear Quality control compartment (JUNQ) bodies, using nocodazole and cytochalasin D, respectively did not affect the fragmentation of dualPIM aggregate and only moderately affected their degradation rates (Extended Data Fig.1 f-h)'.': Typo in nocodazole; also re-order nocodazole and cytochalasin D to match the order of actin filaments and microtubules.

17. Line 386: Figure 6c and d: are in capital letters (possibly not the only time this is wrong).

**For Nature Portfolio general information and news for authors, see <http://npg.nature.com/authors>.

Version 1:

Decision Letter:

Our ref: NCB-A54764A-Z

28th May 2025

Dear Dr. Reggiori,

Thank you for submitting your revised manuscript "A chaperone/proteasome-based fragmentation machinery essential for aggrephagy" (NCB-A54764A-Z). It has now been seen by the original referees and their comments are below. The reviewers find that the paper has improved in revision, and therefore we'll be happy in principle to publish it in Nature Cell Biology, pending minor revisions to satisfy the referees' final requests and to comply with our editorial and formatting guidelines.

Thank you again for your interest in Nature Cell Biology Please do not hesitate to contact me if you have any questions.

Sincerely,

Angela R Parrish, PhD
Locum Senior Editor
Nature Cell Biology

Reviewer #1 (Remarks to the Author):

The authors present in this revised manuscript additional new results that support their findings. I am overall content with the rebuttal, the revised version addressing previous comments and remain enthusiastic about this work. A number of minor comments below, which should not delay acceptance for publication.

Point 1 – I'm not sure what the reason is for the authors to omit Fig. 1 for reviewers, which may be included with the rationale for using dualPIM in this assay.

Point 2 – this is supportive of expectations. I appreciate that the authors call this 'fragmentase activity' in their response. The authors may consider using this term more frequently in the manuscript and highlight the distinction to disaggregate.

Point 3 – this is a comprehensive set of knock-downs. It may be worthwhile to mention that Ub-GFP line likely mediates degradation via the UFD pathway, where 20S CP would be required for degradation, whereas readout by puncta size is reflective of fragmentase activity that may or may not require 20S activity.

Point 4 – Based on the response, I think the authors mean to refer to compaction of aggregates rather than ‘condensation’, which suggests to me phase change or phase separation. If FRAP suggests no exchange, i.e. no recovery, is this still protein condensation in the canonical sense? Do the authors mean/think, with reflection on Fig.7, that it’s possible proteasomes have condensed as has been proposed by e.g. Yasuda et al. Nature 2020?

Point 5 – It is good to see the lysotracker experiment. I also enjoyed seeing proteasome particles separated on native gel, and expected perturbations to proteasomal assemblies with the knock-downs. Perhaps another way to phrase this would be ‘knock-down of 19S subunits increased the relative abundance of 20S’ and vice versa. Overall, the knock-downs did affect the overall level of proteasome particles.

I would be cautious though on conclusions one can draw from proteasome activities here. The in-gel digest is, at best, comparing between different proteasome (enzymes) at a single LLVY-AMC (substrate) concentration. This is insufficient to draw conclusions about true enzymatic activities, which would need to be compared using catalytic efficiency (k_{cat}/K_m), which in turn requires measurements at various substrate concentrations. Perhaps leave out the enzymatic activities altogether here to be vigilant.

Point 6 – I think Figure 7 provides evidence for direct involvement of 19S targeting aggregates and should be considered for publication. Presence of free RPs is not surprising and I notice the authors already cited ref 99; as did they cite ref 67 showing in vitro proteasomal fragmentation activity against aggregates.

Reviewer #2 (Remarks to the Author):

I would like to thank the authors for their extensive efforts in strengthening the manuscript with new experiments. Most of the concerns have been addressed.

However, one main concern remains, relating to the interpretation of the new results shown in Figure 3a-b. The authors demonstrate in Figure 3b that the extent of proteasome inhibition differs considerably between the individual subunits. Of concern is that PSMC1 and PSMC5 (~0.8) subunits are twice as efficient as PSMB2 and PSMB5 (~0.4) at blocking proteasome function: These are the subunits later used for 19S and 20S. Therefore, it cannot be excluded that a certain threshold of proteasome inhibition needs to be met before fragmentation is truly prevented. In line with this, the new experiments in Extended Data Figures 4a and 4b show very similar trends. Inhibiting the catalytic activity of the proteasome after 6 hours with 25nM BTZ results in only a ~0.6-fold increase in proteasome activity reporter, while 100nM BTZ results in a ~0.8-fold increase. This is reflected in the size correlation of the aggregate: while 25nM BTZ shows no phenotype, 100nM BTZ already results in a significant decrease in fragmentation, suggesting that indeed proteasome activity is important.

In order to support their claim that the core function of the proteasome is not important for fragmentation, the authors should ensure that the compared knockdowns have a similar impact on the function of the proteasome. However, in my opinion, the manuscript has many novel aspects, so the necessity of core proteasome function does not need to be discussed in such detail. The fact that fragmentation is necessary for autophagy is already a very important advancement. I suggest toning down the statement of the proteasome core activity and discussing the limitations of these results in the 'Discussion' section.

Reviewer #3 (Remarks to the Author):

The manuscript “A chaperone/proteasome-based fragmentation machinery essential for aggregate degradation” by Mauthe et al. investigates how amorphous protein inclusions are processed for autophagic degradation. The authors show that the 19S regulatory particle of the proteasome and the chaperones DNAJB6, HSP70, and HSP110 are required for fragmenting large intracellular inclusions.

The manuscript has improved significantly. I appreciate the authors' efforts; I have rarely seen such a comprehensive revision. Most points I raised in my initial review, have been addressed sufficiently in the revised version. That said, two points still need further attention:

1. HSPA8 vs. HSPA1A (previous point 3):

Thank you for performing these experiments. It is very interesting that depleting HSPA8 (Hsc70) had no effect on dualPIM fragmentation, while HSPA1A (Hsp70) does. In contrast, disaggregation by DNAJB1-Hsp70 systems is more efficient when Hsc70 is involved, as shown by Gao et al., Mol. Cell, 2015: “DNAJB1-Agp2 combinations containing Hsc70 solubilize fibrils better than those containing stress-inducible Hsp70 (Figures S1A and S1B)”. That difference is important and should be included in the final manuscript, not just in the reviewer response, as it adds valuable insight into chaperone specificity in this context and highlights the difference between the DNAJB1-Hsp70-HSP110 disaggregation system and the fragmentation of large intracellular foci described here.

2. Statistical analysis (previous point 4):

Performing multiple separate t-tests without correction introduces a risk of inflated Type I error due to multiple comparisons. Even if the treatments are not directly compared to one another, repeatedly testing against the same control group increases the overall likelihood of detecting a false positive. For example, if four drug treatments are each compared to the same control using separate t-tests at a significance level of $\alpha = 0.05$, the chance of making at least one Type I error across all tests rises to nearly 19% (calculated as $1 - (1 - 0.05)^4 \approx 0.185$). This means that even if all null hypotheses are true, there is an 18.5% probability of falsely concluding that at least one treatment has a significant effect. This is well above the intended 5% error rate.

To avoid this, the authors should use a one-way ANOVA with a post hoc test like Dunnett's (ideal for multiple comparisons against a control). If the authors prefer to use multiple t-tests, p-values should be adjusted (e.g., Bonferroni correction). Alternatively, if all the experiments were done separately with independent controls, that should be made clear in the figure, text and tests.

Version 2:

Decision Letter:

Dear Dr Reggiori,

I am pleased to inform you that your manuscript, "A chaperone-proteasome-based fragmentation machinery essential for autophagy", has now been accepted for publication in Nature Cell Biology.

Please note that *Nature Cell Biology* is a Transformative Journal (TJ). Authors may publish their research with us through the traditional subscription access route or make their paper immediately open access through payment of an article-processing charge (APC). Authors will not be required to make a final decision about access to their article until it has been accepted. <https://www.springernature.com/gp/open-research/transformative-journals> Find out more about

Transformative Journals

Authors may need to take specific actions to achieve compliance with funder and institutional open access mandates. If your research is supported by a funder that requires immediate open access (e.g. according to [Plan S principles](https://www.springernature.com/gp/open-science/plan-s-compliance) or the [NIH public access policy](https://www.springernature.com/gp/open-science/us-federal-agency-compliance)) then you should select the gold OA route, and we will direct you to the compliant route where possible. Because authors warrant under our subscription licensing terms that they haven't committed to licensing any version of their article under a licence inconsistent with the terms of our agreement – including the applicable embargo period – publication under the subscription model isn't suitable for authors whose funders require no embargo.

If you have not already done so, we strongly recommend that you upload the step-by-step protocols used in this manuscript to protocols.io (<https://protocols.io>), an open online resource that allows researchers to share their detailed experimental know-how. All uploaded protocols are made freely available and are assigned DOIs for ease of citation. Protocols and Nature Portfolio journal papers in which they are used can be linked to one another, and this link is clearly and prominently visible in the online versions of both. Authors who performed the specific experiments can act as primary authors for the Protocol as they will be best placed to share the methodology details, but the Corresponding Author of the present research paper should be included as one of the authors. By uploading your Protocols onto protocols.io, you are enabling researchers to more readily reproduce or adapt the methodology you use, as well as increasing the visibility of your protocols and papers. You can also establish a dedicated workspace to collect your lab Protocols. Further information can be found at <https://www.protocols.io/help/publish-articles>.

Nature Cell Biology encourages authors presenting evidence for cell, biological, molecular, and genetic interactions to consider communicating these findings using Biofactoid (<https://biofactoid.org/>). This tool helps users share a searchable representation of interactions (e.g. binding, gene expression, post-translational modification) between genes, gene products, or chemicals. Information added to Biofactoid, with author attribution, is shared on social media and public databases, such as Pathway Commons, where it can be discovered and analyzed in the context of a large and growing corpus of knowledge.

With kind regards,

Angela R Parrish, PhD
Locum Senior Editor
Nature Cell Biology

** Visit the Springer Nature Editorial and Publishing website at http://editorial-jobs.springernature.com?utm_source=ejp_NCB_email&utm_medium=ejp_NCB_email&utm_campaign=ejp_NCB for more information about our career opportunities. If you have any questions please click [here](mailto:editorial.publishing.jobs@springernature.com).

Point-by-point rebuttal

We thank the three reviewers for their excellent and constructive criticisms, which have helped to improve and complete our manuscript. A point-by-point response to each reviewer's comment is given below. Major additions and changes in the manuscript are highlighted in blue.

Overall, to strengthen the conclusions of our manuscript, we have performed the following new experiments:

- *Measurement of non-degraded dualPIM aggregate by live-cell fluorescence microscopy (Fig. 1 for reviewers)*
- *Quantification of the size of the HTT-polyQ119 inclusion in different conditions by live-cell fluorescence microscopy (Fig. 7d)*
- *Analysis of the relevance of each proteasomal subunits in dualPIM aggregate turnover and proteasomal activity, assessed using the Ub-R-GFP cells (Fig. 3a, b)*
- *FRAP analysis of the dualPIM aggregate liquidity (Extended Data Fig. 7e, f)*
- *Examination of the lysosome functionality using LysoTracker-based imaging (Extended Data Fig. 5f)*
- *Proteasome assembly analysis by native gel electrophoresis (Extended Data Fig. 5e)*
- *Examination of the 19S and 20S ribosomal particle distribution onto mCherryPIM puncta by IF (Fig. 7 for reviewers)*
- *Live-cell imaging of dualPIM aggregate in DNAJB6 or PSMC5 knockdown cells over time (Fig. 8 for reviewers)*
- *Analysis of the effects of different concentrations of Bortezomib (BTZ) on both the dualPIM aggregate fragmentation and proteasomal activity (Extended Data Fig. 4a, b and Fig. 10 for reviewers)*
- *Measurement of the average dualPIM puncta size in cells depleted of DNAJB6 and PSMC5 by live-cell fluorescence microscopy (Fig. 11 for reviewers)*
- *Analyzing dualPIM fragmentation in autophagy-deficient cells by live-cell imaging (Extended Data Fig. 8h)*
- *Electron microscopy and live-cell analysis of puromycin-induced aggregate fragmentation and turnover (Fig. 6e and Extended Data Fig. 9d)*

- *Analysis of PSMC5 and DNBAJB6 association with HTT-polyQ119 inclusion by IF (Extended Data Fig. 10d)*
- *Analysis of NDP52 and TAX1BP1 recruitment to HTT-polyQ119 inclusions upon PSMC5 and DNBAJB6 depletion by IF (Fig. 15a, b for reviewers)*
- *Analysis of HTT-polyQ119 inclusions in the absence of 19S subunits and DNBAJB6 using the Incucyte system (Fig. 15c, d for reviewers, Fig. 16 for reviewers)*
- *Examination of the relevance of each proteasomal subunits in HHT-polyQ119 inclusion accumulation (Extended Data Fig. 10e, f)*
- *Analysis of levels of non-degraded dualPIM full length by western blot (Fig. 19 for reviewers)*
- *Analysis of proteasomal inhibition by the different drugs used in this study, employing the Ub-R-GFP cells (Extended Data Fig. 6a, b)*
- *Analysis of HSPA8 depletion on dualPIM aggregate degradation (Fig. 21 for reviewers)*
- *Effects of different concentrations of the VCP inhibitor on dualPIM aggregate turnover using the Incucyte system (Fig. 22 for reviewers)*
- *Examination of the combinatorial depletion of DNAJ proteins on dualPIM aggregate degradation using the Incucyte system (Fig. 1j-i and Extended Data Fig. 2b)*
- *Analysis of DNAJB4 depletion on dualPIM aggregate turnover using the Incucyte system (Fig. 1j-i and Extended Data Fig. 2b)*

Reviewer #1

Mauthe and colleagues present in this interesting manuscript a comprehensive study that unveils novel cellular aggregate fragmentation pathways linking disaggregation machinery comprising the HSP70-HSP110-DNAJB6 module and the 19S proteasomal regulatory particle of the ubiquitin-proteasome system. Using an inducible aggregate reporter system they've developed, they show that this pathway is crucial for the fragmentation of protein aggregates that necessarily precedes their clearance by aggrephagy. The study employs a range of techniques, including semi-quantitative live-cell imaging, electron tomography, and biochemical methods, to elucidate the

mechanism and physiological relevance of this process. The authors have effectively used this combination of approaches to build a convincing case for their proposed model, and the extension of their findings to puromycin-induced aggregates and disease-associated huntingtin aggregates strengthens the physiological relevance of their work.

This is a well-structured manuscript with thoughtfully designed experiments that provide significant insights into how aggregated protein deposits may be cleared. It is particularly exciting to read a systematic examination of potential synergistic actions of disaggregases, proteasome and autolysosomal systems. The findings presented are novel and impactful, making the work suitable for the wider readership of Nature Cell Biology. I would definitely cite this work.

We thank this reviewer for the very positive evaluation of our manuscript.

I noticed a couple of experimental variations that may benefit from further validation and examination; perhaps the authors may find these points useful:

1. The authors report in the main figures changes to PIM-foci (aggregate) area. As most results are performed by diffraction-limited microscopy, this is not the most suitable analysis. The authors should consider counting the total aggregate numbers, especially as fragmentation is in the focus, likely changing aggregate numbers. There should be mention of how these aggregates and aggregate areas defined.

As suggested, we have now determined the percentage of non-degraded dualPIM puncta at the end of the key experiments shown in the original Fig. 1d, 1g and 3b. How this was determined is described in the figure legend to Fig. 1 for reviewers. The results perfectly mirror the quantification of the dualPIM aggregate degradation (Fig. 1 for reviewers). However, they do not reflect the differences between a block in aggregate degradation (e.g., BAF treatment) and one in a fragmentation (e.g., siPSMC5). Therefore, we still believe that the quantification of the dualPIM puncta size allows better to monitor fragmentation defects.

We have now ameliorated the description of how aggregate and aggregate areas are defined in the measurements in the Methods.

a
Related to Fig. 1d-f in original manuscript

Figure 1 for reviewers (not included in the revised manuscript). Measurement of non-degraded dualPIM aggregate. The percentage of non-degraded dualPIM puncta in Fig. 1d-f (panel a), Fig. 1g-l (panel b) and 3b-d (panel c) was quantified as follow: The puncta area (GFP and mCherry-positive dualPIM puncta) per cell at the end of the experiment was divided by the puncta area per cell at the beginning of the experiment.

b
Related to Fig. 1g-i in original manuscript

c
Related to Fig. 3b-d in original manuscript

2. Only in Figure 7 is aggregate counted, but there aggregate areas not reported - may it interpreted it this figure that knock-down of 19S subunits increases aggregate fragmentation as the aggregate number increases?

We thank the reviewer for this excellent comment. We have now provided the quantification of the polyQ inclusion area (Fig. 2 for reviewers). The average size increases when the 19S RP subunits are depleted or when autophagy is inhibited using SAR405. This result argues against the fact that under these conditions there is more fragmentation. For an explanation of this increase in the inclusion size, please refer to our response to comment #11 of reviewer #2. This figure has been added to the manuscript as Fig. 7d.

Figure 2 for reviewers (included in the revised manuscript). HTT-polyQ119 inclusion size. The HTT-polyQ119 inclusion size was analyzed after 24 h of TTC induction in the inducible HTT-polyQ119 HEK cells in the experiments shown in the original Fig. 7b, c.

3. All knock-down experiments are on base subunits of the 19S RP; the authors may wish to knock-down lid subunits to show whether the observed effect is limited to the ATPase subunits. The authors may cite Braun et al., NSMB 1999 which reported the 19S base with putative chaperone activity.

Thank you for the suggested study, which we have now cited in our revised discussion.

As requested, we have tested the effect of depleting the other 19S and 20S subunits. To do so, we screened all the subunits of the proteasome and measured the effect of their depletion on the dualPIM puncta size. The data show that the individual depletion of most of the 19S RP subunits (independently whether they are in the base or lid) increases dualPIM puncta size (indicative for impaired fragmentation). In contrast, the individual knockdown of the 20S CP subunits does not or only marginally alter the puncta size (Fig. 3a for reviewers). Importantly, measurement of the proteasome degradation activity using the Ub-R-GFP reporter cell line (Dantuma et al, Nat Biotechnol, 2000, 18:538) showed that the knockdown of most of the 20S CP subunits did block the proteasome, highlighting their functional depletion and underscoring that proteasome inhibition per se is not linked to impaired dualPIM aggregate fragmentation (Fig. 3b for reviewers). These results confirm that the 19S RP but not the 20S CP, is required for the

Figure 3 for reviewers (included in the revised manuscript). Proteasomal subunit screen. (a) DualPIM puncta size was measured upon knockdown of the indicated siRNAs after 19 h of aggregate induction in dualPIM cells. **(b)** GFP-positive cells were measured after knockdown of the indicated siRNAs in the Ub-R-GFP reporter cells.

fragmentation of the dualPIM aggregates. They also show that not only the ATPases might be required for this new function of the 19S RP. We have therefore adjusted the following statement in the manuscript: "Altogether, these data suggest that the 19S RP and more specifically its ATPases are required for the fragmentation of dualPIM aggregates." into "Altogether, these data show that the 19S RP is required for the fragmentation of the dualPIM inclusions and suggest that its ATPase subunits could play a key role in this function." These data are included in the manuscript as Fig. 3a and b.

4. I fail to appreciate the experiment that demonstrate aggregate condensation, as claimed in Figure 7, please clarify how aggregate condensation with fragmentation was demonstrated.

The 3D-CLET ultrastructural experiments shown in Fig. 4a-c and Extended Data Fig. 6a-c demonstrate that in contrast to WT or SAR-405-treated cells, depletion of either DNAJB6 or PSMC5 leads to dualPIM inclusions that are less compact (condensed). Since the depletion of these two proteins also blocks aggregate fragmentation, we infer that this increase in compactness/condensation is associated with fragmentation. We have indicated this in the discussion at page 28, line 596 of our manuscript. We have now also provided a more precise explanation of the dualPIM structures and their terminology at page 7, line 115-118 and page 9, line 164-165 in our revised manuscript, and in the Methods section on page 34 (please also see our response to # 11 of this reviewer). FRAP analyses, however, show that depletion of DNAJB6

Figure 4 for reviewers (included in the revised manuscript): FRAP measurement of the dualPIM aggregates. DualPIM cells were depleted of DNAJB6 or PSMC5 and dualPIM inclusions were bleached (white arrow). Fluorescent recovery of the bleached area was measured for 13 min (80 images in 10 s intervals).

or PSMC5 does not alter the biophysical properties within the dualPIM inclusions, i.e., they remain immobile (Fig. 4 for reviewers). These data are included in the manuscript as Extended Data Fig. 7e, f.

5. It is unclear how cellular proteasome and lysosomal activities are affected by the different knock-down experiments. Especially with proteasome knock-down experiments, it may be worthwhile to demonstrate the integrity by native gel assessment of proteasomes (e.g. Elsasser et al. Methods Enzymol. 2005). The authors should also consider whether other 20S CP subunits should be knocked down, as it is possible to replace PSMB2 and PSMB5 with alternative subunits.

We have now performed a LysoTrackerRed staining to test whether the number and function of lysosomes are affected by the knockdown of the proteasomal subunits or DNAJB6 (Fig. 5 for reviewers). This experiment shows that the number of LysoTracker-positive puncta per cell does not significantly decrease in any of the tested conditions. These data are included in the manuscript as Extended Data Fig. 5f.

Figure 5 for reviewers (included in the revised manuscript): Lysosomal activity is not altered by proteasomal depletion. Lysosomal activity was assessed in uninduced dualPIM cells depleted of either DNAJB6, PSMB2, PSMB5, PSMC1 or PSMC5, using lysotracker staining.

Furthermore, we have now tested the relevance of all the 20S CP subunits in the fragmentation of the dualPIM aggregates (see our reply to comment #3 of this reviewer). Moreover, and as suggested, we have also assessed the proteasomes by native gel (Fig. 6 for reviewers). While the abundance of free 19S alone and in the 26S fractions are strongly reduced upon depletion of its subunit PSMC1 or PSMC5 (Fig. 6, upper panel), the amount of 20S remains unaffected (Fig. 6,

Figure 6 for reviewers (included in the revised manuscript): Proteasome analysis by native gel electrophoresis. DualPIM cells were depleted of PSMC1, PSMC5, PSMB2 or PSMB5, or treated with 100 nM BTZ, before to separate proteasomes by native gel electrophoresis. Gels were decorated with either anti-PSMA6 or anti-PSMC2 antibodies to detect the 20 CP and the 19S RP, respectively. Proteasomal activity was measured using the ABP4 probe.

middle panel). Conversely, knockdown of the 20S subunits PSMB2 or PSMB5 reduces the proteasomal particles compared to the controls (middle panel in the figure) but does not abolish the complete proteasome (26S fractions) and 19S particles as seen for PSMC1 and PSMC5 depletion. Importantly, when measuring in gel proteolytic activity (Schipper-Krom et al, *Front Mol Biosci*, 2019, 6:56), the 20S activity is reduced by PSMB2 or PSMB5 depletion but not upon PSMC1 or PSMC5 knockdown (Fig. 6, lower panel), whilst the activity of the complexes is reduced by both. These results confirm that depletion of the 19S and 20S subunits was specific and effective, and that proteasomal degradation capacity per se is not involved in dualPIM inclusion fragmentation. These data are included in the manuscript as *Extended Data Fig. 5e*.

6. It is unclear whether 19S RP-dependent aggregate fragmentation is a direct effect or if it may be mediated through other paths. The authors may consider to co-stain to determine whether the 19S-mediated effect is directly related to free 19S RPs or if the proteasome holoenzyme, which is the most prominent form, is involved. Proteasome-mediated fragmentation/reduction of aggregate size in cells has been observed before (Morten et al., *PNAS* 2022) and should be considered for comparative discussions.

Morten and co-workers indeed showed that alpha-synuclein aggregates entering cells are surrounded by proteasomes and that this correlates with a reduction in the size of the aggregates over time, although this was not directly demonstrated. Nonetheless, these circumstantial observations indeed support our findings, and they are now mentioned in the discussion at page 27 and line 560-562, of the revised manuscript.

To perform the requested analyses, we generated a cell line expressing mCherry-tagged PIM and examined the colocalization of the dualPIM puncta with the 19S RP (PSMC5) or the 20S CP (PSMA7) (Fig. 7 for reviewers). The quantification of this experiment revealed that while around 25% of all PIM puncta are positive for the 19S component (white arrow heads), only 10% of them are also positive for 20S (purple arrow heads). This suggests that a significant proportion of the

Figure 7 for reviewers (not included in the revised manuscript): 19S and 20S distribution onto mCherryPIM puncta. PIM aggregates were induced in mCherryPIM cells for 5 h and the degree of 19S⁺ and 19S⁺/20S⁺ mCherryPIM structures was determined. Red arrows indicate mCherry only; white arrows indicate 19S⁺ structures; purple arrows indicate 19S⁺/20S⁺ structures.

dualPIM puncta are 19S positive only. However, we cannot firmly conclude based on these experiments that the 19S is individually responsible for the fragmentation. To firmly state this, in vitro experiments with extracted 19S would be required and this is beyond the scope of this study. We therefore have not included these data in this manuscript.

7. Proteasome nomenclature should be consistent for the wider readership, please use either PSMC1/PSMC2 etc. or human Rpt2/Rpt3 etc... and either PSMD14, ADRM1 or human Rpn11, Rpn13. It may be worthwhile to indicate the commonly used nomenclatures on first mention e.g. PMSB5/β5 subunit.

Thank you for this suggestion. We have now indicated all the terms commonly used for the proteasomal subunits when first mentioned.

8. The dualPIM system is a specialist reporter system developed by the authors. It may be useful

to describe this system (e.g. that it's based on FKBP, related to AgDD etc.) as I had to investigate through citations to appreciate this.

We have provided a brief description of the multimerization principle of the dualPIM construct at page 12, line 112, of the revised manuscript and indicated that it is similar to the AgDD system.

Other notes:

9. The use of siRNA for knock-downs raises concerns about incomplete depletion and potential compensatory mechanisms. The authors might consider addressing this limitation in their discussion.

Our Western blot analyses (Fig. 3 and 6, and Extended Data Fig. 2) showed an effective depletion of the key proteins targeted by the siRNAs. Moreover, several (new) experiments show that the depletion leads to the related functional blocks (e.g., Fig. 3b, Extended Data Fig. 5d,e). However, we cannot indeed exclude that some of the proteins targeted in the screens (Fig. 1g-i and Fig. 3a) have a residual activity sufficient to in part sustain their cellular function. Therefore, we have indicated this limitation at page 10, line 194-196 of the manuscript.

Of note, the use of siRNAs reduces the probability of inducing compensatory mechanisms, as may happen using knockout approaches.

10. Figure legends slightly repetitive and reads like sentences from Materials and Methods sometimes. It may also be helpful to move control/validation experiments (e.g. of siRNA) to Supplementary Information.

We have removed the repetitive sentences from the figure legends, leaving them only in the Methods section, where we provided complete experimental details (e.g., how dualPIM aggregate induction was achieved). As requested, we also moved the control knockdown experiments from Fig. 3 and Fig. 6 to the supplemental material.

11. Additionally, a more detailed definition of the amorphous aggregates used in this study would be helpful for readers to better understand the nature of the protein assemblies being investigated.

A clear definition of amorphous aggregates has now been given in the introduction, page 3, at line 49, of the manuscript. Moreover, we have refined our terminology and definitions of the dualPIM

structures, by calling them aggregates or inclusions depending on their morphology. Accordingly, we have adjusted our model (Fig. 7g). In brief, amorphous dualPIM aggregates are formed upon addition of rapa2 and when the fragmentation machinery is blocked, these aggregates coalesce into bigger inclusions. Please also see our rebuttal to the comment #2 for reviewer 2 for the details.

Reviewer #2

The manuscript by Mauthe et al. attempts to answer the question of how large aggregates can be degraded by the lysosomal degradation pathway, in particular by autophagy, given the apparent size limitation of the latter. As a model for aggregation, they build on their previous research in which artificial aggregates produced by Particles Induced by Multimerization (PIMs) were shown to be degraded by autophagy and could be tracked with a dual GFP-mCherry tag that quenches GFP in the lysosome. Using this assay, they show that in this system, small pieces of aggregates are pinched off from large aggregates that are subsequently delivered to the lysosome.

This clearance required the HSP70/HSP110/DNAJB6 machinery and/or the proteasomal 19S RP. Interestingly, the core particle of the proteasome was not required for aggregate degradation, suggesting a function independent of the canonical proteasome role. When DNAJB6 or 19S RP was depleted, the aggregates were less compact as shown by CLEM and degraded less. Clearance is dependent on autophagy because when autophagy was blocked, smaller aggregates still formed but were not degraded. It is concluded that DNAJB6 and/or 19S RP form a fragmentation machinery for PIM aggregates, followed by autophagic degradation. Deletion of DNAJB6 or 19S RP impaired the assembly of the autophagic machinery at the aggregate surface.

In the last part, they extend this analysis to other aggregates. They show that the colocalization of p62 with UB in puro-induced aggregates depends on the HSP70/110/DNAJB6, and/or 19S RP machinery. Similarly for Htt, they show that deletion of chaperones increases Htt aggregation (previously shown) and that the 19S PR has a similar effect.

In summary, the study by Mauthe et al. addresses a very important question of how aggregates are

removed by autophagy. Throughout the manuscript, the authors claim to have identified an unknown fragmentation machinery, consisting of the HSP70/110/DNAJB6, and/or 19S RP machinery, that acts directly on aggregates to fragment them into small digestible pieces for autophagic clearance. However, whether this is a direct or indirect effect is poorly characterized and not very convincing at this stage. Both chaperons and proteasomes are known components of the proteostasis network responsible for the removal and disaggregation of protein aggregates. Impairment of these networks has been associated with accumulation of protein aggregates established by many groups. Therefore their involvement is not surprising. Surprising, however, is that only the 19S RP and not the full proteasome is important for the subsequent autophagic removal of aggregates. However, mechanistic insights into the action of the 19S are lacking. Also lacking are any mechanistic insights into how the HSP70/110/DNAJB6 fragment these aggregates into larger junks distinct from normal disaggregates as previously proposed. Overall, the authors provide interesting insights into the degradation of protein aggregates by autophagy, but at the moment most conclusions are not convincingly supported by the experimental data as described below and therefore it is not clear if the proposed fragmentation machinery exist as proposed.

We agree with this reviewer that all the components that we studied have individually been implicated in handling aggregation-prone proteins or the aggregates that they form. However, what we show for the first time is that these 3 components, the HSP70-HSP110-DNAJB6 module, the 19S RP of the proteasome and autophagy, work in concert to remove aggregated material. In the case of DNAJB6 and 19S RP, we assign a new cellular function to these factors. Our data are also the first direct demonstration of a piece meal process (often suggested but never demonstrated) of large aggregates and inclusions before their autolysosomal clearance.

We also realize that the terminology and definitions of amyloids, amorphous aggregates and inclusions, as well as terms like disaggregation versus fragmentation may have created some confusion. We have revised our manuscript carefully so that all the concepts are clearly formulated, also guided by some of this and other reviewers' comments. In particular, we have refined our terminology and definitions of the dualPIM structures, by calling them aggregates or inclusions depending on their size and morphology. Please also see our rebuttal to the comment #2 for this reviewer for the details.

Major concerns:

1. Figure 1 is very exciting because it shows the piecemeal degradation of aggregates over time. Throughout the rest of the manuscript, the authors use endpoint imaging of aggregates to measure their average size. This is very indirect and could result from an increased aggregation rate due to general protein aggregation in proteasome and chaperone mutants. To measure fragmentation more directly, the authors should follow individual aggregates in time-lapse microscopy experiments to measure their size changes over time.

Throughout our whole manuscript, we follow the dualPIM aggregation over time by time-lapse imaging using the Incucyte system (Fig. 1, 3 and Extended Data Figures 1, 2, 4, 5, 6 and 8). From these data, we calculated the degradation rate (slope) from the degradation curves obtained by measuring the decrease in the yellow signal (e.g., see Fig. 1, Extended Data Fig. 1). Within the same datasets, we also determine the average size of the dualPIM puncta at each timepoint. However, we decided to only show the differences in size at the end timepoint (Fig. 1, 3 and Extended Data Figures 1, 2, 4, 5, 6 and 8). In our response to comment #8 of this reviewer, we now also provide time-lapse images (from the Incucyte system) used for the calculations, exemplifying how dualPIM aggregates gets fragmented under autophagy inhibition. Moreover, we are showing an example of how the dualPIM inclusions first get fragmented into smaller pieces before they are transported into lysosomes in video 2, which was already included in the initial submission.

Please also see our response to the comment #2 of this reviewer, in which we provide evidence showing that the aggregate formation rate is not altered by 19S RP or DNAJB6 depletion.

2. Figure 4g: After short times of aggregate formation (compared to long times in Figure 1g). Why are the aggregates more diluted for the dual PIM signal? How can it be excluded that general protein aggregation leads to diluted PIM aggregates that completely change the identity of the aggregate itself, similar to the different packing of aggregates seen in the EM experiments?

None of our pharmacological or genetic manipulations was found to affect the initiation of dualPIM aggregation in the first 1-2 h upon addition of rapa2. All interventions only affect the processes that follows. The dualPIM aggregates do get smaller but are not degraded when autophagy is inhibited (e.g., SAR405 treatment). As shown in Fig. 8 for reviewers, the size of the aggregates does not decrease and several of aggregates coalesce into bigger inclusions over time only when specific factors, e.g., DNAJB6 or PSMC5, are depleted. Based on these data, we have revised our terminology. In brief, amorphous dualPIM aggregates are formed upon addition of rapa2 and when the fragmentation machinery is blocked, these aggregates coalesce into bigger inclusions. Accordingly, we have adjusted our model (Fig. 7g).

Figure 8 for reviewers (not included in the revised manuscript). DualPIM aggregates visualization in DNAJB6 or PSMC5 knockdown cells over time. DualPIM cells were transfected with the indicated siRNAs for 48 h, TTC-treated for 24 h and incubated with 500 nM rapalog2 for the entire duration of the experiment. We performed live-cell microscopy and acquired images every 15 min, starting 15 min prior to rapa2 addition using the Zeiss Cell Discoverer 7. In contrast to siCtr cells, when DNAJB6 or PSMC5 are depleted, the puncta become bigger (white arrowheads) and this appears to be through the coalescence of the small aggregates into big inclusions. Importantly, this analysis also clearly shows that the dualPIM construct does not aggregate without rapa2 addition upon DNAJB6 or PSMC5 depletion.

3. A key finding is that depletion of proteasome core particles does not affect autophagic degradation of PIM and other aggregates. However, these experiments should be complemented with additional controls to assess proteasome activity. The data in Ext. Data Figure 5c are only

partially convincing, as for example PSMB5 has the weakest effect compared to the others, a rather unexpected finding. It is therefore possible that the seen effects correlate with the residual activity. It is standard practice to use model substrates such as ZsProSensor-1 (TaKaRa) to monitor degradation in these mutants.

Please see our response to comment #3 of reviewer #1. In short, we now knocked down all the subunits of the proteasome and measured the effect of their depletion on the dualPIM aggregate size. The data clearly show that the individual depletion of most of the 19S RP subunits (independently whether they are in the base or lid) impairs fragmentation. In contrast, the individual knockdown of the 20S CP subunits does not, even though it did affect proteasomal degradation of the Ub-R-GFP reporter, highlighting their functional depletion and underscoring that proteasome inhibition per se is not linked to impaired dualPIM aggregate fragmentation. These data are included in the manuscript as Fig. 3a, b.

4. In line, complete inhibition of the proteasome by BTZ, which inhibits the catalytic activity of the 20S core particle, showed the opposite effect as the depletion of core particle subunits resulted in large protein aggregates. This is in stark contrast to the depletion of 20S core subunits, which resulted in a strong fragmentation phenotype. Similar experiments as shown in 1f should be performed with the proteasome inhibitor BTZ. In general, the BTZ should be used to impair proteasome inactivation throughout the manuscript. As a note from the Materials and Methods section, it is unclear how the slopes of the curves obtained in these experiments were calculated.

We agree with this reviewer that these observations may be somewhat puzzling. Therefore, we performed a BTZ titration experiment in our dualPIM and Ub-R-GFP cells. The proteasome activity is completely blocked with 25 nM BTZ (Fig. 9a for reviewers). Interestingly, no effect on the dualPIM puncta size was observed up to this concentration (Fig.9b). However, we detected effects on dualPIM puncta size at higher concentrations such as 50 and 100 nM, (Fig.9 b). This result suggests that at high concentrations, BTZ does not only block the 20S CP, but may also affect the 19S RP. This notion is also suggested by data from Tsvetkov and co-workers showing that cells in which the number of 19S RP is reduced, become resistant to BTZ or MG132 treatment (Tsvetkov et al, eLife, 2015, 4:e08467). These observations are fully in line with our new siRNA data and further underscores the different relevance of the 19S RP and 20S CP for aggregate fragmentation. These data are included in the manuscript as Extended Data Fig. 4a, b.

As requested, we show the BTZ analysis in Fig. 10 for the reviewers like in Fig. 1f, and we now better explain how the curves have been calculated in the Methods.

Figure 10 for reviewers (not included in the revised manuscript). 100 nM BTZ blocks dualPIM aggregate degradation and fragmentation. a, TTC-treated dualPIM cells were incubated with 500 nM rapalog2 plus or minus 100 nM BTZ for 19 h and imaged over time with the Incucyte system. **b,** Quantification of dualPIM puncta size at 19 h post-rapalog2 treatment. **c,** The degradation curves and rates of the experiment illustrated in panel **a** are presented.

5. If only the 19S RP is essential, the authors should provide experimental evidence for the statement in lines 289-290 that the ATPase function is required for fragmentation or remove the statement.

This reviewer is correct that we may have somewhat overstated this. After inclusion of the experiments in which we individually knocked down all proteasomal subunits, we now restrict our conclusion to the statement that “Altogether, these data show that the 19S RP is required for the fragmentation of the dualPIM inclusions and suggest that its ATPase subunits could play a key role in this function.”

6. In line 287 – 288 the authors state “While these treatments led to a proteasomal impairment (Extended Data Fig. 5h), the fragmentation of the dualPIM aggregate was unaffected, i.e., the dualPIM aggregates were small.” Inhibition of DUBs blocks deubiquitylation but does not necessarily inhibit proteasome function as the authors suggest. They should test degradation of model proteasome substrates to assess activity in these experiments.

We agree and therefore we have now tested these drugs using the Ub-R-GFP reporter cell line and found that all of them block proteasomal degradation. We have included these data in the manuscript as new Extended Data Fig. 6a, b. See also our reply to comment #23 of this reviewer.

7. In Figure 5, the authors argue that autophagy receptors are less clustered on the aggregate surface. This could simply be due to the increased surface area of the aggregates themselves. The authors should compare the clustering with respect to the aggregate surface to rule out dilution effects.

As shown in Fig. 4h, the dualPIM inclusions are approximately 5 and 8-fold larger upon depletion of siDNAJB6 and siPSMC5, respectively, 5 h after their induction. As a control, we have also measured the dualPIM puncta size and obtained very similar numbers (Fig. 11 for reviewers). This means that indeed the increase in inclusion size is higher than the decrease in SAR intensity, which normally varies between 1.5- and 2-fold (Fig. 5a-d). These values support the notion that the enhanced surface of the inclusions probably impair the clustering of the SARs. This is exactly what we conclude and propose with our model on how the fragmentation and condensation of the aggregates contributes to their autophagic targeting. In particular, as described in the manuscript, our data show that the recruitment of SARs and the ATG machinery alone is not sufficient to initiate phagophore formation nearby aggregates. The action of DNAJB6 and the 19S RP is needed to increase the compactness of the aggregates, which also leads to SAR clustering and phagophore initiation. This concept has been proposed previously for other forms of selective autophagy (Hollenstein et al, Nat Commun, 2021, 12;7194; Gonzalez et al, Nature, 2023, 618:394) and it has been very recently confirmed by two other ones (Eickhorst et al, J Mol Biol, 2024, 436:168631; Licheva et al, 2025, Nat Cell Biol, s41556-024-01572-y), which we have now cited as well.

Figure 11 for reviewers (not included in the revised manuscript). Measurement of the average dualPIM puncta size. The average dualPIM puncta size relative to the siCtrl condition was determined for each IF experiment shown in Fig. 5a-d.

8. Since the authors suggest that fragmentation is upstream of the autophagy machinery, the authors should show by live cell microscopy in time-lapse experiments that large aggregates fragment over time into smaller ones that accumulate due to blocking autophagic degradation (e.g., by depletion of FIP200 or p62). This should lead to a concomitant increase in the number of small aggregates.

We already showed (Fig. 2f) that the dualPIM aggregates remain yellow (due to a block in lysosomal degradation) under autophagy inhibition condition (i.e., BAF) but they do become smaller (because the fragmentation machinery is still active).

We have now also generated a FIP200KO dualPIM cell line and followed dualPIM aggregates within cells treated with BTZ to accumulate large PIM inclusions. After BTZ washout, the size of the inclusions significantly decreases, while their number increases, but they remain yellow as they are not degraded in the lysosome due to the FIP200 knockout (Fig. 12 for reviewers). These data are now included in the manuscript as Extended Data Fig. 8h.

Figure 12 for reviewers (included in the revised manuscript): Fragmentation occurs upstream of the autophagy machinery. FIP200KO dualPIM cells were pre-treated with 100 nM BTZ for 6 h, and dualPIM puncta number and size was analyzed before and after the washout period.

9. In Figures 6 and 7, the authors attempt to extend their findings to other described protein aggregates. While they are able to show that HSP70/110/DNAJB6 and/or 19S have an effect on aggregate size, they do not provide experiments that really confirm the fragmentation of aggregates. It is essential to show that these aggregates are indeed degraded in a piecemeal fashion. Similar to point 1, the authors should try to follow the size of individual aggregates over time.

To show the fragmentation of the puromycin-induced aggregate, we generated a stable, inducible mCherry-GFP-p62-expressing U2OS cell line. We first validated whether the mCherry-GFP-p62 puncta behave as endogenous p62 puncta, which are not degraded when cells were treated with

SAR405 or BAF, BTZ or VER (Fig. 13a for reviewers). As for the dualPIM aggregates, we observed that the mCherry-GFP-p62-positive aggregates undergo fragmentation upon puromycin removal (Fig. 13b for reviewers, white arrows). Furthermore, we have now also investigated the puromycin-induced aggregate structures using TEM (panel c). In line with the IF data and similar to the dualPIM aggregates, the puromycin-induced aggregates after puromycin treatment are much larger than the ones that we detect in the cytoplasm upon treatment with SAR405. This result confirms that these aggregates also undergo fragmentation before degradation. As for the dualPIM aggregates, this fragmentation can be blocked by high concentration of BTZ. These data

are now included as Fig. 6e and Extended Data Fig. 9d. Please also see our reply to comment #10 of this reviewer for the fragmentation of the HTT-polyQ inclusions.

Figure 13 for reviewers (included in the revised manuscript): Puromycin induced aggregates are fragmented. mCherry-GFP-p62 cells were pre-treated with puromycin for 4 h and followed during the washout period. **a**, Representative images of cells treated with CM, 100 nM BAF, 10 μ M SAR, 100 nM BTZ or 40 μ M VER after the washout period (11h post washout) are shown. **b**, Fragmentation events (white arrows) are indicated in a mCherry- GFP-p62 expressing cell. **c**, TEM analysis of U2OS cells left untreated, treated with 5 μ g/ml puromycin (puro) for 2 h or analyzed after 3 h washout in the presence or absence of 10 μ M SAR405, or 100 nM BTZ.

10. Figure 7 appears to be very preliminary. For the disease-relevant polyQ aggregate, no experiments dealing with fragmentation or autophagic turnover are presented. Often the standard deviation is very high, questioning the significance of the experiments presented (Figure 7b).

We agree with this comment and indeed our data for this are indirect (i.e., only showing that same components required for the fragmentation and turnover over the dualPIM and puromycin-induced aggregates also play a role in reducing the formation of HTT amyloids). In fact, when they are at the amyloid stage, HTT-polyQ (amyloid) are no longer an autophagy substrate (Zhao et al, Mol Cell, 2024, 84:1980). HTT-polyQ proteins form condensates before becoming amyloids (Peskest et al, Mol Cell, 2018, 84:588) and subsequently they start to form fibrils from the inner core of these condensates, with the rims remaining fluid or gel-like (Zhao et al, Mol Cell, 2024, 84:1980). Various lines of evidence, including those of our own lab, have shown that DNAJB6 acts before the amyloid stage (also with other substrates) (Kuiper et al, Nat Cell Biol, 2022, 24:1584). Actually, DNAJB6 co-condenses with HTT preventing their transition into amyloid inclusions (Mattos et al, <https://www.biorxiv.org/content/10.1101/2022.08.23.504914v1>). Along this line, we have confirmed that DNAJB6 and also PSMC5 are indeed localizing/co-condensate with the HTT-polyQ119 inclusions in our system (Fig. 14 for reviewer). Therefore, our data suggest a scenario in which pre-amyloid structures surrounding the fibril GFP-HTT-polyQ core can be fragmented, which may lead to the generation of residual, fragmentation-resistant HTT-polyQ119 amyloids. These data are now included as Extended Data Fig. 10d.

Figure 14 for reviewers (included in the revised manuscript): HTT-polyQ119 inclusions are positive for PSMC5 and DNBAJB6. HTT-polyQ119 inclusions were induced for 24 h with TTC and cells were stained with anti-DNAJB6 or anti-PSMC5 antibodies.

In line with such a scenario, we find that the size of the HTT-polyQ119 inclusions increases when autophagy is blocked or when the 19S RP is depleted, but not when the 20S CP is knockdown (Fig. 2 and 13 for reviewers). Moreover, after tetracycline-mediated induction of HTT-polyQ119 aggregation, we observe recruitment of the SARs NDP52 and TAX1BP1 (Fig. 15a for reviewers). Depletion of PSMC5 decreases the recruitment of these two SARs. Finally, the average HTT-polyQ119 inclusion size decreases over time upon washout of tetracycline after 24 h. We attribute this relatively minimal size reduction to the fact that only the outer rim of the inclusions is targeted. Again, depletion of PSMC5 prevents this size reduction (Fig. 15b for reviewers). However, we do not observe neither the reduced recruitment of the SARs to HTT-polyQ119 inclusions, nor the block in aggregate size reduction after washout in DNAJB6-depleted cells. These very interesting and compelling data suggest that both DNAJB6 and PSMC5 play a role in the pre-amyloid HTT inclusion removal (as showed in our original Fig. 7), but apparently only PSMC5 (and the 19S RP) is also involved in the degradation of the amorphous ring around fibrillar HTT-polyQ119 core. Although these data are very intriguing, experiments beyond the scope of this study are required to firmly confirm this scenario. We therefore would like to not include these data in the manuscript and just indicate that for polyQ-HTT our data are indirect, albeit supported by additional data now (see response to comment #11 of this reviewer) to suggest a comparable requirement of protein quality control components for clearance as for PIM or puromycin-induced aggregation, but that for polyQ inclusions the situation is more

Figure 15 for reviewers (not included in the revised manuscript): HTT-polyQ119 inclusion processing in the absence of PSMC5 and DNBAJB6. HTT-polyQ119 inclusion was induced for 24 h with TTC after depletion of DNAJB6 or PSMC5 before to decorate cells with anti-NDP52 (a) or anti-TAX1BP1 (b). HTT-polyQ119 inclusions were induced for 24 h with TCC in cells depleted for either DNAJB6 or PSMC5 and aggregate size was monitored during a 26 h TTC washout period after initial induction (c) and the change in inclusion size was measured (d).

complex due to their structures.

We agree with this reviewer that the standard deviations seem relatively high in the cases of *siPSMC1* and *siPSMC5*, but they are statistically significant. The large error bars are mainly due to variation in the absolute aggregation efficiency in independent experimental repeats rather than an inconsistency in the effects of *PSMC1* or *PSMC5* knockdown (see Fig. 16 for the reviewers, for the independent experiments used for the statistical evaluation).

Figure 16 for reviewers (not included in the revised manuscript). HTT-polyQ119 aggregation upon 19S subunit depletion. The quantification of Fig. 7b is shown with standard error (SE) and not SD (as in the manuscript) as well as the individual biological replicates for the siCtr, siPSMC1 and siPSMC5 conditions from the experiment depicted in Fig. 7b.

11. Figure 7d and e: In this case, all subunits of the proteasome, including the core particle subunits, have a significant effect on the aggregation of polyQ119 compared to the control. However, in Figure 7c, there is no obvious difference in polyQ119 aggregates.

The reviewer is correct, but the formation kinetics of HTT-polyQ119 inclusions (over 24-28 h) is rather different than the one of the dual PIM aggregates (within 15 min to 2 h). Inhibition of the canonical proteasome activity is known to enhance HTT-polyQ aggregation but these effects may be due to multiple causes, including the degradation of monomeric HTT species and the disturbance of protein homeostasis caused by affecting the turnover of other substrates that sequester chaperones away from the fragmentase activity. When we knocked down individually all the proteasomal subunits (Fig. 17 for reviewers), we indeed found that depletion of some of the

20S CP subunits moderately increases HTT-polyQ119 aggregation (Fig. 17a). This effect, however, was much more pronounced when 19S RP subunits were knocked down. Moreover, while depletion of the 20S CP subunits did not increase the aggregate size, the knockdown of the 19S RP ones did (Fig. 17b), highlighting the importance of the fragmentase activity also in delaying the accumulation of HTT inclusions. These data are now included as Extended Data Fig. 10e, f.

Figure 17 for reviewers (included in the revised manuscript). HTT-polyQ119 aggregation upon proteasomal subunit depletion. (a) HTT-polyQ119 inclusion amount (expressed as AUC) (a) and the average size of the individual inclusions (b) upon treatment with the indicated siRNAs and after 24 h of polyQ induction.

Minor concerns:

12. Line 135 – 136 page 5 (Extended Data Fig. 1i). Extended Data Fig. 1i does not exist. In contrast Figure e exists twice.

We apologize: the panels were mislabeled. This has been corrected (note: previous Extended Data Fig. 1h is now Extended Data Fig. 1i).

13. Line 149 (Fig. 1F) should be (Fig. 1f) -> Figure 1f does not support the statement of lysosomal degradation should be double cited with Ext. Data Fig. 1f

Thanks: corrected as suggested.

14. line 159: “While BAF treatment did not increase dualPIM aggregate size (but blocked dualPIM turnover) (Fig. 1h, i),...” this sentence is misleading as it has an effect on aggregate size – it actually decreases it.

Thanks: we changed the sentence into “While BAF treatment did not increase but rather decreased dualPIM aggregate size (but blocked dualPIM turnover)...”

15. Extended Data Fig. 1e (fluorescence images) and 1g does not correlate. In the images SAR405 shows the largest aggregates to the very small ones in Noco treated cells. It should made clear what is actually measured in these experiments.

As indicated in the y-axis of graph presented in Fig. 1g (and also in the other graphs measuring the lysosomal delivery of the dualPIM aggregates over time), we determined the percentage of yellow puncta, i.e., those not delivered into lysosomes.

16. From the legend it is not clear what Extended Data Fig. 1h is actually showing and for which experiment this is the control. I assume Figure 1g.

We apologize: the panels in Extended Data Fig. 1 were mislabeled. This has now been fixed and the citations in the text correspond to the right panels.

17. Figure 2A should be marked that DNAJB6 is overexpressed as it clearly leads to reduction of aggregates pin general

This is now indicated in the figure, figure legend and text at page 11, line 208.

18. Figure 2c lacks a negative control like +BAF. More importantly why does WT DNAJB6 reconstitution not rescue to the full extend as WT, particularly as no aggregates where detected see Figure 2b.

We have now added the requested BAF control to the revised Fig. 2c.

In Fig. 2 b, we exclusively measured the aggregates in V5-DNAJB6-expressing cells, which can be identified by IF. As shown in Fig. 18 for reviewers, not all the cells were transfected and only V5-DNAJB6-expressing cells are protected from aggregation.

In the WB approach (Figure 2C), the analyzed sample is a mixed population of both transfected and non-transfected cells, and this is why there is not a full rescue for this endpoint.

Figure 18 for reviewers (not included in the revised manuscript). DNAJB6 is required to remove dualPIM aggregates. Additional image from the dataset presented in Fig. 2b. The selected image shows dualPIM cells treated with the siRNA targeting DNAJB6 that we are transfected or not with the plasmid overexpressing WT-DNAJB6 (V5). The white arrow highlights a cell that is not transfected with the V5-DNAJB6-expressing plasmid and thus accumulates cytoplasmic dualPIM inclusions because of the siRNA-based depletion of endogenous DNAJB6.

19. Figure 2h degradation rate of control cells is 20x lower for BTZ compared to normal PIM aggregates – why?

BTZ treatment and thus proteasomal inhibition blocks a multitude of cellular processes (see reply to comment #11 of this reviewer). We think that it takes some time for the cell to reach homeostasis again upon BTZ washout and this indirectly affect the flux of aggregophagy, especially the first hours upon BTZ removal.

20. Extended Fig. 2f. This Figure should show more clearly the overlap of the Lamp1 signal with mcherry only signal from the dualPIMs.

We have now provided a zoom and used green arrows to make the overlap between the LAMP1 and mCherry signals clearer.

21. Extended Figure 3c: The full length dualPIM is very low at the 24h timepoint after bafilomycin treatment. Does this argue for proteasomal degradation?

We have provided a better western blot image. We have also quantified the non-degraded full-length dualPIM in this experiment, and there are no significant differences between the 24 h+Baf

Figure 19 for reviewers (not included in the revised manuscript). Levels of non-degraded dualPIM full length. Quantification of the full length dualPIM probe levels in the experiment shown in Extended data Fig. 3c and normalized to the loading control.

sample and the other ones treated with Baf (Figure 19 for reviewers). These data indicate that there is no enhanced proteasomal degradation of the dualPIM aggregates under BAF conditions.

22. Extended data Fig4 representation of data is not intuitive. it should be labelled that the washout phase contains still different drugs.

Thanks: we have labelled the washout phase as suggested.

23. Extended Data Figure 5e lacks a control that the E1 enzyme is really inhibited. Particularly since there is still, also less, clear targeting of Ub-dependent autophagy receptors to the aggregates (Ext. Data Fig. 5f). A Westernblot as in Ext. Data Fig. 5h would clarify the extend of inhibition.

We have used the newly generated cells stably expressing the Ub-R-GFP reporter (see Method section) to measure the proteasomal inhibition upon treatment with 5 μM PYR-41 but also 1 μM CZM, 1 μM RA190 and 0.5 μM b-AP15. We show that the concentration used in our study indeed blocks proteasomal degradation (Fig. 20 for reviewers). We have included these data in Extended Data Fig. 6a, b.

Figure 20 for reviewers (included in the revised manuscript). Proteasomal inhibition by different drugs. Ub-R-GFP cells were treated with 1 μ M CZM, 1 μ M RA190, 0.5 μ M b-AP15 or 5 μ M PYR-41 and the percentage of the cell area positive for GFP was measured at 6 and 20 h post treatment.

24. Line 289 – 290: The authors should provide experimental evidence for the statement that the ATPase function is needed for fragmentation or remove the statement.

See our reply to comment #5 of this reviewer; the conclusion was rephrased.

25. Figure 4G: siDNAJB6 the contrasting of the GFP signal seems very strange and unnatural this should be checked and only the overall brightness contrast should be adjusted not the just one.

Contrasts have now been adjusted in all the panels.

26. Extended Data Figure 8b. What is the rationality that siPSMB5 but not siPSMB2 lead to an increase in Ub/p62-positive aggregates. Also why siPSMB5 shows now one of the strongest phenotypes. Significance should be indicated. In line why does KD in A549 cells shows the opposite?

The significance was added to those samples that show a difference with the control, and the ones treated siPSMB2 or siPSMB5 are not significantly different.

Extended Data Fig. 9c (originally Extended Data Fig. 8b) shows the number of Ub/p62-positive aggregates before and after puromycin treatment, indicating that depletion of the 20S CP subunits is causing cellular stress, which increases the baseline of Ub/p62 positive structures. In contrast, the analysis in A549 cells depicted in Extended Data Fig. 10 a-b (originally 8c-d) is identical to the one depicted in Fig. 6c-d for U2OS cells. The results in U2OS and A549 cells are thus consistent.

27. Video 6 is not working

We apologize for this; this has now been fixed.

Reviewer #3

The manuscript „A chaperone/proteasome-based fragmentation machinery essential for aggregate clearance“ by Mauthe et al. investigates cellular mechanisms of aggregate clearance (i.e. aggregate clearance by autophagy). They show that the 19S proteasomal regulatory particle and the chaperones DNAJB6, HSP70 and HSP110 are required for the delivery of aggregated proteins to the autophagy-lysosomal system.

Overall, the findings are very interesting and relevant to a wide audience, and the chemically inducible particle induction by multimerization (PIM) system they use is very elegant, but the data (although very extensive and of high quality) do not support all of the authors' claims. Therefore, the paper needs to be substantially revised.

We are pleased to read that the reviewer finds our data interesting and of high quality. We clearly feel there is some confusion about terminology that has raised some of this reviewer's concerns as we almost completely agree with his/her evaluation of our data. We have therefore refined our terminology and definitions of the dualPIM structures, by calling them aggregates or inclusions depending on their morphology. In brief, amorphous dualPIM aggregates are formed upon addition of rapa2 and when the fragmentation machinery is blocked, these aggregates coalesce into bigger inclusions. Accordingly, we have now also adjusted our model (Fig. 7g). Please also see comment #2 for reviewer 2, in which we show how the big dualPIM inclusions are formed when the fragmentation machinery is absent. We hope that we have clarified this point in both our replies and revised manuscript.

Specific points

1. The authors imply that DNAJB6 participates in aggregate disaggregation and acts mechanistically similar to DNAJB1. However, there is not a single assay that directly examines

the fragmentation of amorphous aggregates and in particular amyloid fibers. They only monitor the dynamics of fluorescent foci in cells. Upon KD of DNAJB6 these foci persist, no smaller foci are pinched off from the large foci, which would be directed to aggrephagy in the presence of DNAJB6. However, pinching off pieces from the foci does not necessarily mechanistically equate to the amorphous aggregate/amyloid fiber fragmentation mediated by the DNAJB1-HSP70-HSP110 disaggregation machinery. Such visible cellular foci contain a large number of individual aggregates/fibers brought into close proximity at these specific protein quality control regions in the cells for efficient refolding/degradation. The size of these foci does not necessarily correlate with the size of single aggregate particles within them. Therefore, the scale and nature of this type of ‘disaggregation’ is completely different from the disaggregation mediated by the DNAJB1-HSP70-HSP110 machinery.

To substantiate the claim that DNAJB6 is functionally similar to DNAJB1, the authors would need to show at least an effect on the size of individual aggregates in the cells, e.g. by using semidenaturing agarose gel electrophoresis. And even if DNAJB6 is shown to reduce aggregate size, it could just dissolve loosely or laterally associated fibers/aggregates. To demonstrate that DNAJB6 acts mechanistically similar to DNAJB1 would require an *in vitro* assay using purified components, showing that DNAJB6-HSP70-HSP110 cooperate and act specifically on aggregated substrates (also see below), reducing the size of pre-formed amorphous aggregates and shortening amyloid fibers. If such evidence cannot be provided, the authors must clarify that DNAJB6 is important for the dissociation of foci, but that this activity is not comparable to the disaggregation activity of the DNAJB1-HSP70-HSP110 system.

*We did not aim to conclude nor even would like to suggest that DNAJB6 acts mechanistically similar like DNAJB1. As in vitro data revealed, DNAJB1-HSP70-HSP110 machinery has aggregate disaggregation but also fragmentation activities, i.e., it can act **after aggregates have formed**. DNAJB6 has never been reported to have such an activity. Published in vitro data shows that DNAJB6 interacts early on in the aggregation process, not with polyQ monomers but with small oligomeric species and prevents the transition of its substrates into amyloids (Månsson et al, Cell Stress Chaperones, 2014, 19:227; Kakkar et al, Mol Cell, 2016, 62:272; Arosio et al, Nat Commun, 2016, 7:10948; Kuiper et al, Nat Cell Biol, 2022, 24:1584). In other words, it does not change aggregation per se (as we also see here for the dualPIM and puromycin-induced aggregates), but it changes the physicochemical properties of the dualPIM inclusions, allowing*

better access of HSP70 to aggregates, as already shown for other aggregates as well (Klaips et al, Nat Commun, 2020, 11:6271). We now reveal that this DNAJB6 function allows HSP70 (with assistance of HSP110 and in combination with the proteasomal 19S RP) fragmenting amorphous aggregates and inclusions into smaller pieces. Although the required (entropic) pulling activity of HSP70 for this fragmentation may be the same as the one for the DNAJB1-based amyloid fragmentation, DNAJB1 and DNAJB6 act on different types of substrates and in distinct manners, with the DNAJB6-based fragmentation also requiring the presence of 19S RP.

2. The experimental setup does not distinguish whether DNAJB6 acts during aggregate formation or whether it acts on pre-existing aggregates. DNAJB6 is depleted by RNAi before aggregates are formed, so it is not clear whether DNAJB6 acts during aggregate formation or on pre-existing aggregates.

To distinguish between these two possibilities, the authors would need to first induce aggregation and then KD DNAJB6 and other chaperones, etc.

The effect they describe here can also be explained by the fact that DNAJB6 is not present during aggregate formation. The authors have previously shown that DNAJB6 co-condensates with its misfolded substrates during aggregate formation. Foci could consist of such condensates. It is possible that this co-condensation with substrates is necessary to keep the aggregated proteins in a specific state that facilitates disaggregation by the DNAJB1-HSP70-HSP110 machinery and the recruitment autophagy receptors, which subsequently facilitates the pinching off of smaller parts from the large foci. DNAJB6 may therefore only indirectly contribute to the breaking down of foci into smaller units of aggregated proteins by HSP70 disaggregation machinery and the autophagy system.

This would be consistent with the view of some co-authors, stated in a manuscript on bioRxiv: ‘...suggesting that DNAJB6 prevents polyQ condensates to convert from the soluble to the solid state. This in turn, may keep the polyQ peptides competent for (regulated) degradation and accessible to factors allowing its extraction from the condensed state’. The data shown here support exactly this interpretation. The authors provide no evidence here that DNAJB6 is directly involved in the disaggregation.

As stated above, DNAJB6 does not work on pre-formed HTT-polyQ aggregates (Månsson et al, Cell Stress Chaperones, 2014, 19:227; Kakkar et al, Mol Cell, 2016, 62:272). We are also not claiming that DNAJB6 can do this here as carefully was stated in our original discussion: “Based on previous results (Kakkar et al, Mol Cell, 2016, 62:272; Yang, J. & Yang, X. Front Genet, 2020,

11;754), we speculate that the fragmentation machinery targets the HTT-polyQ entities that are not amyloids yet”, and “Therefore, the presence of chaperones and co-chaperones, particularly DNAJB6, during aggregate formation may not only affect the arrangements of aggregates into higher ordered structures (Nillegoda et al, Nature, 2015, 524:247; Kakkar et al, Mol Cell, 2016, 62:272; Hageman et al, Mol Cell, 2010, 37:355; Thiruvalluvan et al, Mol Cell, 2020, 78:346), but also determine how they are disaggregated or fragmented and then disposed”. Therefore, we agree with this reviewer that what we observed is possibly due to the presence of DNAJB6 already during aggregate formation (see also our reply to comment #1 of this reviewer). Whereas we do not exclude that DNAJB1 or another DNAJ protein may contribute to this fragmentation, we conclude that under our conditions, the basal levels of DNAJB6 are sufficient to allow fragmentation even when these other DNAJ proteins are depleted. Finally, using the DNAJB6^{H31Q} variant, we show in Fig. 2 that the direct interaction of DNAJB6 and HSP70 is needed to fragment and degrade the dualPIM aggregates. This result further supports the notion that the DNAJB6-HSP70-HSP110 module is not only sufficient but also required for the dualPIM aggregate turnover by aggrephagy.

3. According to the materials and methods, the authors used only siRNA targeting HSPA1A, an inducible Hsp70, but not HSPA8, the cognate Hsc70. This is surprising as it has been shown that disaggregation activity is mediated by Hsc70 and not Hsp70. The authors should comment on this discrepancy.

First of all, all cells tested express HSPA1A constitutively and both HSPA1A and HSPA8 are stress-inducible. It is also not entirely evident from the literature how much HSPA8 and HSPA1A are interchangeable for different substrates. Nonetheless, to address this reviewer’s question, we have now also depleted HSPA8 (Fig. 21 for reviewers) and although the knockdown was effective (panel d), the depletion of this chaperone did not increase the size of the dualPIM puncta (panel a) and had no effect on their lysosomal delivery (panels b and c). We would like to emphasize again that the fragmentation that we studied here is different from the published disaggregation by the DNAJB1 module, which acts on pre-formed amyloid species. That these fragmentations may require different HSP70 isoforms was (to us) rather surprising and evokes the necessity of a dedicated and detailed study on the functional diversification of the different HSP70 isoforms that, however, goes beyond the scope of the current manuscript.

Figure 21 for reviewers (not included in the revised manuscript). HSPA8 depletion has no effect on dualPIM fragmentation or degradation. DualPIM cells were transfected with the indicated siRNAs for 48 h, TTC-treated for 24 h and incubated with 500 nM rapalog2 for the entire duration of the experiment. **a**, Quantification of dualPIM puncta average size at 24 h post-rapalog2 treatment. **b,c**, DualPIM aggregate degradation curves (b) and rates (c). **d**, Knock-down efficiency of the used siRNA was assessed by western blot.

4. Every figure legend indicates that a two-tailed T-test was used. I want to believe that this is a copy-paste error. The authors should make sure they correctly state the statistical test they have used, and the t-test is not the correct one for most of their assays.

The T-test is used to compare the means between two groups. In our analyses, we always compare the mean of a specific treatment (drug or siRNA) with the control condition alone and not with other conditions, except in Extended Fig. 4, in which we used the ANOVA comparison. Nonetheless, we realized that we compared the siFIP200 condition to multiple other measurements in the original Extended Data Fig. 7g (now Extended Data Fig. 8g) and therefore we have corrected the statistical analysis of this experiment using the ANOVA comparison. For the rest of the graphs, we think the T-test is the appropriate way to determine statistically significant differences.

5. Extended Figure 5: If the authors want to claim that inhibition of VCP has no effect, they should use the typical concentration of the drug, which is 10 μM for U2OS cells (<https://www.abmole.com/products/nms-873.html>). The data differ significantly even when using

low concentrations (2.5 μM) of the inhibitor. I would assume that the effect is stronger when using the appropriate concentration.

We tested a higher concentration of the VCP inhibitor (i.e., 5 μM NMS-873) but dualPIM U2OS cells were dead after 14 h of treatment (Fig. 22a for reviewers). We therefore tested lower concentrations that allowed cells to survive for at least 24 h to complete our measurements and this is what we have used in our manuscript.

Looking at the analysis in which we used 5 μM NMS-873, no differences in the dualPIM puncta size between the control and the treated cells were detected at the 8 h time point (Fig. 22b for reviewers), when the cells showed only weak signs of stress (Fig. 22a for reviewers). However, their delivery to lysosomes was inhibited at this time point (Fig. 22c for reviewers), similarly to what we observed when treating the cells with 2.5 μM NMS-873, suggesting that this lower dose is effective yet less toxic (Extended Data Fig. 5a, b). For this reason, we would like to only show these latter data in our manuscript.

Figure 22 for reviewers (not included in the revised manuscript). VCP inhibition does not block dualPIM aggregate fragmentation. TTC-treated dualPIM cells were incubated with 500 nM rapalog2 plus or minus 5 μM NMS-873. **a**, Representative images of the untreated (0h) and treated cells after 8 and 14 h. **b**, Average size of the dualPIM puncta at 8 h. **c**, DualPIM degradation curve.

6. Expanded Figure 7, panels b and c: The statistics should be based on the data sets that are compared and on which conclusions are drawn: Therefore, the statistics here should be done on the comparison of the effect of siRNAs versus control and not on the comparison of -/+ BAF. Also, the LC3-II/LC3-I ratio of controls is quite different in panel b (CM+BAF \approx 1) compared to panel c (CM+BAF \approx 6), but very homogeneous for the replicates within each data set. Do the authors have an explanation for this?

Mislabeled in panel g: siFIP200 +siPSMC5 image is mislabeled, corresponding graph is correct.

We calculated and added the statistical significances as requested. The difference between the two controls is due to the slightly different exposure times for the detection of the bands, which might have given the impression of a different LC3-II/LC3-I ratio. However, the quantification shows that they are not statistically different upon autophagy stimulation. Moreover, there is a similar increase, approximately 3-fold, in both controls (from 0.3 to 0.9 in panel b, and from 2 to 6 in panel c) when comparing the differences between CM and CM+Baf.

The mislabeling has been corrected; thank you.

7. The discussion lacks clarity. The authors do not mention the substrate specificity of DNAJs, something they should be well aware of. Consequently, they do not differentiate in the discussion between the observed effects on amorphous and fibrillar aggregates, even though studies have shown that there is a strong substrate specificity of DNAJs. DNAJB1 simply might not recognize amorphous dualPIM aggregates, but this does not mean that it is generally not involved in disaggregation of foci. Moreover, the effect of siDNAJB1 was not even tested on HTT-polyQ119-EGFP aggregates in Figure 6, therefore the authors cannot claim that it has no effect here. Because the Hsp70 disaggregation system is somewhat redundant regarding the collaborating DNAJs (DNAJB1, DNAJB4, DNAJA1 and DNAJA2 do all show disaggregation activity in conjunction with Hsp70 and Hsp110), it could be that a KD of a single DNAJ does not yet have a major effect because the others take over. The authors need to consider this in their discussion.

This is an excellent point, which we have now addressed. We have analyzed effects of knocking down a couple of combinations of DNAJ proteins (i.e., DNAJA1+DNAJA2+DNAJB1 and DNAJA2+DNAJA1)(Fig. 23 for reviewer). The results show that these combinatorial knockdowns of these DNAJ proteins do not impair the dualPIM aggregate size and degradation. We have included these data in Extended data Fig. 2f-h.

Moreover, we added to our discussion the possibility that certain other combinations of DNAJ proteins might still have an impact.

Finally, we have also examined DNAJB4 knockdown and detected no differences in dualPIM degradation or aggregate size in comparison to the control

(Figure 24 for reviewers). The DNAJB4 knockdown efficiency was determined here with RT-qPCR since we could not detect DNAJB4 with our antibodies. These data are now included into Fig. 1 (j-i) and Extended Data Fig. 2b.

8. Figure 1c: last panel is wrong: no overlay of EGFP channel with merged channel

We apologize for this mistake. The correct image for the EGFP channel has now been used.

9. Figure 6b: Quantification of 'wash+VER' is missing.

Sorry: this quantification has been added now.

10. Mislabeling in Extended data Figure 1: there are two 1e, but no 1i. References in text don't match.

We have corrected the labelling of the figure and now the citations in the text are matching.

11. Figure panels should match order of appearance. Fig. 2c is mentioned before figure 2a and b. Please change order.

This has been fixed by deleting the first citation of Fig. 2c because it was unnecessary.

12. Line 67: Typo: expressing aggregate-prone proteins

This has been corrected.

13. Line 80-82: The sentence 'So far, only DNAJA1 and DNAJB1 (or their combined action) have been shown to participate in the disaggregation of amorphous or amyloid fibril-containing aggregates' is not correct: also DNAJA2 and DNAJB4 have been shown to participate in disaggregation (see papers of Nillegoda and Nachman).

We apologize for this omission. This information has now been added to the text by changing "only DNAJA1 and DNAJB1 (or their combined action)" into "only DNAJA1, DNAJA2, DNAJB1 and DNAJB4", and the related references cited.

14. Line 112-116: ‘From 2 h onwards, the first lysosomal degradation events, i.e., appearance of red-only puncta, were detected. These events progressively increased with time after rapalog2 treatment, with lysosomal localization being complete approximately one day post-rapalog2 treatment (Extended Data Fig. 1b)’: This sentence should refer to Extended Data Fig. 1b-d to include quantification and not only 1b.

This has been changed as suggested.

15. The statement in line 134-135 saying: ‘...supported by the observation that although almost all the cytoplasmic dualPIM aggregates are completely cleared within 19 h (Extended Data Fig. 1b),’ is exaggerated, as it is not clear from Fig. 1b that the aggregates are cleared. This figure shows that most of the aggregates are in lysosomes (red color), cleared would mean that they are degraded, which is not the case.

We agree with this comment. We have changed the text from “completely cleared” into “delivered into lysosomes”.

16. Line 150-154: ‘In contrast, hindering components of the cytoskeleton network such as the actin filaments and microtubules, which have been involved in the formation and disassembly of aggresomes, stress granules and JUxta Nuclear Quality control compartment (JUNQ) bodies, using nocodazol and cytochalasin D, respectively did not affect the fragmentation of dualPIM aggregate and only moderately affected their degradation rates (Extended Data Fig.1 f-h).’: Typo in nocodazole; also re-order nocodazole and cytochalasin D to match the order of actin filaments and microtubules.

We have corrected the text as suggested.

17. Line 386: Figure 6c and d: are in capital letters (possibly not the only time this is wrong).

Corrected.

Point-by-point rebuttal

We thank the three reviewers for their appreciation of our revision work and the very positive evaluation of the new version of our manuscript. A point-by-point response to reviewers' remaining minor comments is given below. Major additions and changes in the manuscript are highlighted in blue.

Reviewer #1

The authors present in this revised manuscript additional new results that support their findings. I am overall content with the rebuttal, the revised version addressing previous comments and remain enthusiastic about this work. A number of minor comments below, which should not delay acceptance for publication.

We thank this reviewer for the very positive judgement of our revised manuscript and we would like to thank her/him for helping in ameliorating it.

Point 1 – I'm not sure what the reason is for the authors to omit Fig. 1 for reviewers, which may be included with the rationale for using dualPIM in this assay.

We decided to not include these quantifications, since they do not provide additional information to the already presented data. Moreover, adding them would require that we provide similar quantifications to all the figures in which we analyzed the degradation of the dualPIM puncta. This would add many more figure panels to the manuscript, without adding valuable information. Considering that we already have a very extensive manuscript with many figure panels, we decided not to do so.

Point 2 – this is supportive of expectations. I appreciate that the authors call this 'fragmentase activity' in their response. The authors may consider using this term more frequently in the manuscript and highlight the distinction to disaggregase.

We have now revised the text and employed the term fragmentase activity three additional times in the Discussion section.

Point 3 – this is a comprehensive set of knock-downs. It may be worthwhile to mention that

Ub-GFP line likely mediates degradation via the UFD pathway, where 20S CP would be required for degradation, whereas readout by puncta size is reflective of fragmentase activity that may or may not require 20S activity.

We have now provided this explanation at page 11, line 199, where we describe the Ub-R-GFP cell line.

Point 4 – Based on the response, I think the authors mean to refer to compaction of aggregates rather than ‘condensation’, which suggests to me phase change or phase separation. If FRAP suggests no exchange, i.e. no recovery, is this still protein condensation in the canonical sense? Do the authors mean/think, with reflection on Fig.7, that it’s possible proteasomes have condensed as has been proposed by e.g. Yasuda et al. Nature 2020?

We truly appreciate this suggestion and, to avoid confusion, have changed the word condensation into compaction throughout the manuscript.

Point 5 – It is good to see the lysotracker experiment. I also enjoyed seeing proteasome particles separated on native gel, and expected perturbations to proteasomal assemblies with the knock-downs. Perhaps another way to phrase this would be ‘knock-down of 19S subunits increased the relative abundance of 20S’ and vice versa. Overall, the knock-downs did affect the overall level of proteasome particles.

I would be cautious though on conclusions one can draw from proteasome activities here. The in-gel digest is, at best, comparing between different proteasome (enzymes) at a single LLVY-AMC (substrate) concentration. This is insufficient to draw conclusions about true enzymatic activities, which would need to be compared using catalytic efficiency (kcat/Km), which in turn requires measurements at various substrate concentrations. Perhaps leave out the enzymatic activities altogether here to be vigilant.

To avoid overinterpretations, we re-phrased the paragraph according to the reviewer’s suggestion by providing a more careful conclusion about the proteasomal proteolytic activity in the Results section at page 13.

Point 6 – I think Figure 7 provides evidence for direct involvement of 19S targeting aggregates and should be considered for publication. Presence of free RPs is not surprising and I notice the authors already cited ref 99; as did they cite ref 67 showing in vitro proteasomal fragmentation activity against aggregates.

We agree with this suggestion and thus we have included this figure as new Extended Data Fig. 5g and described the findings in the Result section at page 13. We have also added the description of the pcDNA5-FRT-TO-mCherry-APEX2-PIM plasmid and the FLP-IN mCherry-APEX2-PIM U2OS cell line that we used for the experiment described in this figure in the Method section (pages 51 and 52). Finally, the person that has generated and validated the FLP-IN mCherry-APEX2-PIM U2OS cell line has been include within the authors.

Reviewer #2

I would like to thank the authors for their extensive efforts in strengthening the manuscript with new experiments. Most of the concerns have been addressed.

We thank this reviewer for these very positive remarks and the help in improving our manuscript.

However, one main concern remains, relating to the interpretation of the new results shown in Figure 3a-b. The authors demonstrate in Figure 3b that the extent of proteasome inhibition differs considerably between the individual subunits. Of concern is that PSMC1 and PSMC5 (~0.8) subunits are twice as efficient as PSMB2 and PSMB5 (~0.4) at blocking proteasome function: These are the subunits later used for 19S and 20S. Therefore, it cannot be excluded that a certain threshold of proteasome inhibition needs to be met before fragmentation is truly prevented. In line with this, the new experiments in Extended Data Figures 4a and 4b show very similar trends. Inhibiting the catalytic activity of the proteasome after 6 hours with 25 nM BTZ results in only a ~0.6-fold increase in proteasome activity reporter, while 100nM BTZ results in a ~0.8-fold increase. This is reflected in the size correlation of the aggregate: while 25 nM BTZ shows no phenotype, 100 nM BTZ already results in a significant decrease in fragmentation, suggesting that indeed proteasome activity is important.

In order to support their claim that the core function of the proteasome is not important for fragmentation, the authors should ensure that the compared knockdowns have a similar impact on the function of the proteasome. However, in my opinion, the manuscript has many novel aspects, so the necessity of core proteasome function does not need to be discussed in such detail. The fact that fragmentation is necessary for autophagy is already a very important

advancement. I suggest toning down the statement of the proteasome core activity and discussing the limitations of these results in the 'Discussion' section.

We agree with these considerations and therefore we have toned down our claim that the 20S CP is dispensable for the fragmentation function of the proteasome in the Discussion section at page 22.

Reviewer #3

The manuscript “A chaperone/proteasome-based fragmentation machinery essential for aggregophagy” by Mauthe et al. investigates how amorphous protein inclusions are processed for autophagic degradation. The authors show that the 19S regulatory particle of the proteasome and the chaperones DNAJB6, HSP70, and HSP110 are required for fragmenting large intracellular inclusions.

The manuscript has improved significantly. I appreciate the authors' efforts; I have rarely seen such a comprehensive revision. Most points I raised in my initial review, have been addressed sufficiently in the revised version. That said, two points still need further attention:

We thank this reviewer for both the appreciation of the new version of the manuscript and our effort in addressing all the comments. Thank you also for the valuable suggestions that have helped us in this task.

1. HSPA8 vs. HSPA1A (previous point 3):

Thank you for performing these experiments. It is very interesting that depleting HSPA8 (Hsc70) had no effect on dualPIM fragmentation, while HSPA1A (Hsp70) does. In contrast, disaggregation by DNAJB1-Hsp70 systems is more efficient when Hsc70 is involved, as shown by Gao et al., Mol. Cell, 2015: “DNAJB1-Agp2 combinations containing Hsc70 solubilize fibrils better than those containing stress-inducible Hsp70 (Figures S1A and S1B)”. That difference is important and should be included in the final manuscript, not just in the reviewer response, as it adds valuable insight into chaperone specificity in this context and highlights the difference between the DNAJB1-Hsp70-HSP110 disaggregation system and the fragmentation of large intracellular foci described here.

As requested, we have added these data as new Extended Data Fig. 2b to the manuscript (original Extended Data Fig. 2a is now moved to Extended Data Fig. 1j), describing them at page 9 in the Result section.

2. Statistical analysis (previous point 4):

Performing multiple separate t-tests without correction introduces a risk of inflated Type I error due to multiple comparisons. Even if the treatments are not directly compared to one another, repeatedly testing against the same control group increases the overall likelihood of detecting a false positive. For example, if four drug treatments are each compared to the same control using separate t-tests at a significance level of $\alpha = 0.05$, the chance of making at least one Type I error across all tests rises to nearly 19% (calculated as $1 - (1 - 0.05)^4 \approx 0.185$). This means that even if all null hypotheses are true, there is an 18.5% probability of falsely concluding that at least one treatment has a significant effect. This is well above the intended 5% error rate. To avoid this, the authors should use a one-way ANOVA with a post hoc test like Dunnett's (ideal for multiple comparisons against a control). If the authors prefer to use multiple t-tests, p-values should be adjusted (e.g., Bonferroni correction). Alternatively, if all the experiments were done separately with independent controls, that should be made clear in the figure, text and tests.

We have now adjusted our statistics using one-way ANOVA with the Dunnett's post hoc test or correcting the p-values using the Bonferroni correction when multiple treatments were compared to the same controls. We have also explained this in the Methods section (at page 65) and indicating its use in every pertinent figure legend.